

# Variability of $CO_2$ and $CH_4$ in a coastal peatland rewetted with brackish water from the Baltic Sea derived from autonomous high-resolution measurements

Daniel L. Pönisch[1,3], Henry C. Bittig[1], Martin Kolbe[2], Ingo Schuffenhauer[2], Stefan Otto[1], Peter Holtermann[2], Kusala Premaratne[1], and Gregor Rehder[1]

[1] Department of Marine Chemistry, Leibniz Institute for Baltic Sea Research Warnemünde (IOW), Rostock, Germany
[2] Department of Physical Oceanography and Instrumentation, Leibniz Institute for Baltic Sea Research Warnemünde (IOW), Rostock, Germany
[3] Department of Bioeconomy, Fraunhofer Institute for Computer Graphics Research (IGD), Rostock, Germany

*Correspondence to*: Daniel L. Pönisch (daniel.poenisch@igd-r.fraunhofer.de)

**Abstract.** Rewetting peatlands is an important measure to reduce greenhouse gas (GHG) emissions from land use change. After rewetting, the areas can be highly heterogeneous in terms of GHG exchange and depend on water level, vegetation, previous use, and duration of rewetting. Here, we present a study of a coastal peatland that was rewetted by brackish water from the Baltic Sea and thus became part of the coastal shallow Baltic Sea water system through a permanent hydrological connection. Rewetting suppresses carbon dioxide ($CO_2$) emissions by preventing aerobic decomposition of organic matter. Conversely, the anoxic conditions in the soil lead to an increase in methane ($CH_4$) emissions, which counteracts $CO_2$ mitigation effects. Unlike to rewetting with freshwater, the effects of rewetting with brackish, sulfate-containing water are less studied, although positive effects are expected as sulfate-reducing bacteria may become established and might out-compete methane-producing archaea (methanogens) for substrates, resulting in lower $CH_4$ emissions. Both aspects, environmental heterogeneity and the brackish water column formation, require improved quantification techniques to assess local sinks and sources of atmospheric GHGs. We conducted nine weeks of autonomous and high-resolution, sensor-based bottom water measurements of marine physical and chemical variables at two locations in a permanently flooded peatland in summer 2021, and derived GHG fluxes. Results show considerable temporal fluctuations of $CO_2$ and $CH_4$, expressed as multi-day, diurnal and event-based variability and spatial differences for variables dominantly influenced by biological processes. The multi-day variability resulted in a pronounced magnitude of measured GHG partial pressures during the deployment ranging between 295.0–8937.8 µatm ($CO_2$) and 22.8–2681.3 µatm (correspond to 42.7–3568.6 nmol $L^{-1}$; $CH_4$), respectively. In addition, the variability of the GHGs, temperature, and oxygen was characterized by pronounced diurnal cycles, resulting e.g., in a mean daily variability of 4066.9 µatm for $CO_2$ and 1769.6 µatm for $CH_4$. The diurnal variability led to a pronounced discrepancy between the measurements during the day and at night as well as depending on the location, resulting in $CO_2$ and $CH_4$ fluxes that varied by a factor of 2.1–2.3 and 2.3–3.0, respectively. The rewetted peatland was further impacted by fast system changes (events) such as storm, precipitation and major water level changes, which impacted biogeochemical cycling and GHG partial pressures. The derived average GHG exchange amounted to $0.12 \pm 0.16$ g $m^{-2}$ $h^{-1}$ ($CO_2$) and $0.51 \pm 0.56$ mg $m^{-2}$ $h^{-1}$ ($CH_4$), respectively. These fluxes are high ($CO_2$) to low ($CH_4$) compared to studies from temperate peatlands rewetted with freshwater. Comparing these fluxes with the previous year (i.e., results from a reference study), the fluxes decreased by a factor of 1.9 and 2.6, respectively. This was potentially due to a progressive consumption of organic material, a suppression of $CH_4$ production, and aerobic and anaerobic oxidation of $CH_4$, indicating a positive evolution of the rewetted peatland into a site with moderate GHG emissions within the next years.





## 1. Introduction

Mitigating climate change requires a reduction in anthropogenic emissions of the greenhouse gases (GHGs) carbon dioxide ($CO_2$) and methane ($CH_4$) and the effective removal of $CO_2$ from the atmosphere (IPCC 2022). Pristine peatlands and shallow coastal regions can act as sinks for $CO_2$ because they can store large amounts of carbon (C) when primary production exceeds
mineralization and when organic matter (OM) is buried for long-term under anoxic conditions in peat soils (e.g., Mcleod et al., 2011; Harenda et al., 2018). One land use strategy that has emerged in recent years as an appropriate measure to reduce GHG emissions, particularly $CO_2$ emissions, is the rewetting of drained peatlands. In the temperate regions, many of these peatlands are in coastal areas where they are exposed to sea level rise and extreme weather events (UNEB 2022, 2022). This leads to an increased connectivity at the terrestrial-marine interface (e.g., Jurasinski et al., 2018). As a result, coastal peatlands
and their catchment areas are vulnerable to flooding and may become a part of shallow coastal waters due to passive or active inundation.

In general, GHG exchange in peatlands is sensitive to changes in the prevailing physical and biochemical conditions in the soil. For example, the water level, temperature, and vegetation mainly control the availability of oxygen ($O_2$) and thus the extent of oxic and anoxic zones (e.g., Parish, 2008; Kaat and Joosten, 2009). Pristine peatlands that are permanently or
frequently saturated with water act as natural sinks for $CO_2$, as organic C is sequestered in anoxic zones, which results in the occurrence of peat accumulations. In turn, the anoxic conditions of waterlogged peat are very favorable for $CH_4$ production, making global peatlands a moderate source of methane of around 30 Tg $CH_4$ yr$^{-1}$(Frolking et al., 2011). Freshwater and coastal ocean $CH_4$ sources have recently been identified as major contributor to uncertainty for the global atmospheric methane budget (Rosentreter et al., 2024). When considering the total C budget, undisturbed peatlands are a weak C-sink with around
100 Tg C yr$^{-1}$ (Frolking et al., 2011).

Drainage and lowering of the water column lead to infiltration of $O_2$ into the peat layers and aerobic decomposition of OM (Joosten and Clarke, 2002). As a result, drained peatlands become a strong source of $CO_2$, with reduced or negligible $CH_4$ emissions at water levels < 20 cm, since its production is strongly coupled with the water level (Kaat and Joosten, 2009). Drainage and prolonged decomposition also lead to a lowering of the general ground level, and especially drained coastal
peatlands are therefore often below sea level.

Rewetting of degraded peatlands reduces $CO_2$ emissions by preventing aerobic decomposition of OM and is considered as one of the potent measure to mitigate global warming through land-use change. In the long-term, a re-establishment of the natural C-sink function could remove $CO_2$ from the atmosphere. The practice shows that rewetting with freshwater often leads to increased $CH_4$ emissions, while $CO_2$ emissions can remain high, at least certain time after rewetting
(Hahn-Schöfl et al., 2011; Hahn et al., 2015; Franz et al., 2016). These observations are primarily related to a wide range of preconditions and rewetting strategies, and probably at least partly a transient phenomenon. Another strategy is the rewetting coastal peatlands with brackish water. However, the effects of brackish water on GHG emissions are still unclear, although beneficial effects such as lower $CH_4$ emissions compared to rewetting with freshwater are likely due to the availability of sulfate ($SO_4^{2-}$). Furthermore, the flooding with brackish water can reduce $CO_2$ emissions by avoiding the former aerobic peat
decomposition, although this could promote the activity of sulfate-reducing bacteria (SRB). SRB can limit $CH_4$ production, as they outcompete methane-producing microorganisms (methanogens) for substrates (Segers and Kengen, 1998; Jørgensen, 2006; Segarra et al., 2013). Further, the availability of $SO_4^{2-}$ favors anaerobic oxidation of methane (AOM), which could keep $CH_4$ emissions low (e.g., Boetius et al., 2000; Knittel and Boetius, 2009).

The spatial and temporal heterogeneity of environmental conditions is particularly noticeable in the coastal
ecosystems of the Baltic Sea and is determined, for example, by the rapid cycling of elements (HELCOM, 2018; Kuliński et al., 2022) and regular diurnal cyclicity (Honkanen et al., 2021). This becomes more pronounced when coastal peatlands have been rewetted with water from the Baltic Sea, thereby becoming part of the coastal water system of the Baltic Sea, as this results in very shallow subsystems that are rich in organic material and are in a transitional state. Those conditions are




challenging for conventional sampling approaches, such as discrete water samplings, which cannot adequately resolve the

temporal heterogeneity. As a result, systematic long-term data on GHGs with an appropriate resolution are very rare, and studies focus on open waters, estuaries, and discrete samplings (one-point recording). Consequently, coastal zones are not well-implemented in the global C budget due to difficulties in scaling processes (Saunois et al., 2020). Therefore, new techniques and interdisciplinary approaches are essential to address the blind spots and to establish a better monitoring of coastal regions. One possible approach for studying the highly dynamic coastal environment and evaluating the effectiveness

of rewetting strategies is the use of autonomous in situ sensors, which can significantly increase the temporal resolution of data collection but also face problems such as limited battery power (Pönisch, 2023).

In this work, two lander systems were used, which are designed as a platform to deploy a wide range of marine sensors and can be customized for different applications. The systems can be considered as modern, advanced technology for underwater monitoring applications, where sensor operation and sensor data acquisition are managed by a central data

processing unit, which enables efficient processing of incoming data even during long-term missions.

We performed high-resolution sensor measurements in the range of seconds and minutes of physicochemical variables in a flooded peatland over a period of around nine weeks in the summer of 2021. The rewetting of the coastal peatland was achieved in November 2019, two years prior to this study, by the active dredging of a channel in the dike (Pönisch and Breznikar et al., 2023). This led to the formation of a permanent brackish water column and regular water exchange with the

*Kubitzer Bodden* (Baltic Sea). The high-resolution measurements were combined with discrete sample analysis and GHG emissions of $CO_2$ and $CH_4$ were derived. The focus of this study is on exploring the time scales for the variability of GHG distribution and its drivers, as highly variable conditions are assumed, which may occur, for example, as pronounced diurnal cycles. The impact of the temporal variability on the estimation of GHG emissions or with respect to discrete sampling strategies was assessed. By comparing GHG fluxes with a study conducted one year earlier (i.e., 2020, the first year after

rewetting; Pönisch and Breznikar et al., 2023), the potential evolution towards further weakening of the $CO_2$ and $CH_4$ source strength is discussed.

## 2. Material and Methods

### 2.1 The landers

Two submersible stationary landers were used for autonomous multi-parameter investigations in the shallow water (~ 0.5 m)

of the rewetted peatland through integrated high-resolution sensor measurements. Each system was equipped with state-of-the-art sensors that measured the partial pressures of $CO_2$ and $CH_4$ in the water, temperature, salinity, hydrostatic pressure, oxygen ($O_2$), turbidity, and the concentrations of nitrate ($NO_3^-$), phosphate ($PO_4^{3-}$) and chlorophyll *a* (Figure A 1). Sensor scheduling, time stamping, and data recording were centralized in the Data Processing Unit (DPU) and allows customized deployments. Despite of a total weight of ~ 250 kg, the modularity of the landers makes them ideal for areas that are highly

dynamic and therefore cannot be studied with discrete samples. The landers were deployed and maintained using a small working boat, floats and a lift system (Figure A 2). Both landers have already been described and deployed in a near-shore area of the Baltic Sea (Pönisch, 2023), but the system has been adapted in terms of power supply and communications (Figure A 1):

-The landers were powered by a professionally constructed 60 V land station near the observatories together with a

400 m and 800 m cable. To compensate consumption peaks of the sensor a buffer battery was included.

-The wired power cable for each lander was additionally used to establish a powerline communication and for SFTP and HTTP requests. This enabled on-air schedule adjustments, data access, and remote controllability.

-An IP socket was additionally used for power cycling (reset) of the systems.



The deployment took place between 02 June 2021 and 09 August 2021 and ~ 500,000 data points were collected for each $p$CO$_2$

and $p$CH$_4$ sensor, respectively. The time of deployment was chosen based on the study of the annual cycle from the first year after rewetting (based on weekly to bi-weekly sampling), which indicated that the summer season is the most important and dynamic with respect to GHG-fluxes (Pönisch and Breznikar et al., 2023).

### 2.2 The study site and lander locations

The measurements were conducted in the former *Polder Drammendorf*, a coastal peatland that was flooded with

brackish water by an active removal of the protection measures in November 2019. The site is located at the southern Baltic Sea coast (Figure 1a–b). The rewetting transformed the formerly drained and agriculturally used area into a permanently flooded brackish wetland with an estimated mean water column of ~ 0.5 m (Figure 1c). Flooding was accomplished by excavating an approximately 20 m wide section of the dike, creating a channel towards the *Kubitzer Bodden*, which in turn is connected via the *Bodden* chain to the Baltic Sea (Figure 1). The peatland was flooded immediately since decades of peat

degradation formed a land depression that is below the mean water level of the adjacent *Kubitzer Bodden*. The height of the water column of the flooded peatland depends on the water level of the micro-tidal *Kubitzer Bodden* and is described to be very dynamic (Pönisch and Breznikar et al., 2023). The exchange of water between the peatland and the *Kubitzer Bodden* takes place only via the channel, hence the hydrological connection acts as a transport route for dissolved compounds, such as nutrients or GHGs. An established hypsographic curve of the drained area allows calculation of water volume change in from

recorded sea level data; for details see Pönisch and Breznikar et al., 2023.

The topsoil of the rewetted area is characterized by highly degraded peat up to 50–70 cm, with a well-preserved peat layer of ~ 100 cm underneath (Brisch, 2015). The topsoil and vegetation were not removed prior to rewetting, and in the first months after rewetting, the former grassland and ditch vegetation (*Elymus repens* L. (Gould) and *Phragmites australis* (Cav.) Trin. ex Steud.) almost completely died out as a reaction of brackish inundation. As a result, the study site was initially

characterized by high rates of carbon and nutrient cycling. The transition from dry to flooded conditions has been described in detail in (Pönisch and Breznikar et al., 2023).

Lander 1 was deployed in the central part of the flooded peatland at a mean water depth of ~ 0.6 m (Figure 1c). Despite the frequent changes in the water level height (Figure 2e), it can be assumed that the lateral water velocities are low due to the exposed location and that the effects of the processes associated with the peatland are pronounced. Lander 2 was

installed in the middle of the excavated channel at a mean water depth of ~ 0.9 m and thus at the interface between the *Kubitzer Bodden* and the peatland. This location allowed for the monitoring of the biogeochemical properties of the exchanging water masses. Due to the narrow channel and as indicated by a previous investigation (Pönisch and Breznikar et al., 2023), high, intermittent water flow velocities were assumed.





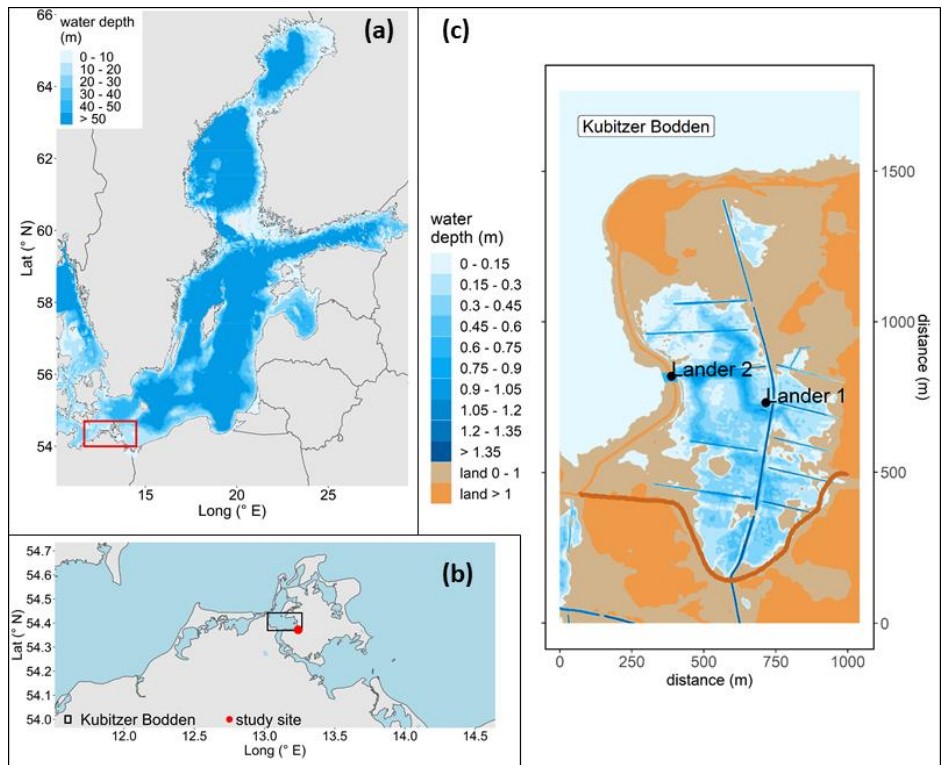

**Figure 1:** Location of the study site showing (a) the Baltic Sea with the study site located in the southern Baltic Sea, (b) the coastline of northeastern Germany with the study site located on the island of *Rügen* and its hydrological linkage to the *Kubitzer Bodden*, and (c) topography and water coverage of the study area at mean sea level as well as the locations of the lander 1 (central area) and lander 2 (sea-peatland interface). The dark brown line in the south shows the dike that was built before rewetting. Bathymetry refers to Seifert et al., 2001, and borders were retrieved from National Oceanic And Atmospheric Administration (NOAA) and National Centers For Environmental Information (NCEI).

### 2.3 Instrumentation

#### 2.3.1 $p$CO$_2$ and $p$CH$_4$ measurements

Measurements of $p$CO$_2$ and $p$CH$_4$ with a logging interval of 10 seconds (s) were carried out by submersible CONTROS HydroC® CO$_2$ (HC-CO$_2$) and CONTROS HydroC® CH$_4$ (HC-CH$_4$) sensors (-4H- JENA Engineering GmbH, Jena, Germany). Equilibration of gases in the seawater and the headspace in the sensor were achieved through a planar membrane and the target molecules were detected by non-dispersive infrared spectrometry (NDIR; CO$_2$) and tunable diode laser absorption spectroscopy (TDLAS; CH$_4$), respectively (Fietzek et al., 2014). Further details on operation mode, sensor calibration, data post-processing, and quality assessment are described in Appendix C1–C3.

#### 2.3.2 CTD-O$_2$ measurements

Conductivity, temperature, pressure (CTD) and dissolved O$_2$ measurements were carried out with a SBE 37-SMP-ODO MicroCAT instruments (CTD-O$_2$; Sea-Bird Electronics Inc., Bellevue, WA, USA). The resolution of data acquisition was set to 5 minutes (min) for lander 1 and 10 min for lander 2. More information about sensor calibration, data post-processing and data availability are provided in Appendix C4.





### 2.3.3 Turbidity and chlorophyll *a* concentration measurements

Turbidity and chlorophyll *a* concentrations were measured using SBE-FLNTUSB-ECO sensors (Sea-Bird Electronics Inc., Bellevue, WA, USA). Measurements were carried out with ~ 1 Hz for 240 s followed by a 60 s sequence of bio-wiper / UV light treatment (information about the used UV light treatment is given in Appendix A2). The values were post-processed according to the calibration data provided by the manufacturer and then the average values were calculated for each interval, which consisted of ~ 240 individual measurements.

### 2.3.4 Nutrient ($NO_3^-$, $PO_4^{3-}$) measurements

Nitrate concentrations were measured using SBE SUNA V2 instruments (Sea-Bird Electronics Inc., Bellevue, WA, USA) with a resolution of measurements of 10 min. The SUNA UV absorption spectra were processed according to Sakamoto et al., (2009) based on the calibration data provided by the manufacturer. Post-processing details are shown in Appendix C5.

Dissolved phosphate was measured using SBE HydroCycle-$PO_4$ sensors (Sea-Bird Electronics Inc., Bellevue, WA,
USA) which were powered with an external battery and data logger. The phosphate sensors were not powered by the DPU due to technical restrictions. The resolution of the measurements was 60 min, with internal on-board calibration after six determinations. A 5-point laboratory calibration was performed prior to deployment and the field data were corrected with a linear regression model ($R^2 = 0.9979$ and $0.9987$, respectively).

### 2.3.5 Water velocity measurements at lander 2

Lander 2 (position in the connecting channel) was additionally equipped with an upward-facing acoustic profiler (Aquadopp 1 MHz, Nortek AS, Norway) that was installed approximately 0.3 m above the sediment. The profiler recorded water velocities between 02 June 2021 and 29 June 2021, in 50 mm cells at a sampling frequency of 1 Hz. The first bin was approximately 0.6 m above the bottom. In a second deployment a different setup of bursts of 600 s sampled with 4 Hz every 1200 s was programmed without changing the cell size. The second deployment was between 29 June 2021 and 29 July 2021. For the
calculation of average velocities, the valid bins within the water column were first vertically averaged and then averaged over 120 s for the first deployment. For the second deployment the bursts of 600 s were averaged, yielding in a vertically and time averaged velocity every 10 min. To distinguish between in- and outflow into the peatland, the velocities were sorted such, that all velocities with a direction between 30 ° and 200 ° is an outflow, meaning that the water is flowing from the peatland to the *Kubitzer Bodden* (outflow; positive sign convention). Every other direction is defined as an inflow into the peatland.

### 2.4 Discrete field sampling and laboratory analysis of pH, total $CO_2$ ($C_T$), total alkalinity ($A_T$), $CH_4$, nutrients ($NO_3^-$, $PO_4^{3-}$)

To validate the sensor-based measurements, discrete field measurements were taken during the lander deployment at lander 1 (in the central peatland area) and at lander 2 (in the connecting channel). Altogether 9-sampling sessions were carried out in the direct proximity to the landers (Table C 1; bottle data for all discrete sampled variables can be found at
https://doi.org/10.1594/PANGAEA.964758; (Pönisch et al., 2024). Undisturbed water was taken manually using a 5-L Niskin bottle. The bottle was deployed horizontally from a small working boat and its closure noted with an exact time stamp. Bottle data were obtained using established laboratory methods as described below.

Subsamples from the Niskin bottle were taken by sample overflow for $CH_4$ (250 mL), pH (250 mL), and total $CO_2$ ($C_T$) / total alkalinity ($A_T$; 250 mL), poisoned with 500 µL ($CH_4$) and 200 µL ($CO_2$ system) of saturated $HgCl_2$ solution,
respectively, and were stored at 4 °C in dark conditions. Nutrient subsamples ($NO_3^-$, $PO_4^{3-}$; 200 mL) were filtered in the field with combusted glass-fiber filters (0.7 µm, GF / F, Whatman®) and stored at − 20 °C. Discrete samples for water temperature and salinity were measured in the field with a HACH HQ40D multimeter (HACH Lange GmbH, Düsseldorf, Germany) equipped with an electrode (CDC401).



The pH was analyzed by a spectrophotometric approach as described by Dickson et al., (2007) and Carter et al., (2013), using the pH-sensitive indicator dye m-cresol purple, and are reported on total scale. $C_T$ was measured with an automated infrared inorganic carbon analyzer (AIRICA; Marianda, Kiel, Germany). $A_T$ was measured by potentiometric titration in the open-cell configuration described by Dickson et al., (2007). Dissolved $CH_4$ concentrations were determined using the gas chromatograph Agilent 7890B (Agilent Technologies, Santa Clara, USA) coupled with a flame ionization detector (FID) and based on the purge-and-trap technique (in-house designed periphery) as described in (Sabbaghzadeh et al., 2021; Pönisch and Breznikar et al., 2023). More details on the individual laboratory methods can be found in Appendix B. The analysis of $NO_3^-$ and $PO_4^{3-}$ were carried out via standard photometric methods (Grasshoff et al., 2009), using a continuous segmented flow analyser (Seal Analytical QuAAtro, SEAL Analytical GmbH, Norderstedt, Germany). Precisions and detection limits were $\pm 4.6\%$ for $NO_3^-$ and $\pm 4.0\%$ for $PO_4^{3-}$ and $0.2\ \mu mol\ L^{-1}$ for $NO_3^-$ and $0.1\ \mu mol\ L^{-1}$ for $PO_4^{3-}$, respectively.

## 2.5 Air-Sea exchange (ASE) calculation

The exchange (F) of $CO_2$ and $CH_4$ across the air-sea interface was calculated from the high-resolution measurements of $pCO_2$ and $pCH_4$ using the general boundary layer flux equation (2.1) and the air-sea exchange parameterization of Wanninkhof, (2014). The air-sea flux is determined by the difference in concentrations of the bulk liquid ($c_w$) in which the sensor measurements were carried out and the top of the liquid boundary layer adjacent to the atmosphere ($c_a$).

$$F = k * (c_w - c_a) \qquad\qquad 2.1$$

The gas transfer velocity ($k$) describes the efficiency of the transfer process and is parameterized as a function of the wind-speed $<U^2>$, an empirical relationship for a gas transport coefficient (0.251) and the Schmidt number (Sc; equation 2.2).

$$k = 0.251\ <U^2>(Sc/666)^{-0.5} \qquad\qquad 2.2$$

For the flux calculations, we assumed that the sensor-based GHG partial pressures were representative of surface water, since no permanent stratification occurred and the maximum water depth of the measurements was < 1.25 m. All involved variables (i.e., GHGs, wind speed, temperature, salinity, atmospheric-equilibrium conditions) were averaged hourly to obtain more robust values and matching timestamps. Wind speeds at 15 m height were retrieved from a monitoring station in vicinity (open data portal of the *Deutscher Wetterdienst (DWD)*, *Putbus* station, distance ~ 15 km, 54.36426° N, 13.47709° E, WMO-ID 10093). Since there were gaps in the measured salinity after post-processing (Figure 2d), the values from both landers were combined by interleaving the values from lander 1 with those from lander 2. This fact was justified because the locations of the two lander are physically connected by a strong water exchange, resulting in very similar salinity conditions, as shown in a previous study (Pönisch and Breznikar et al., 2023). The Schmidt number was approximated by a linear interpolation between the freshwater and seawater values. The equilibrium concentrations ($c_a$) were calculated by using the values of atmospheric $CO_2$ and $CH_4$ mole fractions provided by the ICOS station "Utö" (Finnish Meteorological Institute, Helsinki, ICOS RI et al., 2022). The ASE model of Wanninkhof, (2014) was developed for the open ocean waters and therefore may have reduced applicability for deriving GHG fluxes from an enclosed peatland. However, to the best of our knowledge no better suitable parameterization exists. Moreover, the same parameterization was used in Pönisch and Breznikar et al., 2023, to which the data are compared.

We calculated the ASE for different sampling scenarios, to stress the effect of different methodologies and temporal sampling schemes, with results presented in Table 2:

(1) ASE derived from our fully available high-resolution time series.

(2) ASE using the bottle data from our deployment.

(3) ASE derived from our high-resolution time series by isolating a daily average for ± 1 hour from 00:00 and 12:00 UTC, respectively. This represents the day-night-bias (resulted in ~ 200 data points for each calculation).



(4) ASE derived from the published data of Pönisch and Breznikar et al., 2023, where measurements were made in 2020, one year prior to our study. We chose the period corresponding to the period of this study (e.g., summer 2020).

(5) ASE derived from our high-resolution time series by isolating a daily average between 09:00 and 15:00 UTC. This time period represents the period of bottle data sampling in Pönisch and Breznikar et al., 2023, which is used for a direct comparison of the ASE evaluation described in the number (4).

### 2.6 Data handling and analysis

Data processing, analysis and visualization were performed using R (R Core Team, 2022). The R packages which were used
to calculate $pCO_2$ (based on bottle $C_T$ and pH data), and to convert $pCH_4$ to concentrations are descried in Appendix D.

     Spearman correlation coefficients ($r_s$) were calculated to provide an overview of the first-order linear relationship between the measured variables (Figure 3). A significance level of 0.001 was applied to identify non-correlating variables (empty boxes). In addition, a p-value adjustment was performed using the R package corrplot (Wei and Simko, 2021), as it is a multivariable correlation approach. To interpret the strength of the relationship between the variables, the convention of
Cohen, (1988) was used by introducing different effect sizes of correlations. A $r_s \geq 0.5$ and $r_s \leq -0.5$ represents a large effect size and thus a strong correlation and is indicated by black stars. Accordingly, $r_s \geq 0.3$ and $r_s \leq -0.3$ represents a medium effect size and correlation and is indicated by brown stars.

### 3. Results

### 3.1 Time series of the variables at lander 1 and lander 2

An overview of the time series data for the entire deployment is given in Figure 2. For some of the variables, data is only available for parts of the time series, which can be easily derived from the figure.

     Wind speeds were persistently weak to moderate with a mean of 3.4 m s$^{-1}$ (Figure 2j) and ranged from 0.2 to 10 m s$^{-1}$, indicating that no strong storm occurred during the deployment. Furthermore, precipitation was low with a mean value of 0.08 mm h$^{-1}$ (corresponds to L m$^{-2}$ h$^{-1}$, data not shown).

The water column temperature at both landers showed considerable agreement in average and standard deviation (Table 1) as well as in the pronounced diurnal cycle (Figure 2c). Within the period of simultaneously data coverage of both CTD-O$_2$ sensors, salinity also showed only small differences (Figure 2d). In early July, salinity dropped by about 2–3, separating two periods of relatively stable salinity conditions. Furthermore, bottle data for salinity showed high agreement between both lander locations, supporting the pooling of salinity data from both landers for ASE calculation (Sect. 2.5). Taken
together, the similar temperature and salinity indicate a distinctive water exchange among both landers and hence, with the *Kubitzer Bodden*. This is supported by analysis of the hydrostatic pressure differences from both landers at 10-minute intervals. For this, we calculated an average of the high-resolution data (Figure 2e) every 10 min. The differences between both sites can be used as a proxy for water height compensation, or the speed of water level height adjustment due to water exchange. The differences were only ± 10 cm (minimum / maximum) or ± 5 cm (0.98$^{th}$ percentile / 0.02$^{th}$ percentile), respectively. We
conclude that the adjustment of the water level is quasi-simultaneous between both landers due to the strong hydrological connection. Changes in water level (driven by changes in the connected *Kubitzer Bodden*) led to frequent alterations between inflow and outflow and drove high water velocities through the connection channel (position of lander 2), with the highest measured velocity of ± 0.6 m s$^{-1}$ (Figure E 1). The frequent water level changes in alternating directions are also visible in the measured hydrostatic pressure at both landers, and hence frequent changes in the order of several cm occurred (i.e., ~ 5–15 cm;
Figure 2e). These common inflows and outflows trough the dike opening counteracted stagnant conditions due to water transport. As another important characteristic of the area of investigation, major changes in water level in relatively short



periods of time were observed: For example, in late July, the water column height at both landers rose by 0.46 m within 11.5 h, followed by a period of a rapid falling water level.

Oxygen exhibited large short-term fluctuations on a daily scale (Sect. 3.2), with saturation values ranging from ~ 0
to 180 %, indicating strong alternations between undersaturated and oversaturated conditions. Despite occasional low $O_2$ values, measurements indicate a predominantly oxygenated water column, with slightly lower mean $O_2$ values at lander 2 (Figure 2f, Table 1).

The $p$CO$_2$ varied considerably at both landers during the deployment displaying strong, multi-day, sinusoidal fluctuations (Figure 2a) and short-scale fluctuations < 1 day (d; Sect. 3.2, Table 1). Sustained lower values occurred at the
beginning of the deployment (early June), but then changed to on average higher values during most of the deployment at both locations. The general direction of change in the $CO_2$ signal on the multi-day scale was the same at both sites. While both landers exhibited comparable variability (i.e., standard deviation), lander 1 was characterized by a more than 1000 µatm higher mean $p$CO$_2$ (Table 1). At both landers, there was a negative correlation between $p$CO$_2$ and $O_2$ saturation with a strong effect size ($r_s = -0.58$, $r_s = -0.56$; Figure 3). The comparison of the $p$CO$_2$ calculated from bottle $C_T$ and pH data with the $p$CO$_2$
measured by the sensors showed both good agreements of only $-2$ % (a minus sign means lower values from the bottle data) and larger discrepancies of up to 45 % (Figure C 1). The median of the nine comparisons was 13 % for lander 1 and 12 % for lander 2.

The $p$CH$_4$ signals were characterized by strong and complex fluctuations manifested by multi-day variability during the deployment and by short-term fluctuations (< 1 d) that overlapped. The mean and SD of $p$CH$_4$ from lander 2 was slightly
higher compared to lander 1 (Figure 2b, Table 1). Correlation analysis showed that $p$CH$_4$ at lander 1 had three dominant correlations, namely a positive correlation with temperature ($r_s = 0.30$) and salinity ($r_s = 0.37$) and a negative correlation with wind speed ($r_s = -0.31$; Figure 3). At lander 2, the same correlations were visible, but with lower effect sizes. In addition, $p$CH$_4$ at lander 2 had a positive correlation with a medium effect size with the velocity of the water outflow ($r_s = 0.33$). In comparison with the bottle data, where $p$CH$_4$ was calculated from cCH$_4$, the sensor $p$CH$_4$ data showed both good agreement
with $-21$ % discrepancy but predominantly higher divergences reaching 595 % at maximum (Table C 1). The median of the nine comparisons was 83 % for lander 1 and 77 % for lander 2, with the bottle data mostly higher than the sensor data.

Comparison of mean chlorophyll $a$ concentrations and turbidity between the two landers showed slight differences, with higher values measured at lander 1 (Table 1). The time series of chlorophyll $a$ revealed a more fluctuating pattern in the second half of the deployment (Figure 2g). This observation was accompanied by a rapid decrease in salinity of about 3 within
~ 1 d, and followed by prolonged lower salinity until the end of the deployment. Accordingly, chlorophyll $a$ on both landers showed a potentially significant negative correlation with salinity ($r_s = -0.26$, $r_s = -0.27$; Figure 3). It also correlated negatively with hydrostatic pressure with a medium effect size at both landers ($r_s = -0.34$, $r_s = -0.45$). Turbidity indicated a possible positive relationship with the wind speed ($r_s = 0.36$), at least on lander 1. At lander 2, the turbidity showed a possible positive correlation with the water velocity (inflow and outflow; $r_s = 0.23$ and 0.11).

Sensor measurements of $NO_3^-$ are challenging in coastal areas because optical measurements are affected by several other absorbing species. Post-processing, including discrete coloured dissolved organic matter (CDOM) measurements (data not shown) showed that spectra obtained from the SBE SUNA V2 instruments were dominated by CDOM interferences, which was not unexpected in the flooded peatland. At present, there are no advanced methods to distinguish between CDOM interference and $NO_3^-$ absorbance for application in the Baltic Sea or a peatland area. As a result, we had to conclude that the
SUNA data are not suitable to derive $NO_3^-$ concentrations, in the present environment of very low $NO_3^-$ concentrations, since bottle data were predominantly below the detection limit (0.2 µmol L$^{-1}$; spectra data are available at https://doi.org/10.1594/PANGAEA.964839; (Pönisch et al., 2024).



The phosphate measurements gave comparable mean values for both landers, although their periods of operation were quite different. Measurement at lander 1 ranged from 0.55 to 3.43 µmol L$^{-1}$ with a plateau of elevated values for ~ 12 days in late June (Figure 2i, Table 1). The sensor data were consistent with the bottle data.





**Figure 2:** High resolution time series from lander 1 (black) and lander 2 (red) with discrete bottle data. The panels display post-processed data of (a) $p$CO$_2$ (µatm), (b) $p$CH$_4$ (µatm), (c) water temperature (°C), (d) salinity, (e) O$_2$ saturation (%), (f) pressure (depth; dbar), (g) chlorophyll $a$ concentration (µL L$^{-1}$), (h) turbidity (NTU). (i) PO$_4^{3-}$ concentration (µmol L$^{-1}$), and (j) wind speed (m s$^{-1}$) and precipitation 340 (mm). The sensor signals of $p$CO$_2$ and $p$CH$_4$ were additionally smoothed, as represented by the black and red lines, respectively (description of smoothing in given in Appendix D). The gray rectangles highlight three periods of systems changes (Sect. 3.3).



**Table 1:** Summary of mean, standard deviation (SD), minimum and maximum (as 0.98$^{th}$ and 0.02$^{th}$ percentiles) of available data from lander 1 and lander 2. Furthermore, the mean, minimum and maximum of the calculated daily variability is shown (calculation is outlined in Sect. 3.2). $CH_4$ concentration (nmol L$^{-1}$) was calculated from $pCH_4$ (µatm). $NO_3^-$ concentrations could not be determined due to strong interferences with CDOM.

| | | Lander 1 | Lander 2 |
|---|---|---|---|
| $pCO_2$ (µatm) | mean ± SD | 3399.1 ± 2348.0 | 2339.7 ± 2008.4 |
| | 0.98$^{th}$ pct – 0.02$^{th}$ pct | 295.0 – 8937.8 | 97.4 – 7164.3 |
| | mean (min – max) of daily variability | 3730.8 (172.8 – 8535.0) | 4066.9 (357.6 – 7891.0) |
| $pCH_4$ (µatm) | mean ± SD | 754.3 ± 543.1 | 843.0 ± 708.8 |
| | 0.98$^{th}$ pct – 0.02$^{th}$ pct | 74.3 – 1982.1 | 22.8 – 2681.3 |
| | mean (min – max) of daily variability | 720.2 (93.3 – 1715.5) | 1769.6 (168.7 – 3475.2) |
| cCH$_4$ (nmol L$^{-1}$) | mean ± SD | 1049.1 ± 754.5 | 1186.9 ± 948.4 |
| | 0.98$^{th}$ pct – 0.02$^{th}$ pct | 108.4 – 2677.0 | 42.7 – 3568.6 |
| | mean (min – max) of daily variability | N/A | N/A |
| temperature (°C) | mean ± SD | 21.9 ± 2.7 | 21.8 ± 2.6 |
| | 0.98$^{th}$ pct – 0.02$^{th}$ pct | 16.9 – 28.1 | 16.6 – 27.6 |
| | mean (min – max) of daily variability | 4.2 (1.6 – 8.5) | 4.0 (1.4 – 8.0) |
| pressure (mbar) | mean ± SD | 0.59 ± 0.11 | 0.89 ± 0.12 |
| | 0.98$^{th}$ pct – 0.02$^{th}$ pct | 0.23 – 0.76 | 0.51 – 1.08 |
| | mean (min – max) of daily variability | 0.16 (0.04 – 0.39) | 0.15 (0.04 – 0.39) |
| salinity | mean ± SD | 9.0 ± 0.6* | 9.0 ± 0.9* |
| | 0.98$^{th}$ pct – 0.02$^{th}$ pct | 7.9 – 10.0* | 7.5 – 10.3* |
| | mean (min – max) of daily variability | 0.3 (0.07 – 2.40)* | 0.4 (0.06 – 1.95)* |
| saturation O$_2$ (%) | mean ± SD | 75.9 ± 46.6* | 53.6 ± 40.8* |
| | 0.98$^{th}$ pct – 0.02$^{th}$ pct | 0.1 – 179.5* | ~ 0 – 144.0* |
| | mean (min – max) of daily variability | 77.9 (0.1 – 180.0)* | 70.1 (0.01 – 142.7)* |
| chlorophyll $a$ (µl L$^{-1}$) | mean ± SD | 21.3 ± 10.3 | 16.4 ± 9.8 |
| | 0.98$^{th}$ pct – 0.02$^{th}$ pct | 7.1 – 47.3 | 4.5 – 41.0 |
| | mean (min – max) of daily variability | 16.1 (1.9 – 76.2) | 28.1 (4.1 – 187.8) |
| turbidity (NTU) | mean ± SD | 19.7 ± 9.9 | 12.9 ± 14.8 |
| | 0.98$^{th}$ pct – 0.02$^{th}$ pct | 9.5 – 41.8 | 2.7 – 70.2 |
| | mean (min – max) of daily variability | 43.4 (3.5 – 323.2) | 44.9 (4.2 – 200.5) |
| $NO_3^-$ (µmol L$^{-1}$) | mean ± SD | | |
| | 0.98$^{th}$ pct – 0.02$^{th}$ pct | N/A | N/A |
| | mean (min – max) of daily variability | | |
| $PO_4^{3-}$ (µmol L$^{-1}$) | mean ± SD | 1.59 ± 0.83* | 1.37 ± 0.86* |
| | 0.98$^{th}$ pct – 0.02$^{th}$ pct | 0.55 – 3.43* | 0.24 – 3.68* |
| | mean (min – max) of daily variability | 0.59 (0.03 – 2.19)* | 1.22 (0.27 – 3.28)* |

SD stands for standard deviation; pct stands for percentile; N/A stands for not available; data marked with the asterisk * have major non-equivalent time coverage and are not comparable.

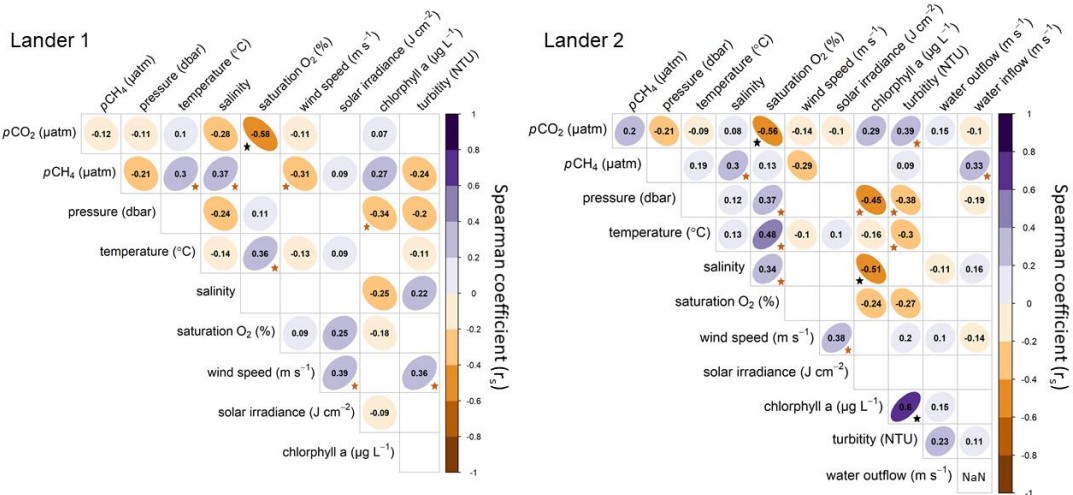

**Figure 3:** Spearman correlation coefficients (r$_s$) between the measured variables, wind speed and solar irradiance of both landers. A correlation level of 0.001 was used to remove non-correlating relationships (empty fields). In addition, the Cohen convention (Cohen, 1988)





was used to interpret the effect size. The black stars represent a large effect size and thus a strong correlation, while the brown stars represent a medium effect size. Wind speeds were retrieved from the station *Putbus* (WMO-ID 10093) and solar irradiance from *Rostock-Warnemünde* (WMO-ID 10170; both DWD).

### 3.2  Short-term variability and diurnal cycles

The variables $pCO_2$, $pCH_4$, temperature, and oxygen showed pronounced short-term variability and diurnal cyclicity, expressed in regular sinusoidal fluctuations, but sometimes superimposed by other fluctuations, especially for the GHG signals (Figure

2). In order to show the diurnal cyclicity and the relationships between the variables affected by the diurnal cycles, the high-resolution measurement time series (lander 1 and lander 2) were divided into one-hour-bins and a mean value was calculated for each hour of the day (Figure 4). The distribution of these mean values over the course of a day indicated that $pCO_2$ and $pCH_4$ showed an inverse character compared to temperature, $O_2$, and the wind speed (Figure 4). The highest mean $pCO_2$ values were observed in the early morning (05:00 UTC) and the lowest values in the late afternoon (17:00 UTC). The daily cycle of

$pCH_4$ was less pronounced, but also revealed higher values in the early morning (~ 04:00 UTC) and lower values in the late afternoon (~ 18:00 UTC). Furthermore, the wind speed was also characterized by a diurnal cycle, with higher speeds during the day and at midday, respectively. Since wind is used for the parameterization of the air-sea exchange, it has an influence on the determination of GHG emissions.

Based on these observations, the mean diurnal variability was calculated to show the magnitude of variability of the

variables. For this, we divided the high-resolution data into 24-hour intervals, each starting at midnight. Then, for each interval, the difference between the minimum and maximum was determined. Subsequent determination of the mean, minimum, and maximum yields an approximation of diurnal variability (see Table 1). The mean daily range for $pCO_2$ of ~ 4000 µatm is substantial. However, this diurnal range differed considerably during the deployment (Figure 2a), with minimum and maximum daily variability ranging from ~ 200 µatm to ~ 8500 µatm, with comparable patterns for both landers (Table 1). For

$pCH_4$, the diurnal variability was also pronounced, but with larger differences between the two landers (Figure 2b): The mean daily variability was ~ 700 µatm and ~ 1800 µatm for lander 1 and lander 2, respectively. The observed minimum and maximum during the deployment ranged from ~ 100 µatm to ~ 3500 µatm (numbers for lander 2). On average, the temperature varied by ~ 4 °C and $O_2$ saturation by ~ 70–75 % over the course of the day.

Analysis of the remaining variables (i.e., salinity, pressure, chlorophyll *a*, turbidity) revealed a low or non-diurnal

behavior (Figure E 3). The relatively high mean daily variability for chlorophyll *a* and turbidity summarized in Table 1 is impacted by a large number of spikes and less by diurnal cyclicity.





**Figure 4:** The distribution of the mean values of $p$CO$_2$, $p$CH$_4$, temperature, O$_2$, and wind speed from one-hour-parts over 24-hours. The box plots show the median and the 25 / 75 % quantiles. The whiskers indicate the 5 / 95 % percentiles, and the red points denote the mean values.

**3.3  Identification of three event-based system changes**

Analysis of the meteorological, hydrographical and biogeochemical parameters resulted in the identification of three events (Figure 2, grey shaded periods). These observations are numbered as (I), (II), and (III) and are described in the following. They serve as the time period for detailed analysis of GHG drivers for the flooded peatland. Being more representative than the location at the connection between the *Kubitzer Bodden* and the rewetted peatland (i.e., position of lander 2), only data from



lander 1 in the inner part of the formerly drained area were used. A shared observation is that all three events, although different in nature, led to a disruption or at least dampening of the diurnal cycle of $pCO_2$ and $pCH_4$, as well as a reduction of the mean $pCO_2$ and $pCH_4$ values, for several days. Furthermore, $pCO_2$ and $pCH_4$ were strongly correlated during all three episodes ($r_s = 0.73–0.86$), and some of the correlation levels to individual drivers were far more pronounced for these shorter periods (Figure E 4).

**Stormy event – (I)**

Between 12–13 June 2021, a short-lived period with sustained high-wind speeds of up to 9–10 m s$^{-1}$ was identified. After an initial decline in water level, this resulted in a rise by ~ 0.4 m with water from the *Kubitzer Bodden* entering the area of the rewetted peatland, and led to an increase in water volume by 2.4 times compared to normal water level conditions (data not shown). Simultaneously, water temperature dropped by ~ 13 K, while salinity remained unaffected. Turbidity increased and

reached a pronounced peak of ~ 60 NTU. The partial pressures of $CO_2$ and $CH_4$ decreased considerably, and the amplitude of the daily cyclicity of both GHGs and $O_2$ was substantially reduced for several days after the event. Correlation analysis revealed a number of correlations with strong and medium effect sizes (Figure E 4). For example, wind speed correlated with turbidity ($r_s = 0.48$). In addition, $pCO_2$ and $pCH_4$ correlated negatively with pressure ($r_s = -0.66$ and $r_s = -0.87$, respectively) and positively with temperature ($r_s = 0.60$ and $0.86$, respectively).

**Fast salinity decrease – (II)**

The second event covers about 48 h in early July and was characterized by an accumulated precipitation of ~ 76 L m$^{-2}$, corresponding to 1.6 L m$^{-2}$ h$^{-1}$. This resulted in a rapid (~ 24 h) decrease in salinity by ~2–3 and a concomitant outflow of water towards the *Kubitzer Bodden,* reinforced by a drop in sea level during the event. The freshwater impact also triggered a slight temperature decrease. On closer inspection, lander 1, which was shallower and enclosed, showed a greater decrease in

salinity than lander 2. Chlorophyll *a* concentration and turbidity showed increased values, while GHGs decreased during this event. The $pCH_4$ showed a strong positive correlation with the salinity ($r_s = 0.77$).

**Water outflow – (III)**

In late July, we observed an outflow of water from the peatland that caused the water column relative to the CTD-O$_2$ sensor to reach ~ 0 m, draining large areas of the rewetted peatland. This drop in water level resulted in an estimated reduction in water

volume by a factor of 5–6 compared to normal water level conditions. Nevertheless, the sensors were covered with water throughout the period, allowing uninterrupted data coverage. The outflow was slow and extended over several days with a subsequent rapid water inflow. Nighttime temperatures were coldest one day before the minimum water level was reached. Water level lowering triggered a decrease in $pCO_2$ and $pCH_4$, and a negative correlation with wind speed ($r_s = -0.57$ and $r_s = -0.64$, respectively), as well as a suppression of the amplitude of the diurnal cycles of both GHGs.

**3.4  GHG fluxes derived from high-resolution data at lander 1 and lander 2**

GHG fluxes for $CO_2$ and $CH_4$ were derived from the entire high-resolution sensor data using a gas transfer velocity model parameterized by wind speed (see Section 2.5). In addition, we chose to calculate GHG fluxes using the same approach but based on different scenarios: from bottle data only, during daytime, during nighttime, and using data of a previous study to isolate GHG fluxes for a direct year-to-year comparison. These results are summarized in Table 2.

Since the $pCO_2$ values were predominantly above atmospheric equilibrium during the deployment, a mean ($\pm$ SD) $CO_2$ flux of $0.15 \pm 0.18$ g m$^{-2}$ h$^{-1}$ and $0.09 \pm 0.13$ g m$^{-2}$ h$^{-1}$ was determined for lander 1 and lander 2, respectively, with fluxes being persistently higher at lander 1 (Table 2). Nevertheless, a short $CO_2$ uptake occurred at both landers in the beginning of the deployment in early June (Figure E 2). The range of measured $pCH_4$ values indicated a permanent supersaturation compared to the atmospheric equilibrium. Consequently, the resulting air-sea flux was $0.48 \pm 0.51$ mg m$^{-2}$ h$^{-1}$ and

$0.54 \pm 0.61$ mg m$^{-2}$ h$^{-1}$ for lander 1 and lander 2, respectively (Table 2). The differences between GHG fluxes ($CO_2$ and $CH_4$) derived from daytime and nighttime are large, which is related to the diurnal cycles of the gas partial pressures and the wind



speed (Figure 4). Hence, atmospheric GHG fluxes are 2.1–2.3-fold higher for $p$CO$_2$ and 2.3–3.0-fold higher for $p$CH$_4$ during the day than at night, depending on the position of the lander (Table 2).

**Table 2:** Greenhouse gas fluxes were calculated for both lander positions as well as by using different data basis. The fluxes were calculated based on the sensor data[1] (bold), and for comparison, the GHG fluxes were additionally calculated using the bottle data[2]. In addition, GHG fluxes were calculated based on the sensor data only for the daytime[3] and nighttime[3] to show the impact of diurnal effects. To obtain robust values, the sensor data for daytime and nighttime were each averaged by ± 1 hour. Moreover, we used the published data from (Pönisch and Breznikar et al., 2023) to calculate the ASE for the corresponding period in 2020[4], where investigations were conducted at the same study site during the first summer after rewetting (column "data from 2020"). In order to achieve the most comparable situation, we calculated the 435 ASE based on the sensor data for the period between 09:00 and 15:00 UTC, since this is the period when the main sampling was performed in the mentioned reference[5]. The calculation procedure of (1) – (5) is described in Sect. 2.6.

| | | sensor data[1] | bottle data[2] | daytime[3] 12:00 UTC | nighttime[3] 00:00 UTC | data from 2020[4]* 09:00 – 15:00 UTC | sensor data[5] 09:00 – 15:00 UTC (from 2021) |
|---|---|---|---|---|---|---|---|
| Lander 1 – CO$_2$ (g m$^{-2}$ h$^{-1}$) | mean ± SD | **0.15 ± 0.18** | 0.16 ± 0.08 | 0.22 ± 0.21 | 0.10 ± 0.14 | | |
| | 0.98th pct – 0.02th pct | **~0.00 – 0.67** | N/A | N/A | N/A | | |
| Lander 2 – CO$_2$ (g m$^{-2}$ h$^{-1}$) | mean ± SD | **0.09 ± 0.13** | 0.10 ± 0.10 | 0.12 ± 0.15 | 0.06 ± 0.08 | N/A | N/A |
| | 0.98th pct – 0.02th pct | **– 0.02 – 0.46** | N/A | N/A | N/A | | |
| Both lander – CO$_2$ (g m$^{-2}$ h$^{-1}$) | mean ± SD | **0.12 ± 0.16** | 0.13 ± 0.09 | 0.17 ± 0.07 | 0.08 ± 0.03 | 0.32 ± 0.22 | 0.17 ± 0.20 |
| Lander 1 – CH$_4$ (mg m$^{-2}$ h$^{-1}$) | mean ± SD | **0.48 ± 0.51** | 0.56 ± 0.22 | 0.82 ± 0.68 | 0.28 ± 0.32 | | |
| | 0.98th pct – 0.02th pct | **0.01 – 2.14** | NA | N/A | N/A | | |
| Lander 2 – CH$_4$ (mg m$^{-2}$ h$^{-1}$) | mean ± SD | **0.54 ± 0.61** | 0.65 ± 0.40 | 0.79 ± 0.76 | 0.35 ± 0.41 | N/A | N/A |
| | 0.98th pct – 0.02th pct | **0.01 – 2.41** | N/A | N/A | N/A | | |
| Both lander – CH$_4$ (mg m$^{-2}$ h$^{-1}$) | mean ± SD | **0.51 ± 0.56** | 0.61 ± 0.32 | 0.80 ± 0.02 | 0.31 ± 0.05 | 1.96 ± 6.59 | 0.76 ± 0.68 |

SD stands for standard deviation; pct stands for percentile; N/A stands for not available; data marked with the asterisk * were retrieved from Pönisch and Breznikar et al., 2023

## 4. Discussion

### 4.1 Biogeochemistry and driving parameters at the two observation sites.

The deployment of two landers equipped with sensors for the high-resolution determination of marine variables in a coastal 440 peatland revealed large temporal and spatial variations of the measured variables. Temporal variability occurred on multi-day scales, diurnal scales, and through event-based changes, with spatial differences occurring for variables dominantly controlled by biological processes (e.g., GHGs). To our knowledge, there is no study covering a comparable study area with similar temporal data resolution. The range of variability was beyond time series for coastal areas which also exhibit diurnal cycles (e.g., Honkanen et al., 2021).

The time series of the physical parameters (i.e., temperature, salinity, water level) showed strongly fluctuating conditions on multi-day and daily scales, but the spatial differences between the two landers were very small. The latter reflects a strong spatial coupling between the two lander positions. The coupling is related, on the one hand, to the short distance between the two landers of ~ 400 m and on the other hand, to the pronounced water exchange with the adjacent *Kubitzer Bodden* driven by frequent water level changes. Together with wind-driven mixing of the shallow water column, this resulted 450 in predominantly mixed water conditions and thus, O$_2$ consumption did not lead to long-term near-bottom anoxic conditions (Figure 2f). This hydrologic coupling implies that the peatland became part of the coastal water region as a result of flooding (i.e., incl. transport of SO$_4{}^{2-}$) and that biogeochemical patterns were influenced by processes occurring in the rewetted peatland as well as in the connected coastal shallow waters.

As a result of hydrological coupling, the time series of the GHGs at the two landers were also partially coupled, as 455 evidenced by similar responses in the form of multi-day variability and event-based changes (Section 4.3). The multi-day variations, especially the decrease in $p$CO$_2$ and $p$CH$_4$ appear to be coupled to increased wind speeds after phases with low





wind velocities, with a parallel decrease in temperature. Conversely, GHG concentrations increased again during periods of low winds, as also indicated by the slightly negative correlations of both GHGs with wind speed. With increasing wind, both ASE and water exchange with *Kubitzer Bodden* were enhanced. This facilitates a decrease of accumulated peatland GHG concentrations to atmospheric equilibrium - or coastal Baltic Sea (*Bodden*) background conditions. As a side effect, lower temperatures may occur due to deeper mixing in the adjacent *Kubitzer Bodden*. The multi-day accumulation is less visible for $O_2$ because of the faster re-equilibration with the atmosphere. Superimposed on the multi-day fluctuations are diurnal cycles, discussed in Sect 4.2.

In addition to the patterns described above, spatial differences were found between the two landers for $pCO_2$, $pCH_4$, and $O_2$ and thus for variables that are affected by processes of biology, water transport and air-sea exchange. The differences are mainly reflected in deviating values for mean and standard deviation (Table 1): While both landers exhibited comparable variability in $pCO_2$, lander 1 was characterized by a higher mean $pCO_2$. The higher values in the central peatland were associated with a high availability of OM for mineralization processes. The high availability could originate from various sources, e.g., from the flooded former vegetation and plant residuals that died after rewetting (Hahn-Schöfl et al., 2011), from the preserved or partially degraded peat layers or from the OM supply from new primary production. The availability and decay of OM has already been identified as a major contributor to elevated $CO_2$ concentrations in a recent study, which covered the same study area in the first year after flooding (Pönisch and Breznikar et al., 2023). Since our investigations took place only one year later, comparable conditions can be expected.

A similar impact of the higher availability of OM in the central region was expected for $pCH_4$. The higher availability of readily degradable OM is a major driver for $CH_4$ formation (Heyer and Berger, 2000; Hahn-Schöfl et al., 2011; Glatzel et al., 2008; Parish, 2008), and should have resulted in higher $CH_4$ concentrations at lander 1. However, lander 2 showed slightly higher and more variable $pCH_4$ values, consistent with lower $O_2$ values. This was likely due to (1) a slightly deeper position compared to lander 1 (~ 0.3 m deeper), which may lead to a faster expansion of $CH_4$-producing zones in the soil along with a stronger $O_2$ depletion during calm conditions, or to (2) locally high water velocities and transport processes. The latter seems to be more important, as the lander 2 was located in the narrow channel, where high water velocities occurred (Figure E 1). High water velocities can lead to local re-suspension, which in turn promotes pore water fluxes and can lead to the transport of soluble compounds into the water column (e.g., Massel, 2001; Beer et al., 2005), including $CH_4$ accumulated in the soil. The stronger control of the velocity is also visible since $pCH_4$ at lander 1 showed a more pronounced temperature control (Table 1). Temperature is another important driver of $CH_4$ concentrations, with higher $CH_4$ concentrations and fluxes occurring at higher temperatures (e.g., Bange et al., 1998; Heyer and Berger, 2000). The higher short-term fluctuations of the $CH_4$ signal (Figure 2b) at the dike opening (lander 2), the higher mean concentrations, and the correlation with water movement, suggests that the local position with high water transport rates and faster alteration of water supply from the *Kubitzer Bodden* and the drained peatland impacted the $CH_4$ signal at this location to some extent. Still, the – relatively weak – diurnal trend and major pattern during and following the discussed three major events (Section 4.3), are in unison.

## 4.2 Diurnal cycles and implication for discrete sampling

The solar irradiance causes diurnal cycles in the physics, chemistry, and biology of water bodies and has a particularly strong effect in shallow waters. As a result, diurnal cycles of $pCO_2$ and $pCH_4$ in shallow waters are coupled to a temperature-controlled cycle (i.e., solubility), air-sea gas exchange (i.e., wind parametrization), and biological processes influenced by eutrophic conditions and temperature (i.e., cycle of primary production and mineralization).

Binning the data set of the entire deployment by hours of daytime (Section 3.2) revealed that the highest GHG partial pressures ($CO_2$, $CH_4$) were observed in the morning and the lowest in the afternoon, with a more pronounced cyclicity for $pCO_2$ (Figure 4). Temperature, and $O_2$-saturation were also characterized by diurnal cycles, phase shifted almost exactly by 12



hours. In addition, a clear diurnal pattern is also visible for wind speed (Figure 4e), enhanced during daytime and with a peak in the hours around noon, indicating a classical sea-breeze situation in summer.

Based on the diurnal water temperature cyclicity, one would expect lower GHG partial pressures to occur at lower temperatures and vice versa, due to the temperature-dependent solubility. Since this was not the case for $pCO_2$, the reduction in $pCO_2$ during daytime caused by primary production and increase during nighttime caused by mineralization, clearly exceeded the temperature effect on solubility: During the day, the biologically controlled $pCO_2$ minimum exceeded the less influential temperature-controlled diurnal $pCO_2$ maximum. At night the opposite occurred, with mineralization being more

important than cooling. The dominance of production (and mineralization) is supported by the strong negative correlation with $O_2$ (Figure 3). The mean oxygen saturation of 75 % and 53 %, respectively, is in line with the high mean $pCO_2$ values, showing a general stronger contribution of mineralization than primary production. This relationship, representing a stronger biological control in comparison to the physical temperature-driven solubility effect, was recently described for $pCO_2$ for a shallow area on the Baltic coast near Utö Island in August, but with a much lower amplitude of 30 μatm (Honkanen et al., 2021).

The cyclicity of $pCH_4$ was lower and could be related to the higher $O_2$ availability and higher wind speeds during the day. In particular, the influence of wind speed - higher wind speeds favor the loss of $CH_4$ towards the atmosphere through air-sea exchange (section 2.5) - likely contributed to the cyclicity in this shallow setting. In shallow waters, the ratio between water volume and water surface is low, and wind-driven reduction of concentrations towards atmospheric equilibrium is fast. Therefore, the higher wind speeds during the day likely contributed to the loss of $CH_4$ during daytime.

The diurnal cyclicity of GHGs, temperature, and $O_2$ in peatlands and / or coastal waters, along with possible additional spatial variability, must be considered when establishing sampling strategies. Discrete sampling campaigns, typically conducted during the day, appear questionable, as according to our data, they have a considerable bias. The effect of this bias on the calculation of air-sea fluxes is discussed in section 4.4. A suitable approach to capture these short-term temporal variations is the use of high-resolution, autonomous measurement techniques such as sensor-based or eddy covariance

measurements. The potential influence of diurnal cycles in the flooded peatland during the winter months has not yet been investigated but is likely lower due to smaller daily changes in temperature. Also, as mentioned above, at least for our study site the summer had been identified as the season with strongest $CH_4$ and $CO_2$ dynamics based on discrete sampling in the year before (Pönisch and Breznikar et al., 2023).

### 4.3 Event driven biogeochemistry in the semi-enclosed peatland

Three system changes occurred during the deployment, which, in addition to the multi-day and diurnal variability, represent another system-describing feature. Like diurnal cycles, these events can only be tracked by high-resolution measurements because changes occurred fast (on the order of hours and days). Although the flooded peatland is semi-enclosed with only a 20 m-wide channel, water mass exchange occurs and the *Kubitzer Bodden* acts as a start and endmember for external signals.

The first event (I) was characterized by elevated wind speeds leading to a short runoff, followed by a substantial

inflow of water, a fast temperature drop of ~ 13 K, a peak in turbidity, and lower GHG concentrations (Figure 2). The inflow transported water with lower GHG concentrations from the *Kubitzer Bodden* and likely led to a dilution effect. In addition, strong cooling occurred during the event, with the strong temperature drop apparently caused by the incoming water from the *Kubitzer Bodden*, which got mixed by the strong winds and entrained colder deeper waters. The strong decrease in temperature likely led to reduced microbial activity, as this relationship has been described especially in coastal regions (e.g., Bange et al.,

1998; Heyer and Berger, 2000). A reduced microbial activity is consistent with the drastically suppressed amplitude of the diurnal cycle of $pCO_2$ and $pCH_4$. During the event, turbidity increased at both lander positions, indicating increased re-suspension. Although re-suspension is known to potentially trigger the exchange of soluble compounds and GHGs between the sediment and the water column (e.g., Massel, 2001; Beer et al., 2005), this is not reflected in our data (i.e., comparably low $pCO_2$ and $pCH_4$ during the turbidity maximum). The partial pressures of the GHGs and the strength of the diurnal cycle



recovered over the course of a few days and simultaneously returned to pre-event temperatures, consistent with the temperature and dilution that mainly caused the observations.

The second event (II) showed an impact of freshwater by precipitation and resulted in a fast drop of salinity within one day and a potential outflow of water due to a positive water balance. In addition, salinity remained low after this event and increased only slowly. Both GHG partial pressures showed a decline and a lower diurnal amplitude. It is assumed that a dilution

effect occurred due to the precipitation, which was also described for event (I), as well as a transport of dissolved compounds towards *Kubitzer Bodden* due to the positive water balance. With strong precipitation, a transport of dissolved compounds, such as nutrients or organic material, from the terrestrial catchment into the peatland water is likely and could have influenced the GHG dynamics afterwards.

The third event (III) comprised a sustained outflow resulting in a water column of ~ 0 m height at lander 1, followed

by an inflow of water up to normal conditions. These very low water levels prevailed for around 24 h. As a consequence of this event, water temperature, $pCO_2$, $pCH_4$, and $O_2$ decreased with an additional decline in diurnal amplitude. With the outflow and lower water volume together with ongoing GHG emissions, this likely led to a decrease in GHG concentrations in the water. In addition, larger areas of the shallow peatland fell dry, so $O_2$ may have infiltrated into the soil. $O_2$-penetration and lower temperature could have led to lower $CH_4$ production after the water level rose, resulting in low $CH_4$ concentrations in

the water column for several days after the event.

All three events show a strong correlation (in fact the strongest Spearman correlation coefficients between all parameters) between $pCO_2$ and $pCH_4$. (Figure E 4). Also, correlations with the physical drivers, in particular temperature, salinity, and wind speed, are far more pronounced then for the entire data set (Figure 3, Figure E 4). The stronger correlation with temperature during these shorter episodes, as the parameter most directly related to the diurnal cyclicity, is dominant in

particular for events I and III. It is noteworthy that this correlation, though clearly inferred from the visualization of the diurnal cycle (Figure 4), is less pronounced for the entire data series in the Spearman correlation analysis (Figure 3), indicating that the episodic changes and long-term trends partially hide this dependence on longer time scales. In summary, during short-term episodes of strong hydrographical forcing, both trace gases are strongly controlled by these physical drivers, leading to a very strong correlation.

### 4.4 Derived GHG fluxes from the flooded peatland

Greenhouse gas fluxes for $CO_2$ and $CH_4$ could be derived from the high-resolution sensor data as measurements were made narrowly below the water surface (< 1.25 m) and a direct coupling of water at the lander with the surface water was assumed. Although the peatland showed a slight $CO_2$ uptake in early June (Figure E 2), accompanied by stable but slightly decreasing chlorophyll *a* concentrations, the ASE was clearly dominated by a flux of $CO_2$ to the atmosphere. This amounted to

$0.12 \pm 0.16$ g m$^{-2}$ h$^{-1}$ derived from both landers. Increasing fluxes through early July stabilized with a simultaneous strong increase in chlorophyll *a* concentration. Overall, $CO_2$ emissions in the peatland were controlled by the simultaneous occurrence of primary production and mineralization, with the latter predominating for an overall net $CO_2$ outgassing. Comparison of our peatland fluxes with other studies revealed emissions that were around one order of magnitude higher. For example, 9 years after flooding with freshwater, $CO_2$ emissions from a shallow lake formed on a formerly drained fen varied from 0.02 g m$^{-2}$ h$^{-1}$

(open water) to 0.09 g m$^{-2}$ h$^{-1}$ (emergent vegetation stands; numbers adapted from Franz et al., (2016)).

In turn, the derived $CH_4$ fluxes of $0.51 \pm 0.56$ mg m$^{-2}$ h$^{-1}$ are significantly lower than those reported for temperate fens rewetted with freshwater with comparable boundary conditions (Couwenberg et al., 2011; Hahn et al., 2015; Franz et al., 2016). For example, $CH_4$ emissions from the shallow lake mentioned above varied from 1.48 mg m$^{-2}$ h$^{-1}$ (emergent vegetation stands) to 6.05 mg m$^{-2}$ h$^{-1}$ (open water) even 9 years after rewetting (numbers adapted from Franz et al., (2016)). In another

study, where a dry fen was converted to a shallow lake with occasional brackish water impact, a $CH_4$ flux of 29.68 mg m$^{-2}$ h$^{-1}$ was reported in the first year after rewetting (Hahn et al., 2015).





### 4.4.1 GHG flux development of the peatland in the second year after rewetting

The above mentioned studies report annual fluxes and since our study covers only the summer months, comparability is limited. As annual $CO_2$ and $CH_4$ emissions are normally highest in the late summer months, our emissions likely represent the higher
level of emissions over the course of a year. To allow a more direct comparison of the development of the GHG emissions at our study site, we used the published data of Pönisch and Breznikar et al., 2023. These data covered the entire year 2020 from the same peatland area, one year prior to the study presented here. The system was described as nitrogen-limited, with availability of $PO_4^{3-}$ in the summer and this is in line with the conditions observed in our study, where discrete samplings for $NO_3^-$ were predominantly below the detection limit (data not shown) while $PO_4^{3-}$ was available (Figure 2). From the data of
Pönisch and Breznikar et al., 2023, we isolated the same period during which the high-resolution measurements were made with the landers to assess the evolution of the ASE over one year (i.e., comparing summer 2020 and summer 2021). From Pönisch and Breznikar et al., 2023 we used ~ 190 measurements for summer 2020, with ~ 35 measurements from the peatland and the remaining from a transect in the vicinity of the later lander 1. We adjusted our high-resolution data to the conditions of their study by using a daily average value of data between 09:00 and 15:00 UTC (Table 2), as discrete sampling was
conducted within this time window in the campaign of the year 2020 (Pönisch and Breznikar et al., 2023).

Besides a high degree of comparability, there are clear limitations to this approach. The most important ones are. Slightly different sampling height, which was in Pönisch and Breznikar et al., 2023 ~ 20 cm below the water surface and ~ 60–90 cm in our study; different sampling approach; and possible inter-annual variations, which cannot be addressed with only two years. Therefore, the comparison only allows the indication of a trend that must be followed with further studies.

The comparison suggests that the $CO_2$ and $CH_4$ fluxes in the second summer after inundation were lower by a factor of 1.9 and 2.6, respectively compared to the first (Table 2). Although, as mentioned above, interannual variability rather than a trend cannot be excluded and the different sampling approaches add uncertainty for comparing the fluxes in the two years, the reduction in $CO_2$ and $CH_4$ emissions is perfectly in line with the hypothesis of decreasing GHG fluxes in the years after inundation. The decay of vegetation from the former drained peatland and the decomposition of the organic-rich top soil, foster
strong mineralization of OM and fuel $CO_2$ and $CH_4$ production (Heyer and Berger, 2000; Hahn-Schöfl et al., 2011). This decomposition of OM was to be expected and represents a transitional phase. With decreasing availability of degradable OM, stabilization of the microbial communities and potentially the establishment of new vegetation characteristic for brackish wetlands, a further reduction in GHG emission is expected.

### 4.4.2 The impact of diurnal GHG variability on GHG flux estimates

The studied peatland revealed a high temporal variability with respect to GHG partial pressures and other variables, characterized by multi-day and diurnal variations, with consequences of ASE estimates. To evaluate the impact, we calculated GHG fluxes from continuous time series when theoretically only a) nighttime, b) daytime was sampled to show the day-night-bias, and ASE only from c) bottle data (Table 2). The latter was done because there was a high discrepancy between sensor data and discrete samples in some cases (Table C 1).

Comparison with the ASE derived from entire high-resolution data showed higher fluxes from bottle data, highest fluxes during daytime, and lowest fluxes during nighttime (Table 2). The factors of $CO_2$ and $CH_4$ emissions between daytime and nighttime was 2.1–2.3, and 2.3–3.0, respectively. In our case, bias was not only caused by the diurnal fluctuation of the GHG concentration data, but superimposed by the diurnal modulation of wind speed (Figure 4) Consequently, studies on organic-rich flooded peatlands and shallow coastal areas, which are based only on bottle data and typically conducted during
daytime may result in biased GHG flux estimations for $CO_2$ and $CH_4$ due to a strong impact of diurnal cyclicity. It is therefore recommended to adjust the sampling and calculation interval of the biogeochemical parameters as well as relevant parameters for flux estimation (e.g. wind speed) to resolve these internal frequencies.





### 4.5 Assessment of the data quality and implications for future lander deployments

The direct comparison of the GHG partial pressures derived from the bottle data with the sensor measurements (calculated

mean of ~ 10 min prior to discrete sampling) showed good agreement in some cases, but also large discrepancies, with differences ranging from − 2.2 to 445.3 % for $p\mathrm{CO_2}$, and with higher discrepancies for the $p\mathrm{CH_4}$ determinations, ranging from 11 to 600 % (Table C 1). This observation demonstrates the complexity of combining both approaches in such a challenging and heterogeneous environment. Reconciliation of discrete bottle data with continuous sensor data is complicated by a variety of potential influences due to variable field conditions, a difficult sampling environment, and long sensor response times

(Canning et al., 2021). For example, discrete sampling of water in such shallow water may lead to minor sediment disturbance (i.e., boat mooring, Niskin bottle movements). In addition, bottom water is very heterogeneous for microbial mediated processes in both lateral and vertical directions, and sampling directly at the pump basket or sensor inlet could not be performed satisfactorily. The effect of water heterogeneity was also noticeable in the duplicates of the bottle $\mathrm{CH_4}$ data, with some varying by > 25 % (corresponding for several 100 nmol $L^{-1}$), even though the duplicates were from the same batch of Niskin bottle

(data not shown). In future deployments, discrete sampling at the sensor inlet by an automated sampler should lead to a more reliable comparison. Still, the different nature of the data, including the smoothing effect of the sensor data even with a state-of-the-art time lapse correction, makes comparison difficult. The similarity of the major trends in the data series on both landers is a strong indicator for the quality of the data, and under unlimited ressources, double sensor setups would always be very favorable.

640        The absolute numbers of a fraction of the data for $p\mathrm{CO_2}$ and $p\mathrm{CH_4}$ from the sensors have a small uncertainty because the selected calibration was exceeded partly ($p\mathrm{CO_2}$), and the pumps did not run reliably over the entire period, resulting in a lower flux onto the membrane. A similar situation occurred with the CTD-$\mathrm{O_2}$ measurement, resulting in parts of the data having to be discarded. These disturbances were related to the shallow water depths and high sediment transport and resulted in a lower data quality. Nevertheless, a robust post-processing was achieved, and hence presenting the data is valid because the

natural heterogeneity is much more pronounced and they address the current scientific and policy issue of peatland rewetting. Moreover, the data showed a detailed view in biogeochemistry processes in a rewetted peatland covered major features in coastal research.

        Improvements in future deployments can be achieved by adjusting the pumps. For example, all sensors should be equipped with a central inlet or revised pumps that are more resistant to environmental influences. In addition, more attention

should be paid to controlling the water flow at the sensor units, such as by monitoring the flow or using self-cleaning pump systems. The use of additional non-pumped loggers, e.g., for CTD-$\mathrm{O_2}$ determinations, can supplement and validate sensor data and the fouling protection could be achieved by using UV light lamps. Furthermore, we plan to update and improve the real-time data transmission to respond quickly to lander systems irregularities.

### 5.  Conclusion

The measurement of key marine physicochemical variables in a shallow brackish water column of a rewetted coastal peatland using sensor-equipped landers was, to the best of our knowledge, demonstrated for the first time. The results showed that the rewetted peatland was characterized by a generally high system variability. In particular, GHG partial pressures were dominated by long-term (multi-day) variability and diurnal cycles. The methodological approach used, combining sensor measurements with discrete, validating samples, appears appropriate because variability over a short period of time can be

resolved. The short-term variability ranged in scales of hours and was pronounced for $p\mathrm{CO_2}$ and $\mathrm{O_2}$, as variables that are predominantly biologically influenced, but also the temperature and $p\mathrm{CH_4}$ were affected by fast changes as well as diurnal cyclicity. Derived from this observation, conditions can be quite different between daytime and nighttime. Hence, to achieve



a comprehensive biogeochemical interpretation, and in particular for the derivation of GHG fluxes, this diurnal effect must be considered in shallow coastal waters and rewetted peatlands with a permanent water column.

665        The rewetting of formerly drained peatlands as well as the extension of shallow coastal ecosystems are currently discussed as potential measures for GHG emission reduction or even the creation of net negative emissions (CDR in Blue Carbon schemes). Our study strongly suggests that the monitoring, reporting and verification (MRV) schemes for these measures might not only require integrated assessment of carbon burial and GHG emissions, but that the short-term variability and cyclicity require new observational approaches to resolve the appropriate time scales.

670        With the deployment of the sensor-equipped landers, new challenges arose, such as the extensive technical infrastructure required or the difficult conditions of the water-covered peatland. The latter has affected the reliability of the state-of-the-art sensors and led to lower data quality. However, this circumstance is less significant when considering the given amplitudes of the fluctuations of the system. With improvements, e.g., new pump generations, sensor measurements can become more reliable in the future. Overall, the lander-based autonomous monitoring technique used in this study was 675 successful in capturing GHG variability in a shallow water approach.

       The evaluation of GHG fluxes showed that the almost permanently oversaturated conditions for $p$CO$_2$ and $p$CH$_4$ mean that the peatland was a source of atmospheric GHGs. Comparing the fluxes to a study conducted in the same area one year prior to our study, emissions decreased by a factor of 1.9 and 2.6, respectively. Although limited data comparability and potential interannual variability have to be acknowledged, the continued decline in GHG emissions, especially in the relatively 680 low CH$_4$ emissions, is in line with the hypothesis of a positive evolution of the peatland into a potential C sink (or small CO$_2$-source) with comparably low CH$_4$ emissions under brackish water influence.



**Appendix A: The landers**

**A1  The landers concept**

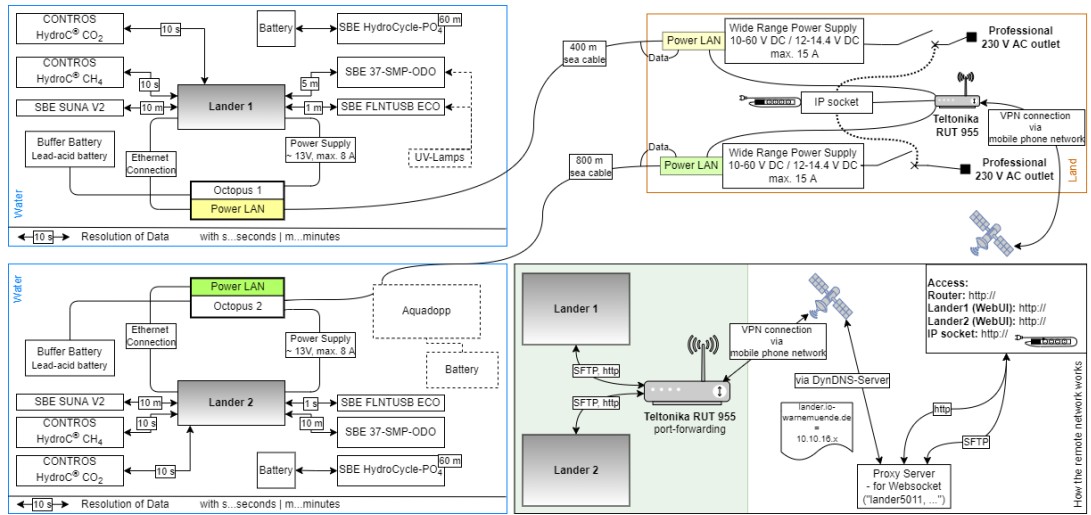

**Figure A 1:** The detailed overview of the measuring system, including the landers (left side), the power supply on land site (right top) and the remote access configuration (right bottom). Left: A dashed line represents equipment, that was used only for the actual lander. The captions at the arrows describes the resolution of measurements. Both landers were connected with a wired cable connection to ensure (right top) power supply and powerline communication. The hardware for this approach was bundled in the in-house developed "Octopus" device. The station on land enables remote power cycling and provides hardware for power supply and VPN connection. Right bottom: A schematic
flowchart of the used VPN infrastructure, which enables control and adjustments to lander data and sensors schedules.

**A2  Maintenance and biofouling**

During the deployment, measures were taken to ensure consistent data collection, as the study site is known to be challenging for sensor measurements, e.g., due to biofouling and re-suspension of particulate material. Hence, eight maintenance and visual inspections were conducted to monitor and remove effects of biofouling (Figure A 2). Furthermore, pump systems and optical
measurement ways were cleaned. To suppress biofouling, we installed two UV light antifouling devices developed at IOW on the CTD-$O_2$ and SBE-FLNTUSB-ECO instruments of lander 1. The UV light antifouling solution uses UV LEDs with a wavelength of 275 nm and customizable optics. The UV lamps were switched on for 25 s every 10 min. The development is patent-pending under DE 10 2019 101 420.4 and was funded by *Bundesamt für Seeschifffahrt und Hydrographie* (BSH). The technology is licensed to Mariscope Meerestechnik e.K. (Kiel, Germany).



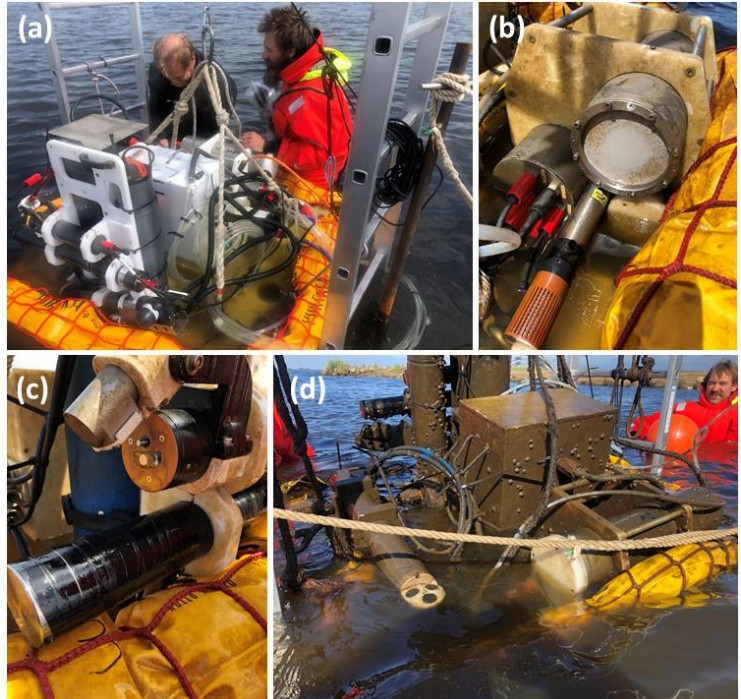


**Figure A 2:** (a) Setting up the lander in the central area of the peatland using a lift construction, (b) image of the HC-CH$_4$ sensor during maintenance after ~ 14 days of deployment, (c) setup and result of the UV antifouling device on the SBE-FLNTUSB-ECO instruments, and (d) the lander before recover on the last day of the deployment.

## Appendix B: Discrete samples analysis

### B1  Analysis of pH

The pH was determined spectrophotometrically using the pH-sensitive indicator dye metacresol purple (mCP; 2 mmol L$^{-1}$; -4H- JENA Engineering GmbH, Jena, Germany) and the measurement principle and instrumental setup are described elsewhere (Dickson et al., 2007; Carter et al., 2013). The absorbance measurements were performed using the UV-visible spectroscopy system Agilent 8453 (Agilent Technology, Waldbronn, Germany). The pH parameterization for brackish waters was calculated

according Müller and Rehder, (2018). A buffer solution with a salinity of 20, prepared according to Müller et al., (2018), and an external buffer solution with a salinity of 35 (University of California, San Diego (USA), Scripps Institution of Oceanography) was used for quality control.

### B2  Total CO$_2$ (C$_T$) analysis

C$_T$ was determined from discrete samples using the automated infrared inorganic carbon analyzer (AIRICA, MARIANDA,

Kiel, Germany). The analysis based on acidification of the sample (phosphoric acid, 10 %) followed by stripping of CO$_2$. The nitrogen carrier gas stream transported the released CO$_2$ to an infrared detector LI-7000 (LI-COR Environmental GmbH, Bad Homburg, Germany), via a Peltier cooler and a Nafion® drying tube to remove water residues. Triplicate measurements and comparison to certified reference material (CRM; University of California, San Diego (USA), Scripps Institution of Oceanography; Dickson et al., 2003) allow the C$_T$ to be calculated with a precision of $\pm$ 5 µmol kg$^{-1}$.



**B3  Total alkalinity ($A_T$) analysis**

$A_T$ was analyzed by open cell titration (glass electrode type LL Electrode plus 6.0262.100, Metrohm AG, Filderstadt, Germany) as described by Dickson et al., (2007) and involved a two-stage titration. After the first addition of hydrochloric acid to reach a pH of 4–3.5, $A_T$ was determined during a stepwise titration to a pH of 3 while the pH was recorded potentiometrically. The calibration was carried out with the same CRM as used for $C_T$ and allowed the same precision.

**B4  $CH_4$ analysis**

Dissolved $CH_4$ from bottle samples was measured using the GC Agilent 7890B gas chromatograph (Agilent Technologies, Santa Clara, USA) based on the purge-and-trap technique coupled with a flame ionization detector (FID) and detailed described elsewhere (Sabbaghzadeh et al., 2021; Pönisch and Breznikar et al., 2023). A purified helium gas stream purges $CH_4$ and other volatile compounds out from the seawater sample and is subsequently dried by passing a Nafion® tube (Perma Pure Nafion®,

Ansyco GmbH, Karlsruhe, Germany) and a SICAPENT® tube (Merck KGaA, Darmstdt, Germany). $CH_4$ is accumulated by cryofocusing ($-$ 125 °C) on a trap filled with HayeSep D® (CS Chromatographie Service GmbH, Langerwehe, Germany) using an ethanol/nitrogen cooling bath. After ten minutes, the desorption occurs by heating the trap (water, 95 °C) and the separation of the target molecule from other compounds was achieved by two capillary columns and a Deans Switch (Pönisch, 2018). A $CH_4$ containing standard (9.9379 $\pm$ 0.0159 ppm) was measured daily before and after the sample measurement for quality

control with a standard deviation of less than 1 %.

**Appendix C: Sensors and data processing**

**C1  Sensor operation mode of HC-$CO_2$ sensors and data post-processing**

The frequency of $p$CO$_2$ measurements was adjusted to 10 s, with one reading consisting of an average of ten readings calculated by the sensor software. The HC-$CO_2$ sensors were operated with 1 W pumps (SBE-5M; Sea-Bird Electronics Inc., Bellevue,

USA) and factory-calibrated for a measuring range of 200–6000 µatm before and after the deployment. The instruments regularly recorded zeroing values for 120 s by removing $CO_2$ from the gas stream every twelve hours. The zeroing was followed by a flush period of 780 s and the signal readjusted to environment conditions. Zero and flush intervals were removed before biogeochemical analysis.

Post-processing of $p$CO$_2$, as described in Fietzek et al., 2014, was applied to the sensor data by including regular

zeroing signals and a span drift correction using pre- and post-calibration polynomials. To correct the data for sensors response time ($\tau$), the actual in situ $\tau$ was determined in advance for each flush period. The signal recovery after zeroing intervals were filtered based on standard deviation with respect to the predecessor and was set to < 5 µatm in a series of three values. The analysis of the pump power consumption during the deployment showed an occasionally decrease reaching minimum values of ~ 0.5 W (data not shown), resulting in a lower flow rate at the membrane and likely caused by a clogging of the filter basket

and / or pump propeller due to sediment. As a result, the signal recovery to environmental conditions takes longer and the selection criterion (standard deviation < 5 µatm in a series of three) was not achieved in every zeroing interval, and hence some in situ determination of the response times had to be discarded. The filtered data passing the selection criteria were used to fit an exponential function to the signal recovery and followed the descriptions of Fiedler et al., (2013) and Bittig et al., (2014). A mean $\tau$ of 329 s and 380 s was obtained and used for data correction, while a rolling mean and median with a window

width of 25 measurements before and after $\tau$ correction was applied to the response time correction to remove short-term noise. Short-term interruptions in the continuous data and artifacts from post-processing were removed.

The NDIR units were calibrated in the range between 200–6000 µatm, and the measured values exceeded this range occasionally and reached maxima of ~ 10,000 µatm. Based on the calibration curve (not shown and provided by the manufacturer), the sensor followed a cubic signal-response, and extrapolation cannot be done with confidence. However, the





lower range (< 2000 µatm) is described to be more cubic, but is well described by the calibration. In a recent study, an NDIR unit was calibrated up to 25,000 µatm with only minor adjustments of the polynomial calibration curve (Canning et al., 2021). Hence, the accuracy of values exceeding 6000 µatm is slightly reduced, but we think that reporting these high $pCO_2$ values is of scientific importance. In support of this assumption, sensor data and bottle data were compared (Table C 1), and this showed that the relative differences in the upper range compared to the lower range are not unremarkable.

**C2  Sensor operation mode of HC-CH$_4$ sensors and data post-processing**

The frequency of $pCH_4$ measurements was set to 10 s, with one reading consisting of an average of ten readings calculated by the sensor software. The HC-CH$_4$ sensors were calibrated before and after deployment at 10 °C water temperature for a range of 1–40,000 µatm by the manufacturer, to prove the linearity and stability of the TDLAS unit throughout the deployment. Equipped with external 7.5 W pumps (SBE-5T; Sea-Bird Electronics Inc, Bellevue, USA), a response time of 2400 s was

determined for both sensors. The analysis of the pump power consumption during the deployment showed an occasionally decrease reaching minimum values of ~ 0.5 W (data not shown), resulting in a lower flow rate at the membrane and likely caused by a clogging of the filter basket and / or pump propeller due to sediment. Based on laboratory tests, pump consumption (W) can be converted to a water flow rate (L min$^{-1}$) and was simulated in the experiment with a SBE-5T pump between ~ 5 W (maximal pump capacity) and ~ 3 W (restricted pump capacity) due to artificial clogging of the filter basket. The decrease in

water flow rate from ~ 8 L min$^{-1}$ to ~3 L min$^{-1}$ follows a linear slope ($R^2 = 0.83$), and the lower limit of the water flow rate is equivalent with a SBE-5M under maximal pump capacity. Based on this observation and to correct the in situ $pCH_4$ values, the response times for the HC-CH$_4$ were determined using a SBE-5T and SBE-5M pump, respectively, resulting in an increase of the response time by a factor of 1.7. Hence, we chose to linear interpolate the response time between the two limits (limits between maximum pump power of ~ 6 W and 3 W), and extrapolate when pump power was below the threshold of 3 W. (N.

Kinski, -4H- JENA Engineering GmbH, personal communication, 2022). Since biofouling can be neglected (regular maintenance and cleaning), the response time was not additionally affected. The interpolated / extrapolated response time ranged between 2400 s and 4800 s and was used to correct the signal delay for each recorded $pCH_4$ value (same procedure like for HC-CO$_2$ sensor see above; described by Fiedler et al., (2013) and Bittig et al., (2014). A rolling mean with a window width of 180 before and a rolling median with a window width of 50 measurements after the calculation of the in situ $pCH_4$ value

was applied to remove short-term noise. Measurement interruptions and artifacts from post-processing lasting longer than 180 measurements were removed from the continuous data.

**C3  Quality assessment**

The application of the post-processing procedures described above (i.e., zerorings, span drift and τ correction) led to an improvement of the sensor data. However, the study site is in particular characterized by conditions that are challenging for

marine instrumentation and no sophisticated hardware adjustment (i.e., pump operated reliable in flat water) or post-processing are available for these environmental conditions. This resulted in higher uncertainties in absolute $pCO_2$ and $pCH_4$ values. But, since ~ 500.000 measurements are included for each compound, the time series is statistically robust. Nevertheless, some uncertainties will be mentioned in the following.

-The $pCO_2$ values reached > 10,000 µatm and therefore exceeded the calibration range, resulting in greater

uncertainties in the upper values.

-The determination of in situ τ for $pCO_2$ was challenging during periods of low pump activity (i.e., ~ 0.5 W power consumption) resulting in a higher spread for each actual τ determination and hence a slightly lower post-processing quality.

- The above-mentioned issue resulted in the lack of a sufficient amount of calculated in situ τ during periods of lower

pumping activity. Therefore, only an average τ could be calculated (i.e., 329 s and 380 s).



-The HC-CH$_4$ sensors occasionally showed gaps in the continuous data and rapid dynamics in the signal, which posed challenges for the post-processing techniques used. Consequently, a relatively wide window width for signal smoothing had to be applied (> 180 points), otherwise this would have resulted in many artifacts and values < 0 µatm. In turn, this signal smoothing resulted in a loss of micro-dynamics in the sensor signal, which can be quite pronounced compared to a narrower window widget. This is shown exemplary in Figure C 1.

**Figure C 1:** The effects of the different post-processing methods on the sensor data (red). Strong smoothing (blue) resulted in a lower level of signal detail compared to weak smoothing (gold), but the impact of different window widths is predominantly low.

**Table C 1:** Direct comparison of the GHG partial pressures derived from sensor data with bottle data. Sensor values were obtained by calculating an average of a 10-minute interval of the continuous data before discrete sampling. Values in bold show results for $p$CO$_2$ above the calibration range.

| | | $p$CO$_2$ | | | | |
|---|---|---|---|---|---|---|
| | | lander 1 | | | lander 2 | |
| date | sensor (µatm) | bottle data (µatm) | relative (%) | sensor (µatm) | bottle data (µatm) | relative (%) |
| 10-June-2021 | 1592.1 | 2120.7 | 33.2 | 232.1 | 184.7 | −20.4 |
| 15-June-2021 | 1767.2 | 2559.4 | 44.8 | 655.4 | 732.4 | 11.8 |
| 22-June-2021 | 2987.2 | 3925.1 | 31.4 | 727.1 | 531.9 | −26.8 |
| 29-June-2021 | **8832.3** | **8031.0** | **−9.1** | **6738.1** | **7875.5** | **16.9** |
| 06-July-2021 | **7042.1** | **6871.1** | **−2.4** | 5919.8 | 7140.0 | 20.6 |
| 15-July-2021 | **7355.5** | **7998.7** | **8.7** | **6220.1** | **7361.1** | **18.3** |
| 20-July-2021 | 3013.8 | 3389.7 | 12.5 | 2569.5 | 3307.1 | 28.7 |
| 27-July-2021 | **9145.9** | **8949.0** | **−2.2** | **7824.1** | **8735.6** | **11.7** |
| 03-August-2021 | 2474.8 | 3543.6 | 43.2 | 1752.6 | 1407.8 | −19.7 |
| | | $p$CH$_4$ | | | | |
| 10-June-2021 | 1775.7 | 1394.9 | −21.4 | 1408.1 | 1053.0 | −25.2 |
| 15-June-2021 | 451.5 | 742.4 | 64.4 | 371.0 | 509.4 | 37.3 |
| 22-June-2021 | 268.4 | 733.9 | 173.4 | 245.1 | 433.6 | 76.9 |
| 29-June-2021 | 336.34 | 1834.2 | 445.3 | 885.1 | 2048.7 | 131.5 |
| 06-July-2021 | 195.8 | 1041.9 | 432.1 | 911.7 | 1249.3 | 37.0 |
| 15-July-2021 | 637.4 | 1167.6 | 83.2 | 503.1 | 2171.5 | 331.6 |
| 20-July-2021 | 348.0 | 601.1 | 72.7 | 788.3 | 1151.8 | 46.1 |
| 27-July-2021 | 1334.2 | 2080.3 | 55.9 | 1214.7 | 2527.9 | 108.1 |
| 03-August-2021 | 150.3 | 605.9 | 303.1 | 125.8 | 874.2 | 594.9 |



### C4  Data post-processing of CTD-O₂ sensors

Pre-deployment calibration of temperature and salinity and post-calibration of the $O_2$ was carried out in the accredited
calibration laboratory of the IOW. Accordingly, $O_2$ was calibrated against a reference optode at zero $O_2$ concentration and
equilibrated $O_2$ concentration in a water bath and this 2-point calibration was used for a linear adjustment of the manufacturer
calibration. The assessment and correction of the oxygen optode drift during the deployment time (~ 9 weeks) was addressed
by in-air oxygen optode measurements before and after the sensor deployment. A slope correction between measured in-air
samples and the calculated atmospheric partial pressure was carried out each for pre- and post-deployment (Bittig and
Körtzinger, 2015). The average of each slope correction was calculated and hence, a coefficient for correction of 1.056 and
1.033 for CTD-O₂ 20514 and 20515, respectively, was used. In occasional periods of the time series, $O_2$ and conductivity had
to be removed from the data because the time series showed a not comprehensible steep decrease in values and a discrepancy
with discrete samples. This was likely related to the ingress of air bubbles into the measuring circuit. Although measures were
taken to avoid these air locks during maintenance, the low pump capacity of the CTD-O₂ was unable to maintain a continuous,
undisturbed air-free circuit. Temperature and pressure were unaffected due to their exposed position in the circuit.

### C5  Data post-processing of SUNA V2 sensors (NO₃⁻)

In the post processing, in situ temperature-corrected salinity absorbance (bromide absorption) was subtracted from the total
measured absorbance. The remaining temperature-corrected salinity-subtracted absorbance in the wavelength range 217–
240 nm was used in a regression analysis with $NO_3^-$ absorbance and a linear baseline as components. In addition, the technical
drift of light source intensity during the deployment was compensated by measuring the UV absorption spectra of MilliQ water
before and after the deployment. The post-processing revealed that the sensor spectra were dominated by CDOM interferences.
Therefore, it was assumed that the SUNA data are not suitable for deriving $NO_3^-$ concentrations, in the present environment
of very low $NO_3^-$ concentrations.

### Appendix D Data handling and calculations

The data processing, analysis and visualization was carried out in R (R Core Team, 2022) by using the packages tidyverse
(Wickham et al., 2019), patchwork (Pedersen, 2020), and DBI (Wickham and Müller, 2021). The data smoothing shown in
Figure 2 and Figure E 2 was performed with *geom_smooth()* using the method *gam* and a value k = 30 and 5, respectively.

Bottle data of $C_T$ and pH were used to calculate $pCO_2$ (e.g., Figure 2) by using the R package seacarb (Gattuso et al.,
2019) with $K_1$ and $K_2$ from Millero, (2010), $K_s$ from Dickson, (1990), and $K_f$ from Dickson and Riley, (1979).

CH₄ concentrations were calculated from sensor-based $pCH_4$ measurements by calculating a mean value for $pCH_4$,
salinity, and temperature for one-hour-periods. Since salinity is not consistently available at both landers, a combined salinity
time series was created by filling the values from lander 1 with those from lander 2. This appears to justified because the two
landers are connected by a strong water exchange, resulting in very similar conditions in physical variables, as shown in our
study as well as in a previous study (Pönisch and Breznikar et al., 2023).



845 **Appendix E : Results**

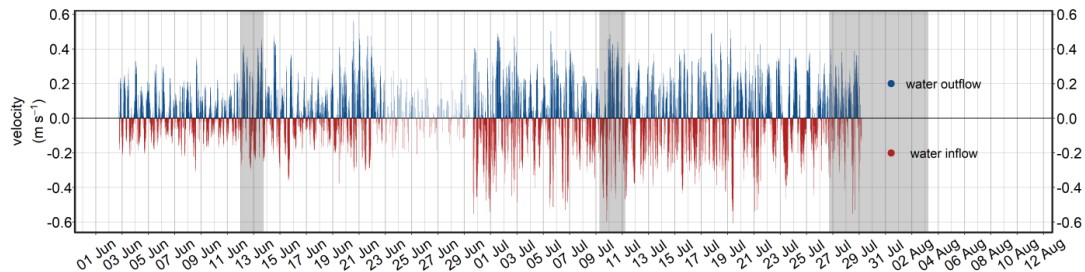

**Figure E 1:** a) High-resolution time series of the water velocity (m s$^{-1}$) measured at lander 2. Water outflow means that water flows from the peatland toward the *Kubitzer Bodden* driven by changes in the water level and vice versa.

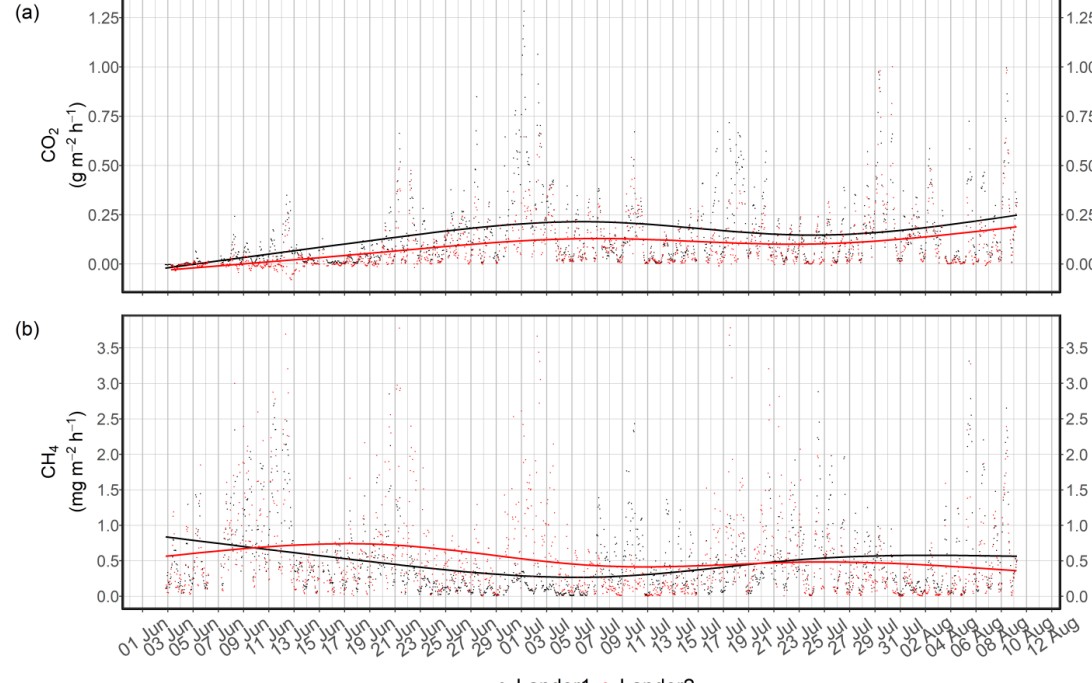

850 **Figure E 2:** Calculated GHG fluxes using the Wanninkhof, (2014) wind speed parameterization approach for CO$_2$ and CH$_4$ at both landers. A smoothing was used to highlight the trend and differences between the landers (description of smoothing in given in Appendix D).





**Figure E 3:** The distribution of the mean values of salinity, pressure, chlorophyll *a*, and turbidity from one-hour-parts over 24-hours. The box plots show the median and the 25 / 75 % quantiles. The whiskers indicate the 5 / 95 % percentiles, and the red points denote the mean values.

855



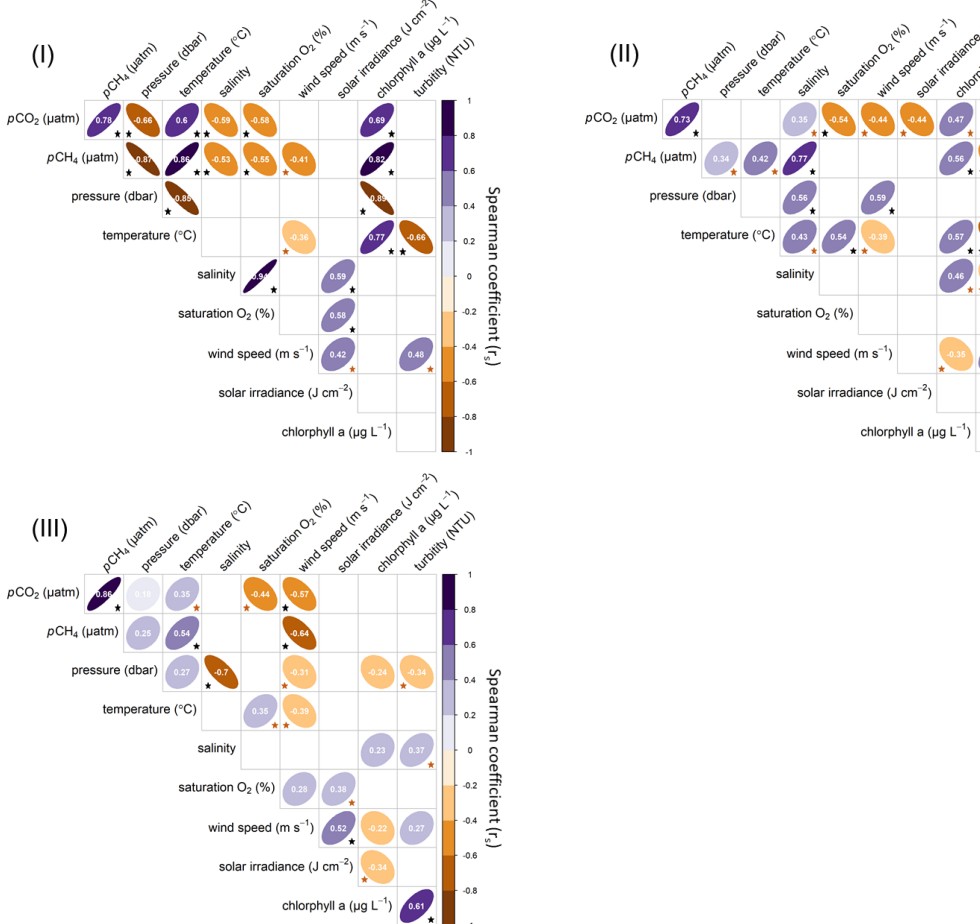

**Figure E 4:** Spearman correlation coefficients ($r_s$) between the measured variables of lander 1, wind speed, and solar irradiance limited to the three time periods (I) – (III). A correlation level of 0.001 was used to remove non-correlating relationships (empty fields). In addition, the Cohen convention (Cohen, 1988) was used to interpret the effect size. The black stars represent a large effect size and thus a strong correlation, while the brown stars represent a medium effect size. Wind speeds were retrieved from the station *Putbus* (WMO-ID 10093) and solar irradiance from *Rostock-Warnemünde* (WMO-ID 10170; both DWD).



*Data availability*. The raw and processed data used in this study are archived at the data system PANGAEA and are available
at https://doi.org/10.1594/PANGAEA.964839 (Pönisch et al., 2024). The water velocity measurements at lander 2 are archived
and available at https://doi.io-warnemuende.de/10.12754/data-2024-0013 (Pönisch and Holtermann, 2024).

*Author contribution*. DLP was the lead author of this study and was involved in all parts, including technical administration,
field work, data analysis, and writing. HCB assisted with data processing and interpretation of sensor data and contributed
significantly to the editing of the manuscript. IS and MK worked on the technical design and provided technical maintenance
during the study. SO was responsible for laboratory analysis of discrete samples. PH helped with the instrumentation of water
velocity measurements and conducted data post-processing. KP gave suggestions during the writing and helped with text
editing. GR was involved in the conceptual, project management, supervision, and paper writing. All authors contributed to
the internal review and editing of the manuscript.

*Competing interests*. The contact author has declared that none of the authors has any competing interests.

*Acknowledgments*. The successful deployment of the two landers was only possible through the cooperation of different
disciplines and the great commitment of many colleagues at IOW. The interplay of the several expertise led to the establishment
of an infrastructure necessary for the deployment, including the modification of the systems from battery operation mode to
wired operation mode, field sampling and maintenance, as well as the biogeochemical analysis in such a challenging
environment. In this context, the IOW working group "Marine Instrumentation" has made a significant contribution, in
particular Siegfried Krüger and Robert Wagner, who were involved in the adaptation and development of the landers. In
addition, Benny Baumann and Nadja Kinski (both from -4H- JENA Engineering GmbH) provided important support in the
technical implementation of the landers. During the field work, Benjamin Schönherr and Nils Trautmann helped with the setup
and maintenance of the landers as well as with the frequent sampling, which always involved a high logistical effort. Finally,
Birgit Sadkowiak provided important support during the laboratory work.

*Financial support*. This study was conducted within the framework of the Research Training Group "Baltic TRANSCOAST"
funded by the DFG (Deutsche Forschungsgemeinschaft; grant no. GRK 2000; https://www.baltic-transcoast.uni-rostock.de,
last access: 12 January 2023). This is Baltic TRANSCOAST publication no. GRK2000/xxxx. In addition, this research was
carried out as part of the research focus "**S**hallow water processes and **T**ransitions to the **B**altic Scale, (STB)" of the Leibniz
Institute for Baltic Sea Research Warnemünde (IOW).

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
