# Peer review of "Variability of CO2 and CH4 in a coastal peatland rewetted with brackish water from the Baltic Sea derived from autonomous highresolution measurements"

_EGUsphere, 2024_

## Author Comment (AC1)

Reviewer 1,

thank you very much for your detailed and constructive feedback provided within your major and smaller comments, which helped us to improve the manuscript.

The original manuscript and the study design focused on determining temporal variability and exploring biogeochemical interactions in summer, probably the most interesting season regarding GHG dynamics, and in a very complex and heterogeneous ecosystem. Many coastal peatlands (about 40,000 ha in MV) could be rewetted in this way, so there is a large potential and importance for rewetting coastal peatlands, partly with the option of rewetting with brackish waters. Due to the complex ecosystem and novel technology we used, we put strong emphasis on technical description, data analysis and interpretation of the underlying drivers and biogeochemical processes.

We also saw the issue that there is little literature on the rewetting of coastal peatlands by means of dyke removal. Furthermore, comparability to other studies is limited by very different environmental settings, so that the existing literature and data must be analyzed in a complex way, for example, by extracting only measurements from the summer months, to reach comparability to our measurements.

However, we understand that both reviewers saw the shortcoming of not putting our work in a broader perspective and suggested a better discussion of the findings in the framework of existing literature, and we are very thankful that the reviewers pointed this out.

We have taken many measures to reorganize and improve the text based on the major comments. We have analyzed the comments of both referees (RF1 and RF2) and identified major commonalities and strong similarities in the argumentation. We adressed these similarities between the referee comments by joint measures in the two responses to the referees (authors response (ACs)) which prompted us to formulate identical responses that address the following:

A)  Restructuring of the abstract to better address the scientific question of greenhouse gas dynamics and the underlying biogeochemical drivers
B)  Restructuring of the introduction to provide more context on the relevance of greenhouse gas reduction and to indicate that novel technology was used for this study
C)  We tried to improve the clarity of the scientific question by emphasizing the biogeochemical interpretation before the technical description at appropriate points. For this reason, we also moved the discussion on the match between sensor data and results from discrete water sampling (former chapter 4.5) to the appendix.
D)  We did our best to discuss our results on CO2 and CH4 fluxes in a broader scientific context by adding a new section 4.4.1. to the discussion.

In the following, we have reposted the comments by the reviewer (in bold) and placed our responses below them. Envisaged text changes/amendments are indicated by quotation marks.

1. **Major Comment(s)**

**Unclear balance between technical and scientific aspect of paper**

**The paper mostly reads as a technical paper, and describes these parts very well (technical aspects and comparison between methods). Nevertheless, the authors also claim that they want to assess a scientific question (last sentence of Introduction). This part however seems a little lit snowed under in the Discussion. A little bit more context on this in the Introduction as well as in the Discussion would be nice.**

**For example: can the authors place their results in a clearer/ broader perspective ? I am aware that there are limited studies on brackish ecosystems, but a comparison can be made between concentrations and fluxes of different (rewetted) peatlands, not only of brackish systems, and maybe a quick number of surroundings seas and rivers as well. Place the results into perspective: if moving from the terrestrial area (rivers and/or dry peatlands) to the sea, what is the gradient of fluxes, and how do your results fit in? Also, a table (maybe in Appendix) might be a nice way to present the different flux magnitudes of different (rewetted) peatland studies, which will help the reader placing the results in perspective. Also, are your results specific for this climate region? These climate rewetting activities are mostly active in Europe, but would it be possible to make a reference to a same situation in a tropical region?**

**Overall, I think that the interpretation of the scientific part needs to be elaborated. Nevertheless, the comments above serve only as ideas, feel free to find a different way to place your results in a broader perspective.**

Reply:

A) Abstract

Significant changes are proposed for the abstract to better reflect the clarifications and extensions envisaged for the revised manuscript.

- Shortening of a section which will be moved to the introduction (line 17-21)
- Inclusion of a sentence highlighting the novelty of the measurement technology (line 24)
- Inclusion of the Spearman correlation to better indicate the focus on the analysis of drivers and biogeochemical processes (line 26)

Taking account also minor changes requested by RF1 and RF2, we wil provide a new version of the abstract:

"**Abstract.** Rewetting peatlands is an important measure to reduce greenhouse gas (GHG) emissions from land use change. After rewetting, the areas can be highly heterogeneous in terms of GHG exchange and depend, for example, on water level, vegetation, temperature, previous use, and duration of rewetting. Here, we present a study of a coastal peatland that was rewetted by brackish water from the Baltic Sea and thus became part of the coastal shallow Baltic Sea water system through a permanent hydrological connection. Environmental heterogeneity and the brackish water column formation, require improved quantification techniques to assess local sinks and sources of atmospheric GHGs. We conducted nine weeks of autonomous and high-resolution, sensor-based bottom water measurements of marine physical and chemical variables at two locations in a permanently flooded peatland in summer 2021, the 2$^{nd}$ year after rewetting. For the study, we used newly developed multi-sensor platforms (landers) customized for this operation. Results show considerable temporal fluctuations of $CO_2$ and $CH_4$, expressed as multi-day, diurnal and event-based variability and spatial

differences for variables dominantly influenced by biological processes. Episodic and diurnal drivers are identified and discussed based on Spearman correlation analysis. The multi-day variability resulted in a pronounced magnitude of measured GHG partial pressures during the deployment ranging between 295.0–8937.8 µatm ($CO_2$) and 22.8–2681.3 µatm (correspond to 42.7–3568.6 nmol $L^{-1}$; $CH_4$), respectively. In addition, the variability of the GHGs, temperature, and oxygen was characterized by pronounced diurnal cycles, resulting e.g., in a mean daily variability of 4066.9 µatm for $CO_2$ and 1769.6 µatm for $CH_4$. Depending on the location, the diurnal variability led to pronounced differences between the measurements during the day and night, so that the $CO_2$ and $CH_4$ fluxes varied by a factor of 2.1–2.3 and 2.3–3.0, respectively, with higher fluxes occurring over daytime. The rewetted peatland was further impacted by fast system changes (events) such as storm, precipitation and major water level changes, which impacted biogeochemical cycling and GHG partial pressures. The derived average GHG exchange amounted to 0.12 ± 0.16 g $m^{-2}$ $h^{-1}$ ($CO_2$) and 0.51 ± 0.56 mg $m^{-2}$ $h^{-1}$ ($CH_4$), respectively. These fluxes are high ($CO_2$) to low ($CH_4$) compared to studies from temperate peatlands rewetted with freshwater. Comparing these fluxes with the previous year (i.e., results from a reference study), the fluxes decreased by a factor of 1.9 and 2.6, respectively. This was potentially due to a progressive consumption of organic material, a suppression of $CH_4$ production, and aerobic and anaerobic oxidation of $CH_4$, indicating a positive evolution of the rewetted peatland into a site with moderate GHG emissions within the next years."

B) Introduction

Significant changes are proposed in the introduction.

> To set the topic into a broader context, we want to give more information in the first paragraph (from line 43).

> "Mitigating climate change requires a reduction in anthropogenic emissions of the greenhouse gases (GHGs) carbon dioxide ($CO_2$) and methane ($CH_4$) and the effective removal of $CO_2$ from the atmosphere (IPCC 2022). In all climate scenarios with a realistic probability to reach the Paris Agreement, aiming to keep anthropogenic temperature increase "well below 2 °C" (IPCCSR15, IPCC2023), land use, land use changes and forestry (LUCLUF) play an important role. Still, a large part of the hard to abate residual emissions projected in these scenarios for the 2nd half of this century come from the agricultural sector. Land use options with a large potential for climate mitigation include, for example, forestry, agriculture (pasture and cropland), wetlands, and bioenergy, (Roe et al, 2019, Shukla et al., 2019). In addition, in coastal areas, blue carbon options such as restoration and expansion of mangroves, salt marshes and seagrass meadows are suggested to have some potential for $CO_2$ removal (Macreadie et al., 2019, Duarte et al., 2013). The rewetting of formerly drained peatlands has beeen identified as one of the most promising approaches to lower $CO_2$ emission of used land, potentially even allowing turning (or re-establishing) some of these areas into $CO_2$ sinks (IPCC 2014, Wilson et al., 2016). Peatlands cover vast areas in partiuclar in Northern Europe, Northern Asia and western North America (Global Peatland Database / Greifswald Mire Centre (2024), and a large fraction of this area has been drained for agricultural use (UNEP (2022)).

> Pristine peatlands and shallow coastal regions [...]"

Additional changes will be made from line 66 onwards.

"Rewetting of degraded peatlands reduces $CO_2$ emissions by preventing aerobic decomposition of OM. The low solubility of $O_2$ and the slower transport across the overlying water body limits the availability of oxygen in the waterlogged peat soils for soil decomposition, which reduces aerobic mineralization and favors anoxic conditions, enhancing organic carbon burial (Parish et al., 2008; Kaat and Joosten, 2009). In the long-term, a re-establishment [...]"

In the sentence that begins in line 72, an addition is made.

"However, the effects of brackish water on GHG emissions are still unclear, although beneficial effects such as lower $CH_4$ emissions compared to rewetting with freshwater are likely due to the availability of sulfate ($SO_4^{2-}$), a phenomenon better investigated for some coastal ecosystems, e.g. mangroves (Cotovicz et al., 2024), which could promote the activity of sulfate-reducing bacteria (SRB)."

In the paragraph beginning in line 92, some minor changes are made to emphasize the novelty of the instrumentation we used, resulting in the following text.

"In this work, two newly developed, mostly identical lander systems were deployed, which are designed as autonomous platforms hosting a wide range of marine sensors. The landers were placed as fixed platforms on the sediment surface, and were customized for this deployment with cabled power supply and uninterrupted high-resolution data acquisition. The systems can [...]"

We would like to add more information from line 96 onward.

"We performed sensor measurements of the partial pressures of $CO_2$ and $CH_4$ and a suite of physicochemical variables (i.e, water temperature, salinity, water level, $O_2$ saturation, turbidity, phosphate, nitrate, water velocity, and chlorophyll *a*) with high temporal resolution in the range of seconds and minutes in a recently flooded peatland over a period of around nine weeks in the summer of 2021. The high-resolution measurements were combined with discrete sample analysis, and GHG emissions of $CO_2$ and $CH_4$ were derived.

   The rewetting of the coastal peatland [...]"

Due to reconstruction, the sentence in line 100„The high-resolution measurements were combined with discrete sample analysis, and GHG emissions of $CO_2$ and $CH_4$ were derived." is deleted/is moved to another position

The paragraph beginning in line 101 will be modified,resulting in the following text:

"The focus of this study is on exploring the time scales for the variability of GHG distribution and its drivers, as highly variable conditions are assumed. The nine week time-series is used to derive main cyclic as well as episodic variability in $CO_2$ and $CH_4$ concentrations and fluxes, and relate it physicochemical drivers. The impact of the temporal variability on the estimation of GHG emissions or with respect to discrete sampling strategies is assessed. By comparing GHG fluxes with a study conducted one year earlier (i.e., 2020, the first year after rewetting; Pönisch and Breznikar et al., 2023), the potential evolution towards further weakening of the $CO_2$ and $CH_4$ source strength is discussed."

C) Emphasizing the biogeochemical interpretation before the technical description/measurement technique

With this points we implemented the following major changes:

a. We have changed the sequence of chapters 2.2 and 2.1 so that the study site and the study design are introduced first and then the technology.

A new sentence must be included in line 147: "Two submersible landers equipped with sensors were used for autonomous multi-parameter investigations in the shallow water of the rewetted peatland through integrated high-resolution measurements."

The first sentence in the line 109 must be shortened: "The two novel submersible landers are platforms for advanced autonomous multi-parameter investigations in shallow water. The entirety of the carrier frame [...]"

b. Moving section 4.5, which is indeed very technical and deals with the quality of the data and improvements for further missions, to the appendix. The new section will be "Appendix F: Assessment of the data quality and implications for future lander deployments" in line 863.

In addition, the heading in line 787 is renamed to avoid confusion into „C3 Quality assessment of sensor GHG measurements".

To assure the reader that we carried out a series of quality assessment measures, we would like to insert a short paragraph at the beginning of the discussion section starting in line 438 with a link to the corresponding appendix F. Subsequently, we start the discussion of the biogeochemical findings.

"With the deployment of two novel landers in a complex and heterogeneous environment of the rewetted peatland, it is important to integrate strategies to assess the quality of the sensor data. Therefore, we have conducted various measures and analyses to build confidence in the sensor data, which are discussed in detail in Appendix F together with the future implications for the deployment of the landers. Apart from the fact that quality assessment is complex, we can show that the sensor data are suitable for interpretation based on two main analyzes: First, the similarity of the main trends in the data series from both landers strongly suggests the appropriate sampling strategy for dynamic ecosystems. Second, with strong effort on discrete samplings and laboratory analysis, we observed both good agreements but also discrepancies compared to the sensor data. With all quality measures applied, we were able to achieve a robust post-processing which allows comprehensive biogeochemical interpretations."

D) Discussion of our results on CO2 and CH4 fluxes in a broader scientific/existing literature

a) We summarized GHG flux data from other environments at the terrestrial-marine interface, which will placed in the Appendix F2 "Selected greenhouse gas emissions of CO2 and CH4 along the land-sea-interface" in line 863:

"Table 1: Selected greenhouse gas emissions of CO2 and CH4 along the land-sea-interface in relation to the derived GHG fluxes from our study.

Carbon dioxide fluxes

| from land | from streams | from restored peatland (s) | from this study | from open shallow water (brackish/salty) |
|-----------|-------------|---------------------------|-----------------|------------------------------------------|

| | | 0.02 g m⁻² h⁻¹ (open water) to 0.09 g m⁻² h⁻¹ (emergent vegetation stands, Germany)[3]

 −0.04 g CO₂ eq. m⁻² h⁻¹ (review of 38 restored peatlands)[4] | | |
|---|---|---|---|---|
| 0.07 g m⁻² h⁻¹ (drained unutilized land)[1] | −0.03–0.24 g m⁻² h⁻¹ (review with 34 study sites about streams in temperate Europe)[2] | | 0.12 ± 0.16g m⁻² h⁻¹ | 0.01 g m⁻² h⁻¹ (Bornholm sea)[5] 0.0007 g m⁻² h⁻¹ (Bothnian Bay)[6] |
| References of adapted numbers: 1 Tiemeyer et al., (2020); 2 Mwangada et al., (2023); 3 Franz et al., (2016); 4 Bianchi et al., (2020); 5 Thomas and Schneider, (1999); 6 Löffler et al., (2012) | | | | |

Methane fluxes

| from land | from streams | from restored peatland (s) | from this study | from open shallow water (brackish/salty) |
|---|---|---|---|---|
| 0.6 mg m⁻² h⁻¹ (drained unutilized land)[1] | 1.3–12.8 mg m⁻² h⁻¹ (Donau river, Germany)[2] | 1.48 mg m⁻² h⁻¹ (emergent vegetation stands) to 6.05 mg m⁻² h⁻¹ (open water, Germany)[3]

 29.68 mg m⁻² h⁻¹ (occasional brackish impact)[4]

 3.2 mg m⁻² h⁻¹ (rewetted organic soils)[1] | 0.51 ± 0.56 mg m⁻² h⁻¹ | 39.9– 104.2 mg m⁻² h⁻¹ (June/July, shallow water of the Baltic Sea)[5]

 0.015– 0.024 mg m⁻² h⁻¹ (continental shelves)[6] |
| References of adapted numbers: 1 Tiemeyer et al., (2020); 2 Lorke and Burgis, (xxxx); 3 Franz et al., (2016); 4 Hahn et al., (2015), 5 Heyer and Berger, (2000), 6 Bange et al., (1994) | | | | |

"

b) We plan to insert a new section "4.4.1 Assessment of the GHG fluxes with fluxes at the land-sea-interface", in which the fluxes from our peatland are placed in a broader context, based on the almost complete Table 1.

Section 4.4 must be altered and shortened so that 4.4.1 can be added. The old text was changed from line 572.

"Greenhouse gas fluxes for $CO_2$ and $CH_4$ could be derived from the high-resolution sensor data as measurements were made narrowly below the water surface (< 1.25 m) and a direct coupling of water at the lander with the surface water was assumed. Although the peatland showed a slight $CO_2$ uptake in early June (**Fehler! Verweisquelle konnte nicht gefunden werden.**a), accompanied by stable but slightly decreasing chlorophyll *a* concentrations, the ASE was clearly dominated by a flux of $CO_2$ to the atmosphere. This amounted to $0.12 \pm 0.16$ g m⁻² h⁻¹ derived from both landers. Increasing fluxes through early July stabilized with a simultaneous strong increase in chlorophyll *a* concentration. Overall, $CO_2$ emissions in the peatland were controlled by the simultaneous occurrence of primary production and mineralization, with the latter predominating for an overall net $CO_2$ outgassing. The derived $CH_4$ fluxes of $0.51 \pm 0.56$ mg m⁻² h⁻¹ showed a stable development during the measurement period with a slight trend to lower fluxes in August (**Fehler! Verweisquelle konnte nicht gefunden werden.**b), also strongly controlled by mineralization prozesses of OM."

The planned section 4.4.1 "Assessment of the GHG fluxes with fluxes at the land-sea-interface" will be structured by following the bullet points. The main focus will be on the assessment of our fluxes in relation to the magnitude of the fluxes at the sea-land-interface, in relation to the variability (e.g., diurnal cyclicity) and in relation to the magnitude of CH4 release compared to freshwater systems

CO2

- Peatland fluxes in the study area are around one order of magnitude higher than emissions reported in other peatland studies.
- In a shallow lake formed on a formerly drained fen, CO2 emissions 9 years after flooding ranged from $0.02$ g m$^{-2}$ h$^{-1}$ (open water) to $0.09$ g m$^{-2}$ h$^{-1}$ (emergent vegetation stands) (Franz et al., 2016).
- Compared to land-based emissions (e.g., drained unused land, cropland, forestry), emissions in the study area are significantly higher (Tiemeyer et al., 2020).
- Greenhouse gas fluxes from rivers and streams in temperate European latitudes, often influenced by human activities, are of a similar magnitude to the study area ($-0.03$ to $0.24$ g m$^{-2}$ h$^{-1}$; Mwangada et al., 2023).
- Emissions from shallow waters of the Baltic Sea or the North Sea are much smaller than those from the rewetted brackish peatland (Thomas and Schneider, 1994; Löffler et al., 2012).

CH4

- Derived $CH_4$ fluxes are significantly lower than those from peatlands which were rewetted with brackish water (Couwenberg et al., 2011; Hahn et al., 2015; Franz et al., 2016).
- $CH_4$ emissions from a shallow lake rewetted with freshwater ranged from $1.48$ mg m$^{-2}$ h$^{-1}$ (emergent vegetation stands) to $6.05$ mg m$^{-2}$ h$^{-1}$ (open water), even 9 years post-rewetting (Franz et al., 2016).
- In a dry fen converted to a shallow lake with occasional brackish water, $CH_4$ fluxes reached $29.68$ mg m$^{-2}$ h$^{-1}$ in the first year post-rewetting (Hahn et al., 2015).
- Derived $CH_4$ fluxes are comparable to $CH_4$ emissions from drained, unused land-based systems (Tiemeyer et al., 2020).
- Derived $CH_4$ fluxes are much lower than those reported for the German river Donau ($1.3$–$12.8$ mg m$^{-2}$ h$^{-1}$; Lorke and Burgis). Also variability is much lower.
- Shallow coastal waters show high $CH_4$ flux variability, e.g., Baltic Sea summer fluxes: $39.9$–$104.2$ mg m$^{-2}$ h$^{-1}$
- Continental shelf fluxes are lower compared to our fluxes and amounted to $0.015$–$0.024$ mg m$^{-2}$ h$^{-1}$ (Heyer and Berger, 2020; Bange et al., 1994).

**2. Smaller comments**

**Line 30)** *The diurnal variability led to a pronounced discrepancy between the measurements during the day and at night as well as depending on the location, resulting in CO 2 and CH4 fluxes that varied by a factor of 2.1–2.3 and 2.3–3.0, respectively.*

**The sentence is not fluent, and discrepancy might sound too negative for what you are describing.**

> Reply: Thank you for pointing this out. Differences between daytime and nighttime GHG fluxes were expected, but of this magnitude were a key finding of this study. Such differences make it clear that discrete samplings during daytime can lead to difficulties in interpretation. We would suggest the following adjustments:

> "Depending on the location, the diurnal variability led to pronounced differences between the measurements during the day and night, so that the CO2 and CH4 fluxes varied by a factor of 2.1–2.3 and 2.3–3.0, respectively, with higher fluxes occurring over daytime."

> Furthermore, to make the figures easier to understand, we provided the information on the diurnal differences recorded by the landers in line 427:

> "Hence, atmospheric GHG fluxes are 2.1- (lander 2) and 2.3-fold (lander 1) higher for $pCO_2$ and 2.3- (lander 2) and 3.0-fold (lander 1) higher for $pCH_4$ during the day than at night (**Fehler! Verweisquelle konnte nicht gefunden werden.**)."

**Line 74)** *However, the effects of brackish water on GHG emissions are still unclear, although beneficial effects such as lower CH4 emissions compared to rewetting with freshwater are likely due to the availability of sulfate (SO42−)*

**One sentence to explain here why SO4 is beneficial would be good**.

> Reply: The information as to why SO4 is probably positive comes in the following sentences and is closely related to the triggered activity of SRB. However, I can understand your point and therefore we restructured the entire part, beginning from line 72 until the end of the paragraph:

> "However, the effects of brackish water on GHG emissions are still unclear, although beneficial effects such as lower $CH_4$ emissions compared to rewetting with freshwater are likely due to the availability of sulfate ($SO_4^{2-}$), which could promote the activity of sulfate-reducing bacteria (SRB). SRB can limit $CH_4$ production, as they outcompete methane-producing microorganisms (methanogens) for substrates (Segers and Kengen, 1998; Jørgensen, 2006; Segarra et al., 2013). Further, the availability of SO42− favors anaerobic oxidation of methane (AOM), which could keep $CH_4$ emissions low (e.g., Boetius et al., 2000; Knittel and Boetius, 2009). In addition to the expected positive effect on the reduction of $CH_4$ emissions, flooding with brackish water can reduce $CO_2$ emissions by avoiding the former aerobic peat decomposition."

**Line 92) Is it possible to add one sentence, somehow describing what a lander is? Most people wont be familiar with this term. Is it a floating device? Ankered to the ground? Does it move around within a certain area? Approx size? I know that it is described in more detail later, but it helps readers to have a little bit of an idea already.**

> Reply: This is an important point, therefore we would like to add the following in line 93: "In this work, two lander systems were used, […]. The landers were placed as fixed platforms on the sediment surface with cabled power supply."

**Line 105: In the Introduction, twice a reference to an earlier study is made (Pönisch and Breznikar et al., 2023). Since your study seems to be a follow up study of this earlier study, maybe the reader can be informed briefly what were the main findings of the first study were, and how your set up/goals were different (maybe more details in discussion, but maybe already 1 or 2 sentences in the Introduction). In this way, your article is readable without forcing the reader to look up the othr article.**

> Reply: This is a good hint. One main outcome was the comparison of the "brackish GHG fluxes" to "freshwater GHG fluxes". Therefore, we will add two sentence after line 101. :
>
> "[…]. Pönisch and Breznikar et al. (2023) showed that $CO_2$ fluxes were high in the first year of rewetting with brackish water, while $CH_4$ fluxes were low compared to freshwater rewetting. Their study however relied on weekly to biweekly discrete water sampling and could not ressolve variabilty on shorter time scales."

**Line 146: wrong reference formatting. This is visible at more places in article, please check this. (for example, line 433:**

*Moreover, we used the published data from (Pönisch and Breznikar et al., 2023)*

**instead of**

*Moreover, we used the published data from Pönisch and Breznikar et al., (2023)*

> Reply: Thanks for the comment. The formatting will match the journal's guidelines in the typeset article. For clarification on this particular reference: This publication is a shared first-authorship with both first authors contributing equally to the work.. The citation program presently used has trouble to correctly represent this, but it will be adjusted to match the journal's guidelines in the final typeset manuscript.

**Line 164: The CONTROS HydroC® CO2 only goes to 1000 uatm. I assume that the authors have ordered a different version/adaptation? Specify that.**

> Reply: Thank you for your comment and attention to detail. The manufacturer's product data sheet states the measuring range as 200 - 1000 µatm, however, with the footnote that other ranges are possible on request. This is the case here: We use the regular version of the sensor, but have asked the manufacturer whether he can extend the calibration range until 6000 µatm. This is possible without technical adjustments, only with adaptation of the calibration points and calibration curve to the desired, extended range. This process was also done in another study and this is already stated in the appendix (see line 760): "In a recent study, an NDIR unit was calibrated up to 25,000 µatm with only minor adjustments of the polynomial calibration curve (Canning et al., 2021)."

**Line 171: *The resolution of data acquisition was set to 5 minutes (min) for lander 1 and 10 min for lander 2:* Why the difference? Add a sentence to clarify this.**

> Reply: The difference is due to a problem with the programming of the sensor or, better stated, a problem with the data processing unit (software problem). After discovering this during deployment, we decided to leave the settings as they were. There is no scientific reason for this. I will insert the following sentence after the sentence in line 172:
>
> "[…] was set to 5 minutes (min) for lander 1 and 10 min for lander 2. A technical limitation led to the different recording interval."

**Line 314 (and other places): it is unclear which instrument measures pCH4 (ppm) and which measure cCH4 (nmol L), and which formula is used to convert between both. Add a reference or formula.**

Reply: Thanks for the comment. I will try to solve this situation before line 314 by adapting the data handling section in line 259:

"Data processing, analysis and visualization were performed using R (R Core Team, 2022). The R packages that were used to calculate $p$CO2 (based on bottle CT and pH data), to convert $p$CH4 (measured by HC-CH4 sensors) to concentrations, and CH4 concentrations (derived from bottle data) to pCH4 are descried in Appendix D."

Furthermore, we added more information in Appendix D. We added a further sentence and a reference in line 844.

"The conversion between pCH4 and cCH4 was performed according to Wiesenburg and Guinasso (1979)."

**Line 497: white space missing**

Reply: Thanks. I also changed the number "12" to the word "twelve", as the numbers up to twelve are normally written as words.

**Line 565: Clear comparison to literature for fluxes. But why is this not done a little bit for concentrations pCH4 and pCO2. Are these concentrations high compared to sea water? Or (German) rivers? As also mentioned in the first general comment, place your results better in perspective.**

Reply: With the two major comments by you and by referee 2, we have changed parts of the discussion section to put the results in a broader context.

**Line 596: *The most important ones are*. → sentence incomplete**

Reply: Thanks. A punctuation point was wrong. The sentence will be changed to: "The most important ones are slightly different sampling height, which was in Pönisch and Breznikar et al., 2023 ~ 20 cm below the water surface and ~ 60–90 cm in our study; different sampling approach; and possible inter-annual variations, which cannot be addressed with only two years."

**Line 600: *The comparison suggests that the CO2 and CH4 fluxes in the second summer after inundation were lower by a factor of1.9 and 2.6, respectively compared to the first (Table 2).***

**These are interesting numbers. Can they be compared to numbers from other studies? Could such a reduction of 1.9 to 2.6 really be real? The authors bring these numbers back in the Conclusion, so it is an important number. Discuss better.**

Reply: That's right, we also found these figures interesting, but they should be treated with caution. We also pointed this out in the subsequent discussion - interannual variability and different methods lead to complex comparability.

Unfortunately, a comparison with other studies is very difficult, as there are hardly any/no comparable boundary conditions for which a comparison would be appropriate.

We assume that the decline is in particular due to the progressive decomposition of organic material. OM can come from dead plant biomass, peat or newly formed biomass through primary production. As the Pönisch and Breznikar study identified this availability of OM as

the main driver for high CO2 emissions, it seems realistic that a reduction in the availability of OM for biological processes will reduce emissions. In principle, the reduction seems realistic to us.

Overall, we think that the basic arguments of the discussion are already there, but need to be better linked in the text. We have therefore rearranged the section and linked it better. We have also added literature. We did changes from line 604

"Although, as mentioned above, interannual variability rather than a trend cannot be excluded [...]. In general, the decay of vegetation from the formerly drained peatland and the decomposition of the organic-rich topsoil, foster a strong mineralization of OM and promote CO2 and CH4 production (Heyer and Berger, 2000; Hahn-Schöfl et al., 2011). The observed decline between the first and second year is realistic, as this decomposition of OM was to be expected and represents a typical transition phase (Kalhori et al., 2024)."

---

## Author Comment (AC2)

Reviewer 2,

thank you very much for your detailed and constructive feedback provided within your major and smaller comments, which helped us to improve the manuscript.

The original manuscript and the study design focused on determining temporal variability and exploring biogeochemical interactions in summer, probably the most interesting season regarding GHG dynamics, and in a very complex and heterogeneous ecosystem. Many coastal peatlands (about 40,000 ha in MV) could be rewetted in this way, so there is a large potential and importance for rewetting coastal peatlands, partly with the option of rewetting with brackish waters. Due to the complex ecosystem and novel technology we used, we put strong emphasis on technical description, data analysis and interpretation of the underlying drivers and biogeochemical processes.

We also saw the issue that there is little literature on the rewetting of coastal peatlands by means of dyke removal. Furthermore, comparability to other studies is limited by very different environmental settings, so that the existing literature and data must be analyzed in a complex way, for example, by extracting only measurements from the summer months, to reach comparability to our measurements.

However, we understand that both reviewers saw the shortcoming of not putting our work in a broader perspective and suggested a better discussion of the findings in the framework of existing literature, and we are very thankful that the reviewers pointed this out.

We have taken many measures to reorganize and improve the text based on the major comments. We have analyzed the comments of both referees (RF1 and RF2) and identified major commonalities and strong similarities in the argumentation. We adressed these similarities between the referee comments by joint measures in the two responses to the referees (authors response (ACs)) which prompted us to formulate identical responses that address the following:

A) Restructuring of the abstract to better address the scientific question of greenhouse gas dynamics and the underlying biogeochemical drivers
B) Restructuring of the introduction to provide more context on the relevance of greenhouse gas reduction and to indicate that novel technology was used for this study
C) We tried to improve the clarity of the scientific question by emphasizing the biogeochemical interpretation before the technical description at appropriate points. For this reason, we also moved the discussion on the match between sensor data and results from discrete water sampling (former chapter 4.5) to the appendix.
D) We did our best to discuss our results on CO2 and CH4 fluxes in a broader scientific context by adding a new section 4.4.1. to the discussion.

In the following, we have reposted the comments by the reviewer (in bold) and placed our responses below them. Envisaged text changes/amendments are indicated by quotation marks.

1. **General comments**

The article presents interesting new data from an understudied ecosystem. It is well written and graphs are well presented and appropriate, while data and observed phenomena are well described. However, it is insufficiently placed within the context of the ecosystem measured and the potential wider implications of the ecosystem processes revealed. The emphasis of the article is overly weighted towards the measurement technique which is not claimed as novel. As I read it, the interest here is instead in presenting high resolution data from this particular ecosystem. Detail about "the landers" is placed ahead of the ecosystem in question throughout the article. The analysis of the data is there but it is not sufficiently presented in contrast with existing literature on ecosystem GHG production, fluxes and the processes that drive them.

The data and analysis presented here are certainly worthy of publishing, but I would say the article needs a substantial restructuring to change the focus, starting with a stronger emphasis on the research questions relating to the ecosystem.

Reply:

A) Abstract

Significant changes are proposed for the abstract to better reflect the clarifications and extensions envisaged for the revised manuscript.

- Shortening of a section which will be moved to the introduction (line 17-21)
- Inclusion of a sentence highlighting the novelty of the measurement technology (line 24)
- Inclusion of the Spearman correlation to better indicate the focus on the analysis of drivers and biogeochemical processes (line 26)

Taking account also minor changes requested by RF1 and RF2, we wil provide a new version of the abstract:

**"Abstract.** Rewetting peatlands is an important measure to reduce greenhouse gas (GHG) emissions from land use change. After rewetting, the areas can be highly heterogeneous in terms of GHG exchange and depend, for example, on water level, vegetation, temperature, previous use, and duration of rewetting. Here, we present a study of a coastal peatland that was rewetted by brackish water from the Baltic Sea and thus became part of the coastal shallow Baltic Sea water system through a permanent hydrological connection. Environmental heterogeneity and the brackish water column formation, require improved quantification techniques to assess local sinks and sources of atmospheric GHGs. We conducted nine weeks of autonomous and high-resolution, sensor-based bottom water measurements of marine physical and chemical variables at two locations in a permanently flooded peatland in summer 2021, the $2^{nd}$ year after rewetting. For the study, we used newly developed multi-sensor platforms (landers) customized for this operation. Results show considerable temporal fluctuations of $CO_2$ and $CH_4$, expressed as multi-day, diurnal and event-based variability and spatial differences for variables dominantly influenced by biological processes. Episodic and diurnal drivers are identified and discussed based on Spearman correlation analysis. The multi-day variability resulted in a pronounced magnitude of measured GHG partial pressures during the deployment ranging between 295.0–8937.8 µatm ($CO_2$) and 22.8–2681.3 µatm (correspond to 42.7–3568.6 nmol $L^{-1}$; $CH_4$), respectively. In addition, the variability of the GHGs, temperature, and oxygen was characterized by pronounced diurnal cycles, resulting e.g., in a mean daily variability of 4066.9 µatm for $CO_2$ and 1769.6 µatm for $CH_4$. Depending on the location, the diurnal variability led to pronounced differences between the measurements during the day and night, so that the $CO_2$ and $CH_4$ fluxes varied by a factor

of 2.1–2.3 and 2.3–3.0, respectively, with higher fluxes occurring over daytime. The rewetted peatland was further impacted by fast system changes (events) such as storm, precipitation and major water level changes, which impacted biogeochemical cycling and GHG partial pressures. The derived average GHG exchange amounted to $0.12 \pm 0.16$ g m$^{-2}$ h$^{-1}$ ($CO_2$) and $0.51 \pm 0.56$ mg m$^{-2}$ h$^{-1}$ ($CH_4$), respectively. These fluxes are high ($CO_2$) to low ($CH_4$) compared to studies from temperate peatlands rewetted with freshwater. Comparing these fluxes with the previous year (i.e., results from a reference study), the fluxes decreased by a factor of 1.9 and 2.6, respectively. This was potentially due to a progressive consumption of organic material, a suppression of $CH_4$ production, and aerobic and anaerobic oxidation of $CH_4$, indicating a positive evolution of the rewetted peatland into a site with moderate GHG emissions within the next years."

B) Introduction

Significant changes are proposed in the introduction.

To set the topic into a broader context, we want to give more information in the first paragraph (from line 43).

"Mitigating climate change requires a reduction in anthropogenic emissions of the greenhouse gases (GHGs) carbon dioxide ($CO_2$) and methane ($CH_4$) and the effective removal of $CO_2$ from the atmosphere (IPCC 2022). In all climate scenarios with a realistic probability to reach the Paris Agreement, aiming to keep anthropogenic temperature increase "well below 2 °C" (IPCCSR15, IPCC2023), land use, land use changes and forestry (LUCLUF) play an important role. Still, a large part of the hard to abate residual emissions projected in these scenarios for the 2$^{nd}$ half of this century come from the agricultural sector. Land use options with a large potential for climate mitigation include, for example, forestry, agriculture (pasture and cropland), wetlands, and bioenergy, (Roe et al, 2019, Shukla et al., 2019). In addition, in coastal areas, blue carbon options such as restoration and expansion of mangroves, salt marshes and seagrass meadows are suggested to have some potential for $CO_2$ removal (Macreadie et al., 2019, Duarte et al., 2013). The rewetting of formerly drained peatlands has beeen identified as one of the most promising approaches to lower $CO_2$ emission of used land, potentially even allowing turning (or re-establishing) some of these areas into $CO_2$ sinks (IPCC 2014, Wilson et al., 2016). Peatlands cover vast areas in partiuclar in Northern Europe, Northern Asia and western North America (Global Peatland Database / Greifswald Mire Centre (2024), and a large fraction of this area has been drained for agricultural use (UNEP (2022)).

Pristine peatlands and shallow coastal regions [...]"

Additional changes will be made from line 66 onwards.

"Rewetting of degraded peatlands reduces $CO_2$ emissions by preventing aerobic decomposition of OM. The low solubility of $O_2$ and the slower transport across the overlying water body limits the availability of oxygen in the waterlogged peat soils for soil decomposition, which reduces aerobic mineralization and favors anoxic conditions, enhancing organic carbon burial (Parish et al., 2008; Kaat and Joosten, 2009). In the long-term, a re-establishment [...]"

In the sentence that begins in line 72, an addition is made.

"However, the effects of brackish water on GHG emissions are still unclear, although beneficial effects such as lower $CH_4$ emissions compared to rewetting with freshwater are likely due to the availability of sulfate ($SO_4^{2-}$), a phenomenon better investigated for some coastal ecosystems, e.g. mangroves (Cotovicz et al., 2024), which could promote the activity of sulfate-reducing bacteria (SRB)."

In the paragraph beginning in line 92, some minor changes are made to emphasize the novelty of the instrumentation we used, resulting in the following text.

"In this work, two newly developed, mostly identical lander systems were deployed, which are designed as autonomous platforms hosting a wide range of marine sensors. The landers were placed as fixed platforms on the sediment surface, and were customized for this deployment with cabled power supply and uninterrupted high-resolution data acquisition. The systems can [...]"

We would like to add more information from line 96 onward.

"We performed sensor measurements of the partial pressures of $CO_2$ and $CH_4$ and a suite of physicochemical variables (i.e, water temperature, salinity, water level, $O_2$ saturation, turbidity, phosphate, nitrate, water velocity, and chlorophyll *a*) with high temporal resolution in the range of seconds and minutes in a recently flooded peatland over a period of around nine weeks in the summer of 2021. The high-resolution measurements were combined with discrete sample analysis, and GHG emissions of $CO_2$ and $CH_4$ were derived.

The rewetting of the coastal peatland [...]"

Due to reconstruction, the sentence in line 100 „The high-resolution measurements were combined with discrete sample analysis, and GHG emissions of $CO_2$ and $CH_4$ were derived." is deleted/is moved to another position

The paragraph beginning in line 101 will be modified,resulting in the following text:

"The focus of this study is on exploring the time scales for the variability of GHG distribution and its drivers, as highly variable conditions are assumed. The nine week time-series is used to derive main cyclic as well as episodic variability in $CO_2$ and $CH_4$ concentrations and fluxes, and relate it physicochemical drivers. The impact of the temporal variability on the estimation of GHG emissions or with respect to discrete sampling strategies is assessed. By comparing GHG fluxes with a study conducted one year earlier (i.e., 2020, the first year after rewetting; Pönisch and Breznikar et al., 2023), the potential evolution towards further weakening of the $CO_2$ and $CH_4$ source strength is discussed."

**C) Emphasizing the biogeochemical interpretation before the technical description/measurement technique**

With this points we implemented the following major changes:

a. We have changed the sequence of chapters 2.2 and 2.1 so that the study site and the study design are introduced first and then the technology.

A new sentence must be included in line 147: "Two submersible landers equipped with sensors were used for autonomous multi-parameter investigations in the shallow water of the rewetted peatland through integrated high-resolution measurements."

The first sentence in the line 109 must be shortened: "The two novel submersible landers are platforms for advanced autonomous multi-parameter investigations in shallow water. The entirety of the carrier frame [...]"

b.  Moving section 4.5, which is indeed very technical and deals with the quality of the data and improvements for further missions, to the appendix. The new section will be "Appendix F: Assessment of the data quality and implications for future lander deployments" in line 863.

In addition, the heading in line 787 is renamed to avoid confusion into „C3 Quality assessment of sensor GHG measurements".

To assure the reader that we carried out a series of quality assessment measures, we would like to insert a short paragraph at the beginning of the discussion section starting in line 438 with a link to the corresponding appendix F. Subsequently, we start the discussion of the biogeochemical findings.

"With the deployment of two novel landers in a complex and heterogeneous environment of the rewetted peatland, it is important to integrate strategies to assess the quality of the sensor data. Therefore, we have conducted various measures and analyses to build confidence in the sensor data, which are discussed in detail in Appendix F together with the future implications for the deployment of the landers. Apart from the fact that quality assessment is complex, we can show that the sensor data are suitable for interpretation based on two main analyzes: First, the similarity of the main trends in the data series from both landers strongly suggests the appropriate sampling strategy for dynamic ecosystems. Second, with strong effort on discrete samplings and laboratory analysis, we observed both good agreements but also discrepancies compared to the sensor data. With all quality measures applied, we were able to achieve a robust post-processing which allows comprehensive biogeochemical interpretations."

**D) Discussion of our results on CO2 and CH4 fluxes in a broader scientific/existing literature**

a)  We summarized GHG flux data from other environments at the terrestrial-marine interface, which will placed in the Appendix F2 "Selected greenhouse gas emissions of CO2 and CH4 along the land-sea-interface" in line 863:

*"Table 1: Selected greenhouse gas emissions of CO2 and CH4 along the land-sea-interface in relation to the derived GHG fluxes from our study.*

Carbon dioxide fluxes

| from land | from streams | from restored peatland (s) | from this study | from open shallow water (brackish/salty) |
|---|---|---|---|---|
| 0.07 g m$^{-2}$ h$^{-1}$ (drained unutilized land)[1] | −0.03–0.24 g m$^{-2}$ h$^{-1}$ (review with 34 study sites about streams in temperate Europe)[2] | 0.02 g m$^{-2}$ h$^{-1}$ (open water) to 0.09 g m$^{-2}$ h$^{-1}$ (emergent vegetation stands, Germany)[3]  −0.04 g $CO_2$ eq. m$^{-2}$ h$^{-1}$ (review of 38 restored peatlands)[4] | 0.12 ± 0.16g m$^{-2}$ h$^{-1}$ | 0.01 g m$^{-2}$ h$^{-1}$ (Bornholm sea)[5] 0.0007 g m$^{-2}$ h$^{-1}$ (Bothnian Bay)[6] |
| References of adapted numbers: 1 Tiemeyer et al., (2020); 2 Mwangada et al., (2023); 3 Franz et al., (2016); 4 Bianchi et al., (2020); 5 Thomas and Schneider, (1999); 6 Löffler et al., (2012) | | | | |

Methane fluxes

| from land | from streams | from restored peatland (s) | from this study | from open shallow water (brackish/salty) |
|---|---|---|---|---|
| 0.6 mg m$^{-2}$ h$^{-1}$ (drained unutilized land)[1] | 1.3–12.8 mg m$^{-2}$ h$^{-1}$ (Donau river, Germany)[2] | 1.48 mg m$^{-2}$ h$^{-1}$ (emergent vegetation stands) to 6.05 mg m$^{-2}$ h$^{-1}$ (open water, Germany)[3]

29.68 mg m$^{-2}$ h$^{-1}$ (occasional brackish impact)[4]

3.2 mg m$^{-2}$ h$^{-1}$ (rewetted organic soils)[1] | 0.51 ± 0.56 mg m$^{-2}$ h$^{-1}$ | 39.9–104.2 mg m$^{-2}$ h$^{-1}$ (June/July, shallow water of the Baltic Sea)[5]

0.015–0.024 mg m$^{-2}$ h$^{-1}$ (continental shelves)[6] |
| References of adapted numbers: 1 Tiemeyer et al., (2020); 2 Lorke and Burgis, (xxxx); 3 Franz et al., (2016); 4 Hahn et al., (2015), 5 Heyer and Berger, (2000), 6 Bange et al. (1994) | | | | |

"

b) We plan to insert a new section "4.4.1 Assessment of the GHG fluxes with fluxes at the land-sea-interface", in which the fluxes from our peatland are placed in a broader context, based on the almost complete Table 1.

Section 4.4 must be altered and shortened so that 4.4.1 can be added. The old text was changed from line 572.

"Greenhouse gas fluxes for $CO_2$ and $CH_4$ could be derived from the high-resolution sensor data as measurements were made narrowly below the water surface (< 1.25 m) and a direct coupling of water at the lander with the surface water was assumed. Although the peatland showed a slight $CO_2$ uptake in early June (**Fehler! Verweisquelle konnte nicht gefunden werden.**a), accompanied by stable but slightly decreasing chlorophyll $a$ concentrations, the ASE was clearly dominated by a flux of $CO_2$ to the atmosphere. This amounted to 0.12 ± 0.16 g m$^{-2}$ h$^{-1}$ derived from both landers. Increasing fluxes through early July stabilized with a simultaneous strong increase in chlorophyll $a$ concentration. Overall, $CO_2$ emissions in the peatland were controlled by the simultaneous occurrence of primary production and mineralization, with the latter predominating for an overall net $CO_2$ outgassing. The derived $CH_4$ fluxes of 0.51 ± 0.56 mg m$^{-2}$ h$^{-1}$ showed a stable development during the measurement period with a slight trend to lower fluxes in August (**Fehler! Verweisquelle konnte nicht gefunden werden.**b), also strongly controlled by mineralization prozesses of OM."

The planned section 4.4.1 "Assessment of the GHG fluxes with fluxes at the land-sea-interface" will be structured by following the bullet points. The main focus will be on the assessment of our fluxes in relation to the magnitude of the fluxes at the sea-land-interface, in relation to the variability (e.g., diurnal cyclicity) and in relation to the magnitude of CH4 release compared to freshwater systems

CO2

- Peatland fluxes in the study area are around one order of magnitude higher than emissions reported in other peatland studies.
- In a shallow lake formed on a formerly drained fen, CO2 emissions 9 years after flooding ranged from 0.02 g m$^{-2}$ h$^{-1}$ (open water) to 0.09 g m$^{-2}$ h$^{-1}$ (emergent vegetation stands) (Franz et al., 2016).
- Compared to land-based emissions (e.g., drained unused land, cropland, forestry), emissions in the study area are significantly higher (Tiemeyer et al., 2020).

- Greenhouse gas fluxes from rivers and streams in temperate European latitudes, often influenced by human activities, are of a similar magnitude to the study area ($-0.03$ to $0.24$ g m$^{-2}$ h$^{-1}$; Mwangada et al., 2023).
- Emissions from shallow waters of the Baltic Sea or the North Sea are much smaller than those from the rewetted brackish peatland (Thomas and Schneider, 1994; Löffler et al., 2012).

CH4

- Derived CH$_4$ fluxes are significantly lower than those from peatlands which were rewetted with brackish water (Couwenberg et al., 2011; Hahn et al., 2015; Franz et al., 2016).
- CH$_4$ emissions from a shallow lake rewetted with freshwater ranged from $1.48$ mg m$^{-2}$ h$^{-1}$ (emergent vegetation stands) to $6.05$ mg m$^{-2}$ h$^{-1}$ (open water), even 9 years post-rewetting (Franz et al., 2016).
- In a dry fen converted to a shallow lake with occasional brackish water, CH$_4$ fluxes reached $29.68$ mg m$^{-2}$ h$^{-1}$ in the first year post-rewetting (Hahn et al., 2015).
- Derived CH$_4$ fluxes are comparable to CH$_4$ emissions from drained, unused land-based systems (Tiemeyer et al., 2020).
- Derived CH$_4$ fluxes are much lower than those reported for the German river Donau ($1.3$–$12.8$ mg m$^{-2}$ h$^{-1}$; Lorke and Burgis). Also variability is much lower.
- Shallow coastal waters show high CH$_4$ flux variability, e.g., Baltic Sea summer fluxes: $39.9$–$104.2$ mg m$^{-2}$ h$^{-1}$
- Continental shelf fluxes are lower compared to our fluxes and amounted to $0.015$–$0.024$ mg m$^{-2}$ h$^{-1}$ (Heyer and Berger, 2020; Bange et al., 1994).

**2. Technical comments**

**Line 19 – "to" is not an appropriate connector in "unlike to rewetting" and to make a subtler point there is a logical inconsistency in saying "unlike x, y is less studied", either of these alternatives would be more natural: "compared with x, y is less studied" or "unlike x, y remains understudied".**

Reply: Thank you for this comment. We want to change the sentence in line 19 to:

"Compared with rewetting with freshwater, the effects of rewetting with brackish, sulfate-containing water are less studied, although positive effects are expected as sulfate-reducing bacteria may become established and might out-compete methane-producing archaea (methanogens) for substrates, resulting in lower CH4 emissions."

**Line 67 – grammar errors in "considered as one of the potent measure". "Considered to be" is the appropriate construction here and "measure" should be plural. But it would be more appropriate to use an indefinite article eg. "considered to be a potent measure".**

Reply: Thank you, we want to correct the error by changing the sentences in line 67 to:

"Rewetting of degraded peatlands reduces $CO_2$ emissions by preventing aerobic decomposition of OM and is considered to be a potent measure to mitigate global warming through land-use change."

**Line 105 – I have not seen this double author et al. style before. It seems you want to give equal credit to the first two authors? In any case, this is a question for the editors.**

Reply: This publication is a "shared first authorship" with both first authors contributing equally to the work. The citation style of "Pönisch and Breznikar et al. (2023)" will be adjusted to match the journal's guidelines in the final typeset manuscript.

**Line 108 - Perhaps a different term to 'lander' could be used since this term is not likely familiar to more terrestrial peatland ecologists.**

Reply: You may be right. Reviewer 1 also made a similar comment. We have therefore already inserted a sentence in line 93 ("The landers were placed as fixed platforms on the sediment surface with cabled power supply."). In addition, we would like to make the following changes to the sentence from line 109 onwards.

"Two submersible platforms were used for autonomous multi-parameter investigations in the shallow water (~ 0.5 m) of the rewetted peatland through integrated high-resolution sensor measurements. The entirety of the carrier frame, the power supply, the technical units for sensor control and the sensors are referred to as landers and were deployed as stationary measuring units at the sediment-water surface. Each lander system was equipped with […]"

**Line 113 – "customized deployments"? What information is meant to be conveyed here?**

Reply: The landers are highly modular in their equipment and programming. They can therefore be applied to various scopes, such as extreme events (storms), which require a particularly high measurement resolution. To convey this, we want to add the following to line 113:

"Sensor scheduling, time stamping, and data recording were centralized in the Data Processing Unit (DPU) and allow customized deployments, for example long-term deployments or shorter-term deployments during extreme events such as storms."

**Line 119 – is the indent necessary?**

Reply: We think that the list is helpful to quickly recognize important adjustments on the one hand and to signal important features of the deployment very quickly on the other hand. The final layout and level of indentation will be determined by the journal's guidelines in the typeset manuscript.

**Line 125 – annual cycle of what?**

Reply: The term can be confusing. Therefore, we want to change the sentences in line 125 to:

"The time of deployment was chosen based on the study of the annual cycle of GHG dynamics from the first year after rewetting and based on weekly to bi-weekly sampling (Pönisch and Breznikar et al., 2023), which indicated that the summer season is the most important and dynamic with respect to GHG-fluxes."

**Line 204 – "bottle data" is not the most appropriate term to refer to the manually collected dissolved gas concentration data, perhaps another phrase could be found.**

Reply: To clearly distinguish measurements based on manually collected water samples and sensor-based measurements, we want to clearly indicate their origin from the Niskin bottle sampling. Therefore, we want to keep the term "bottle data", which is also commonly used for manually sampled validation data in other hydrographic studies. However, we suggest to rearrange the paragraph from line 202 to introduce the bottle data sampling device first.

"To validate the sensor-based measurements, discrete field measurements were taken during the lander deployment at lander 1 (in the central peatland area) and at lander 2 (in the connecting channel). Undisturbed water was taken manually using a 5-L Niskin bottle. The bottle was deployed horizontally from a small working boat and its closure noted with an exact time stamp. Altogether 9-sampling sessions were carried out in the direct proximity to the landers (Table C 1; bottle data for all discrete sampled variables can be found at https://doi.org/10.1594/PANGAEA.964758; (Pönisch et al., 2024)). Water from Niskin bottle sampling was analyzed using established laboratory methods as described below."

**Line 206-207 – these 2 sentences are unnecessary.**

Reply: We think that in the context of the very strong spatial variability in the peatland water column (discussed e.g. in section 4.5) together with a shallow water column (~1 m) the information of a horizontally deployed Niskin bottle (itself being 0.4 m long) is very important. Furthermore, in a water column in which scientists can move around with waders, it cannot be readily assumed that a boat was used for sampling. To encourage good practice in follow-up studies, we want to stick to give those details in the methods section.

**Line 234 – GHGs should be $CO_2$ and $CH_4$**

Reply: We will change the term "GHGs" into CO2 and CH4 in line 234:

"All involved variables (i.e., $CO_2$, $CH_4$, wind speed, temperature, salinity, atmospheric-equilibrium conditions) were averaged hourly to obtain more robust values and matching timestamps."

**Line 240 – perhaps more could be said about the Schmidt number and the linear interpolation.**

Reply: We suggest to change the text as follows, which gives both more detail on the interpolation and reference to the source used with detailed background on the Schmidt number itself:

"The Schmidt number was approximated by a linear interpolation in salinity between the freshwater and seawater values (Wanninkhof, 2014). The Schmidt number depends on the gas, the temperature and to a minor degree, the salinity of the water."

**Line 242 - Is Utö the most appropriate data source for atmospheric GHG concentrations? I presume the authors chose it because it is also situated in a Baltic coastal setting, however it is 800 km away. If this is the most appropriate measurement site for atmospheric concentrations, some justification for its choice could be included**

Reply: The ICOS Station Üto is the nearest high quality atmospheric measurement station reflecting marine air. Inland stations, of which there are only very few in higher proximity, are more prone to local land-derived sources. An analysis of the "offset" between the marine station Mace Head (Altlantic Air masses) and Utö is given in Appendix Figure 2 of Jacobs et al, 2021 (https://doi.org/10.5194/bg-18-2679-2021). We therefore believe that Utö is the best choice for our study site.

**Line 245 - awkward reference to the previous work here, consider rephrasing**

Reply: We would like to repharse the sentence in line 245:

"Moreover, the same parameterization was used in Pönisch and Breznikar et al. (2023). The use of the same parameterization facilitated comparison of the results of our work with this previsou study."

**Line 248 – no need to mention the results table in methods.**

Reply: We will delete the link.

**Line 251 - could there be a clearer way to explain this?**

Reply: Sure. We would like change the sentences from line 251 into:

"ASE derived from our high-resolution time series to represent the day-night bias. Daily averages for 00:00 UTC ± 1 hour and 12:00 UTC ± 1 hour were isolated (resulting in ~ 200 data points for each calculation)."

**Line 258 – since this data is presented before the fluxes should this section on the methods go before ASE**

Reply: Of course and thank you for the comment, we will change section 2.6 with 2.5 to put "Data processing and analysis" before "Air-sea exchange (ASE) calculation".

**Line 260 – presumably "descried" is a typo here.**

Reply: Correct, it should read "described" and will be changed accordingly.

**Line 261 – for what scientific purpose were the correlations conducted?**

Reply: To make the purpose more clear, we want to add the following sentence in line 262:

"Spearman correlation coefficients (rs) were calculated [...]. To identify potential drivers, processes and mechanisms of CO2 and CH4 variability, the presence or absence of (strong) correlations helps to identify and discuss potential causal relationships."

**Line 262 – save figure references for results.**

Reply: Agreed. The Figure 3 reference was removed from the methods section.

**Line 277 – no unit for salinity?**

Reply: We have followed the work of Millero, F.J. 1993, (What is PSU? Oceanography 6(3):67) and Millero, F.J. 2015 (History of the Equation of State of Seawater, https://doi.org/10.5670/oceanog.2010.21), and a summary exists here (http://www.coastalwiki.org/wiki/Salinity). It was stated: "The practical salinity scale was defined as conductivity ratio with no units" and "This definition is adopted by all national and international oceanographic organizations". On this basis we suggest to keep salinity without units.

**Line 295 – unnecessary clause**

Reply: We will change to:

"Although occasionally low $O_2$ values were detected, measurements indicate a predominantly oxygenated water column, with slightly lower mean $O_2$ values at lander 2 (Figure 2f, Table 1)."

**Line 299 – "short-term"; not sure what is meant by "d; Sect. 3.2, Table 1"; how low are "lower values".**

Reply: We will change the term "short-scale" into "short-term" in line 299.

The "d" in the term "d; Sect. 3.2, Table 1" is the abbreviation for day. As the definition is not necessary because it is generally valid, we will delete the abbreviation.

In June, the pCO2 value was generally lower compared to the rest of the deployment, as can be seen from the smoothing line in the graph in Figure 2a "lower values". We would like insert "average" to make this more clear in line 299:

"Sustained lower average values occurred at the beginning of the deployment (early June), but then changed to on average higher values (> 1000 µatm) during most of the deployment at both locations."

**Line 303 – "strong negative correlation"**

Reply: As we have introduced various effect sizes with the Spearman correlation, we refer to this information for the strength of a correlation and must be linked there: The strength of a correlation must therefore be "with a strong effect size". If I also changed to "strong negative correlation", I would have "strong" twice in the sentence. We would therefore like to stick with the original sentence.

**Line 304 – the results might start with this comparison and also the part on the same comparison for $CH_4$**

Reply: With this comparison, we show how the sensor data behaves compared to the discrete data. As the results and biogeochemical features of the sensor-measurements should be the main focus, and both reviewers suggested to put a stronger focus on the biogeochemical results rather on technical considerations, we have decided to leave the statements on sensor comparison with the discrete data at the end of the respective paragraphs.

**Line 313 – rather than always referring to the landers, the locations could be used, since the location is what is of interest. This point applies throughout.**

Reply: Though the positions of the two landers is clearly described in the site description, we will introduce the site information ("central part" ; "channel); at some places in the text to make it easier for the reader to follow. Thank you for the suggestion.

**Line 352 – Short-term variability of what?**

Reply: Thank you. We would like to change the headline into:

"Short-term variability and diurnal cycles of the measured variables"

**Line 355-357 - This sentence belongs in methods.**

Reply: Agree. Please find the answer under the next comment.

**Line 364-366 – These 2 sentences also to methods**

Reply: We agree that both lines 355-357 and 364-367 should transfer into the methods. We have inserted a new paragraph after the line 267 in the section "2.6 Data handling and analysis"

"In order to show the diurnal cyclicity and the relationships between the variables affected by the diurnal cycles, the high-resolution measurement time series (lander 1 and lander 2) were divided into one-hour-bins and a mean value was calculated for each hour of the day, resulting in a diurnal distribution pattern. Furthermore, to show the magnitude of diarunal variability of the variables, the mean diurnal variability was calculated. For this purpose, we divided the high-resolution data into 24-hour intervals, each starting at midnight. Then, for each interval, the difference between the minimum and maximum was determined. Subsequent determination of the mean, minimum, and maximum yields an approximation of the magnitude of diurnal variability."

The sentences in section 3.2 will be changed accordingly from line 384:

"The variables $pCO_2$, $pCH_4$, temperature, and oxygen showed pronounced short-term variability and diurnal cyclicity, [...]. The diurnal cyclicity and the relationships between the variables affected by the diurnal cycles was made visible by calculating the distribution of hourly mean values (see section 2.5). The distribution indicated that $pCO_2$ and $pCH_4$ showed an inverse character compared to temperature, $O_2$, and the wind speed (**Fehler! Verweisquelle konnte nicht gefunden werden.**). [...]"

The sentences in section 3.2 will be changed accordingly from line 395:

"To show the magnitude of daily variability, the mean, minimum, and maximum of 24-hour intervals were calculated (see section 2.5) and summarized in Table 1. The mean daily range for $pCO_2$ of ~ 4000 µatm is substantial. [...]"

**Line 378 - why not just say hourly?**

Reply: Yes, we would like to change "one-hour-parts" into "hourly sections" in line 378 and also in line 853.

"The distribution of the mean values of pCO2, pCH4, temperature, O2 and wind speed of hourly sections over 24-hours."

**Line 379 – why switch between quantile and percentile?**

Reply: Agreed. We change the text to use percentile throughout. .

**Line 385 – "shared observation" is not really the right phrase here**

Reply: Yes, we will simplyfy as follows:

"All three events, although different in nature, […]"

**Line 408 – What caused this outflow? Some explanation would be interesting if there is one; the phrasing about the water level relative to the sensor is a little clumsy.**

Reply: The outflow occurs because the peatland is directly connected to the Kubitzer Bodden and this is connected to the Baltic Sea. If the water levels drop there, the peatland follows. In addition, local wind regimes can lead to an outflow or inflow. This has already been described in line 135. Nevertheless, we want make a change and rephrase the sentence to make it less clumsy in line in line 408.

"In late July, we observed an outflow of water from the peatland towards the *Kubitzer Bodden.* The outflow caused a strong lowering in the water column, which drained large areas of the rewetted peatland and caused the pressure measurements of the CTD-O2 sensor to reach ~ 0 dbar."

**Line 416-419 – Is this not already in methods, if not it should be moved there.**

Reply: Yes, we streamlined the first paragraph of this section as follows

"GHG fluxes for $CO_2$ and $CH_4$ were derived from the entire high-resolution sensor data and from the different scenarios: from bottle data only, during daytime, during nighttime, and using data of a previous study to isolate GHG fluxes for a direct year-to-year comparison (Table 2)."

**Line 429-436 – this table heading seems more like a section of the methods.**

Reply: You are right. With the figures and tables, we follow the idea that the objects should be understandable as independent/stand-alone objects. That is why we have added more information in the caption so that the figures can be understood without searching in the manuscript. As the calculation is not so comprehensible, we would like to stick to the rather methodical description.

**Line 439 – tell the variables**

Reply: We will change the sentence in line 439 to:

"The deployment of two landers equipped with sensors for the high-resolution determination of the marine variables pCO2, pCH4, temperature, salinity, hydrostatic pressure, $O_2$, turbidity, water velocity, $c(PO_4^{3-})$ and chlorophyll *a* in a coastal peatland revealed large temporal and spatial variations of the measured variables."

**Line 442 – A claim including the phrase "covering a comparable study area" is problematic when the study only includes measurements from 2 points, consider rephrasing.**

Reply: We understand the point. Therefore, we would like rephrase into:

"To our knowledge, there is no study that covers a comparable environmental setting with a similar temporal data resolution."

**Line 518 - remove sentence starting "The effect of this…"**

Reply: We will remove the sentence.

**Line 583-586 – this section is a little clunky**

Reply: Yes, we want to make some changes:

"The studies mentioned above report annual GHG fluxes. Since our study only covers the summer months, comparability with these studies is limited, as annual CO2 and CH4 emissions are normally highest in the late summer months. To allow a more direct comparison of the development of greenhouse gas emissions at our study site, we used the published data from Pönisch and Breznikar et al. 2023."

**Line 596 – Grammar**

Reply: Thank you. We would like to change the both sentences in line 596 into:

„In addition to a high degree of comparability due to comparable boundary conditions, this approach also has limitations. The most important of these are slightly different sampling height […]"

**Addtional changes:** We would like to make an additional change to Figure A2 by pixelating the faces.

---

## Author Response (AR2)

Dear Reviewer 1, Reviewer 2, and Editor,

thank you very much for your detailed and constructive feedback provided within your major and smaller comments, which helped us to improve the manuscript.

The original manuscript and the study design focused on determining temporal variability and exploring biogeochemical interactions in summer, probably the most interesting season regarding GHG dynamics, and in a very complex and heterogeneous ecosystem. Many coastal peatlands (about 40,000 ha in MV) could be rewetted in this way, so there is a large potential and importance for rewetting coastal peatlands, partly with the option of rewetting with brackish waters. Due to the complex ecosystem and novel technology we used, we put strong emphasis on technical description, data analysis and interpretation of the underlying drivers and biogeochemical processes.

We also saw the issue that there is little literature on the rewetting of coastal peatlands by means of dyke removal. Furthermore, comparability to other studies is limited by very different environmental settings, so that the existing literature and data must be analyzed in a complex way, for example, by extracting only measurements from the summer months, to reach comparability to our measurements.

However, we understand that both reviewers saw the shortcoming of not putting our work in a broader perspective and suggested a better discussion of the findings in the framework of existing literature, and we are very thankful that the reviewers pointed this out.

We have taken many measures to reorganize and improve the text based on the major comments. We have analyzed the comments of both referees (RF1 and RF2) and identified major commonalities and strong similarities in the argumentation. We address these similarities between the referee comments by joint measures in the two responses to the referees (authors response (ACs)) which lead us to formulate identical responses that address the following:

A) Restructuring of the abstract to better address the scientific question of greenhouse gas dynamics and the underlying biogeochemical drivers
B) Restructuring of the introduction to provide more context on the relevance of greenhouse gas reduction and to indicate that novel technology was used for this study
C) We tried to improve the clarity of the scientific question by emphasizing the biogeochemical interpretation before the technical description at appropriate points. For this reason, we have also moved the discussion on the match between sensor data and results from discrete water sampling (former chapter 4.5) to the appendix. This also helped to avoid increasing the length of the manuscript, which was requested by the editor.
D) We did our best to discuss our results on CO2 and CH4 fluxes in a broader scientific context by adding a new section 4.4.1. to the discussion.

In the following, we have listed the joint measures that address the major comments of both reviewers. Following this, we have reposted the minor/technical comments by the respective reviewer (in bold) and placed our responses directly below them. Proposed text changes/amendments are indicated by quotation marks.

**1. Major Comment(s)**

Because of the major commonalities and strong similarities in the argumentation of both reviewers, we have included the following points.

Reply:

A) Abstract

Significant changes have been made for the abstract to better reflect the clarifications and extensions of the revised manuscript.

- Shortening of a section, that is moved to the introduction (line 17-21)
- Inclusion of a sentence highlighting the novelty of the measurement technology (line 24)
- Inclusion of the Spearman correlation to better indicate the focus on the analysis of drivers and biogeochemical processes (line 26)

Taking into account also minor changes requested by RF1 and RF2, we provide the following new version of the abstract:

**"Abstract.** Rewetting peatlands is an important measure to reduce greenhouse gas (GHG) emissions from land use change. After rewetting, the areas can be highly heterogeneous in terms of GHG exchange and depend, for example, on water level, vegetation, temperature, previous use, and duration of rewetting. Here, we present a study of a coastal peatland that was rewetted by brackish water from the Baltic Sea and thus became part of the coastal shallow Baltic Sea water system through a permanent hydrological connection. Environmental heterogeneity and the brackish water column formation require improved quantification techniques to assess local sinks and sources of atmospheric GHGs. We conducted nine weeks of autonomous and high-resolution, sensor-based bottom water measurements of marine physical and chemical variables at two locations in a permanently flooded peatland in summer 2021, the 2$^{nd}$ year after rewetting. For the study, we used newly developed multi-sensor platforms (landers) customized for this operation. Results show considerable temporal fluctuations of $CO_2$ and $CH_4$, expressed as multi-day, diurnal and event-based variability, and spatial differences for variables dominantly influenced by biological processes. Episodic and diurnal drivers are identified and discussed based on Spearman correlation analysis. The multi-day variability resulted in a pronounced variability of measured GHG partial pressures during the deployment ranging between 295.0–8937.8 µatm ($CO_2$) and 22.8–2681.3 µatm (correspond to 42.7–3568.6 nmol L$^{-1}$; $CH_4$), respectively. In addition, the variability of the GHGs, temperature, and oxygen was characterized by pronounced diurnal cycles, resulting e.g., in a mean daily variability of 4066.9 µatm for $CO_2$ and 1769.6 µatm for $CH_4$. Depending on the location, the diurnal variability led to pronounced differences between the measurements during the day and night, so that the $CO_2$ and $CH_4$ fluxes varied by a factor of 2.1–2.3 and 2.3–3.0, respectively, with higher fluxes occurring over daytime. The rewetted peatland was further impacted by fast system changes (events) such as storm, precipitation and major water level changes, which impacted biogeochemical cycling and GHG partial pressures. The derived average GHG exchange amounted to 0.12 ± 0.16 g m$^{-2}$ h$^{-1}$ ($CO_2$) and 0.51 ± 0.56 mg m$^{-2}$ h$^{-1}$ ($CH_4$), respectively. These fluxes are high ($CO_2$) to low ($CH_4$) compared to studies from temperate peatlands rewetted with freshwater. Comparing these fluxes with the previous year (i.e., results from a reference study), the fluxes decreased by a factor of 1.9 and 2.6, respectively. This was potentially due to a progressive consumption of organic material, a suppression of $CH_4$ production, and aerobic and anaerobic oxidation of $CH_4$, indicating a positive evolution of the rewetted peatland into a site with moderate GHG emissions within the next years."**

B) Introduction

Significant changes are included in the introduction.

To set the topic into a broader context, we give more information in the first paragraph (from line 43).

"Mitigating climate change requires a reduction in anthropogenic emissions of the greenhouse gases (GHGs) carbon dioxide ($CO_2$) and methane ($CH_4$) and the effective removal of $CO_2$ from the atmosphere (IPCC 2022). In all climate scenarios with a realistic probability to reach the Paris Agreement, aiming to keep anthropogenic temperature increase "well below 2 °C" (IPCCSR15, IPCC2023), land use, land use changes and forestry (LUCLUF sector) play an important role. Still, a large part of the hard to abate residual emissions projected in these scenarios for the $2^{nd}$ half of this century come from the agricultural sector. Land use options with a large potential for climate mitigation include, for example, forestry, agriculture (pasture and cropland), wetlands, and bioenergy, (Roe et al, 2019, Shukla et al., 2019). In addition, in coastal areas, blue carbon options such as restoration and expansion of mangroves, salt marshes and seagrass meadows are suggested to have some potential for $CO_2$ removal (Macreadie et al., 2019, Duarte et al., 2013). The rewetting of formerly drained peatlands has been identified as one of the most promising approaches to lower $CO_2$ emission of used land, potentially even allowing turning (or re-establishing) some of these areas into $CO_2$ sinks (IPCC 2014, Wilson et al., 2016). Peatlands cover vast areas in partiuclar in Northern Europe, Northern Asia and western North America (Global Peatland Database / Greifswald Mire Centre (2024), and a large fraction of this area has been drained for agricultural use (UNEP (2022)).

Pristine peatlands and shallow coastal regions [...]"

Additional changes are made from line 66 onwards.

"Rewetting of degraded peatlands reduces $CO_2$ emissions by preventing aerobic decomposition of OM. The low solubility of $O_2$ and the slower transport across the overlying water body limits the availability of oxygen in the waterlogged peat soils for soil decomposition, which reduces aerobic mineralization and favors anoxic conditions, enhancing organic carbon burial (Parish et al., 2008; Kaat and Joosten, 2009). In the long-term, a re-establishment [...]"

In the sentence that begins in line 72, the following addition was made.

"However, the effects of brackish water on GHG emissions are still unclear, although beneficial effects such as lower $CH_4$ emissions compared to rewetting with freshwater are likely due to the availability of sulfate ($SO_4^{2-}$), a phenomenon better investigated for some coastal ecosystems, e.g. mangroves (Cotovicz et al., 2024). The availability of sulfate can promote the activity of sulfate-reducing bacteria (SRB)."

In the paragraph beginning in line 92, some minor changes are made to emphasize the novelty of the instrumentation we used, resulting in the following text.

"In this work, two newly developed, mostly identical lander systems were deployed, which are designed as autonomous platforms hosting a wide range of marine sensors. The landers were placed as fixed platforms on the sediment surface, and were customized for this deployment with cabled power supply and uninterrupted high-resolution data acquisition. The systems can [...]"

We would like to add more information from line 96 onward.

"We performed sensor measurements of the partial pressures of $CO_2$ and $CH_4$ and a suite of physicochemical variables (i.e., water temperature, salinity, water level, $O_2$ saturation, turbidity, phosphate, nitrate, water velocity, and chlorophyll *a*) with high temporal resolution in the range of seconds and minutes in a recently flooded peatland over a period of around nine weeks in the summer of 2021. The high-resolution measurements were combined with discrete sample analysis, and GHG emissions of $CO_2$ and $CH_4$ were derived.

   The rewetting of the coastal peatland [...]"

Due to reconstruction, the sentence in line 100 „The high-resolution measurements were combined with discrete sample analysis, and GHG emissions of $CO_2$ and $CH_4$ were derived." is deleted/is moved to another position

The paragraph beginning in line 101 is modified, resulting in the following text:

"The focus of this study is on exploring the time scales for the variability of GHG distribution and its drivers, as highly variable conditions are assumed. The nine-week time-series is used to derive main cyclic as well as episodic variability in $CO_2$ and $CH_4$ concentrations and fluxes, and relate it physicochemical drivers. The impact of the temporal variability on the estimation of GHG emissions or with respect to discrete sampling strategies is assessed. By comparing GHG fluxes with a study conducted one year earlier (i.e., 2020, the first year after rewetting; Pönisch and Breznikar et al., 2023), the potential evolution towards further weakening of the $CO_2$ and $CH_4$ source strength is discussed."

C) Emphasizing the biogeochemical interpretation before the technical description/measurement technique

With these points we implement the following major changes:

a.  We have changed the sequence of chapters 2.2 and 2.1 so that the study site and the study design are introduced first and then the technology.

   A new sentence must be included in line 147: "Two submersible landers equipped with sensors were used for autonomous multi-parameter investigations in the shallow water of the rewetted peatland through integrated high-resolution measurements."

   The first sentence in the line 109 must be shortened: "The two novel so-called landers are submersible platforms for advanced autonomous multi-parameter investigations in shallow water. The entirety of the carrier frame [...]"

b.  Moving section 4.5, which is indeed very technical and deals with the quality of the data and improvements for further missions, to the appendix. The new section will be "Appendix F: Assessment of the data quality and implications for future lander deployments" in line 863.

   In addition, the heading in line 787 is renamed to avoid confusion into „C3 Quality assessment of sensor GHG measurements."

   To assure the reader that we carried out a series of quality assessment measures, we would like to insert a short paragraph at the beginning of the discussion section

starting in line 438 with a link to the corresponding appendix F. Subsequently, we start the discussion of the biogeochemical findings.

"For the deployment of two novel landers in the complex and heterogeneous environment of a rewetted peatland, it was important to integrate strategies to assess the quality of the sensor data. Therefore, we have conducted various measures and analyses to build confidence in the sensor data, which are discussed in detail in Appendix F1 together with the future implications for the deployment of the landers. Despite the fact that quality assessment turned out to be complex, we can show that the sensor data are suitable for interpretation based on two main analyzes: First, the similarity of the main trends in the data series from both landers strongly suggests the appropriate sampling strategy for dynamic ecosystems. Second, with strong effort on discrete samplings and laboratory analysis, we observed both good agreements but also discrepancies compared to the sensor data. Still, with all quality measures applied, we were able to achieve a robust post-processing which allows comprehensive biogeochemical interpretation, which we address in the following."

**D) Discussion of our results on CO2 and CH4 fluxes in a broader scientific/existing literature**

a) We summarized GHG flux data from other environments at the terrestrial-marine interface, which will placed in the Appendix F2 "Selected greenhouse gas emissions of CO2 and CH4 along the land-sea-interface" in line 863:

*"Table 1: Selected greenhouse gas emissions of CO2 and CH4 along the land-sea-interface in relation to the derived GHG fluxes from our study.*

**Carbon dioxide fluxes**

| from land | from streams | from restored peatland(s) | from this study | from open shallow water (brackish/salty) |
|---|---|---|---|---|
| 0.07 g m$^{-2}$ h$^{-1}$ (drained unutilized land)[1] | −0.03−0.24 g m$^{-2}$ h$^{-1}$ (review with 34 study sites about streams in temperate Europe)[2] | 0.02 g m$^{-2}$ h$^{-1}$ (open water) to 0.09 g m$^{-2}$ h$^{-1}$ (emergent vegetation stands, Germany)[3]  −0.04 g $CO_2$ eq. m$^{-2}$ h$^{-1}$ (review of 38 restored peatlands)[4] | 0.12 ± 0.16g m$^{-2}$ h$^{-1}$ | 0.01 g m$^{-2}$ h$^{-1}$ (Bornholm sea)[5] 0.0007 g m$^{-2}$ h$^{-1}$ (Bothnian Bay)[6] |

References of adapted numbers: 1 Tiemeyer et al., (2020); 2 Mwanake et al., (2023); 3 Franz et al., (2016); 4 Bianchi et al., (2021); 5 Thomas and Schneider, (1999); 6 Löffler et al., (2012)

**Methane fluxes**

| from land | from streams | from restored peatland(s) | from this study | from open shallow water (brackish/salty) |
|---|---|---|---|---|
| 0.6 mg m$^{-2}$ h$^{-1}$ (drained unutilized land)[7] | 1.3–12.8 mg m$^{-2}$ h$^{-1}$ (Donau river, Germany)[8] | 1.48 mg m$^{-2}$ h$^{-1}$ (emergent vegetation stands) to 6.05 mg m$^{-2}$ h$^{-1}$ (open water, Germany)[9]  29.68 mg m$^{-2}$ h$^{-1}$ (occasional brackish impact)[10] | 0.51 ± 0.56 mg m$^{-2}$ h$^{-1}$ | 39.9–104.2 mg m$^{-2}$ h$^{-1}$ (June/July, shallow water of the Baltic Sea)[11]  0.015–0.024 mg m$^{-2}$ h$^{-1}$ (continental shelves)[12] |

| | | 3.2 mg m$^{-2}$ h$^{-1}$ (rewetted organic soils)[7] | | |
|---|---|---|---|---|
| | | | | |

References of adapted numbers: 7 Tiemeyer et al., (2020); 8 Lorke and Burgis, (2018); 9 Franz et al., (2016); 10 Hahn et al., (2015), 11 Heyer and Berger, (2000); 12 Bange et al., (1994)

"

b) We insert a new section "4.4.1 CO$_2$ and CH$_4$ fluxes at the land-sea-interface", in which the fluxes from our peatland are placed in a broader context, based on the Table 1.

Section 4.4 must be altered and shortened so that 4.4.1 can be added. The old text was changed from line 572.

"Greenhouse gas fluxes for CO$_2$ and CH$_4$ could be derived from the high-resolution sensor data as measurements were made narrowly below the water surface (< 1.25 m) and a direct coupling of water at the lander with the surface water was assumed. Although the peatland showed a slight CO$_2$ uptake in early June (Figure E 2a), accompanied by stable but slightly decreasing chlorophyll *a* concentrations, the ASE was clearly dominated by a flux of CO$_2$ to the atmosphere. This amounted to 0.12 ± 0.16 g m$^{-2}$ h$^{-1}$ derived from both landers. Increasing fluxes through early July stabilized with a simultaneous strong increase in chlorophyll *a* concentration. Overall, CO$_2$ emissions in the peatland were controlled by the simultaneous occurrence of primary production and mineralization, with the latter predominating for an overall net CO$_2$ outgassing. The derived CH$_4$ fluxes of 0.51 ± 0.56 mg m$^{-2}$ h$^{-1}$ showed a stable development during the measurement period with a slight trend to lower fluxes in August (Figure E 2b), also strongly controlled by mineralization processes of OM. In the following, we discuss these fluxes in the context of other coastal studies, also addressing the main findings of an apparently low CH$_4$ emission under brackish inundation, a decrease in summerly CO$_2$ and CH$_4$ emissions in comparison to the first year after inundation, and a strong diurnal cyclicity."

We add the section 4.4.1 "CO$_2$ and CH$_4$ fluxes at the land-sea-interface".

"We put our derived GHG fluxes into a broader context by comparing them with fluxes reported for different ecosystems along the land-sea interface, summarized in Table F 1.

CO2 fluxes from our study are around one order of magnitude higher than those reported from other restored peatland sites. For example, 9 years after flooding with freshwater, CO$_2$ emissions from a shallow lake formed on a formerly drained fen varied from 0.02 g m$^{-2}$ h$^{-1}$ (open water) to 0.09 g m$^{-2}$ h$^{-1}$ (emergent vegetation stands; numbers adapted from Franz et al., (2016)). In a recent review, Bianchi et al. (2020) derived a negative flux of -0.04 g m$^{-2}$ h$^{-1}$ as mean of 38 studies on restored peatland sites, supporting the interest of peatland restoration as a mean for climate mitigation.

Compared to land-based emissions, e.g., from drained unutilized land, cropland or forestry, the fluxes from our study area are a factor of two higher (Tiemeyer et al., 2020). Rivers and streams in the temperate latitudes of Europe, which are to a large extend anthropologically influenced, have GHG fluxes of approximately the same order of magnitude as our study area, ranging between −0.03 and 0.24 g m$^{-2}$ h$^{-1}$ (Mwangada et al., 2023). Open, shallow waters, which are directly connected to processes on land via runoff, and often influenced by human activities, are sometimes reported as sources of CO2. A comparison with these areas shows that the fluxes from the shallow waters of the

Baltic Sea or the North Sea are much smaller than those from the rewetted brackish peatland (Thomas and Schneider, 1994, Löffler et al., 2012).

The derived $CH_4$ fluxes of $0.51 \pm 0.56$ mg m$^{-2}$ h$^{-1}$ from our study are significantly lower than those reported for temperate fens rewetted with freshwater with comparable environmental settings (Hahn et al., 2011, Franz et al., 2016). For example, $CH_4$ emissions from the shallow lake mentioned above varied from 1.48 mg m$^{-2}$ h$^{-1}$ (emergent vegetation stands) to 6.05 mg m$^{-2}$ h$^{-1}$ (open water) even 9 years after rewetting (numbers adapted from Franz et al., (2016)). In another study, where a dry fen was converted to a shallow lake with occasional brackish water impact, a $CH_4$ flux of 29.68 mg m$^{-2}$ h$^{-1}$ was reported in the first year after rewetting (Hahn et al., 2015).

The derived $CH_4$ fluxes of our study are in the same range as those from drained unutilized $CH_4$ emissions from land (Tiemeyer et al., 2020). CH4 fluxes from the German section of the Danube River ranged between 1.3–12.8 mg m$^{-2}$ h$^{-1}$ (Lorke and Burgis, 2018), considerably higher that the fluxes derived in our study. A wide range of CH4 emission rates have been reported from shallow coastal waters; e.g., very high $CH_4$ fluxes have been reported from the shallow waters of the Baltic Sea in summer, ranging from 39.9–104.2 mg m$^{-2}$ h$^{-1}$, while fluxes from 0.015–0.024 mg m$^{-2}$ h$^{-1}$ have been reported for continental shelves (Bange et al., 1994; Heyer and Berger, 2020).

The high CO2 and low CH4 fluxes from our study compared to other studies of rewetted peatland support the hypothesis of GHG fluxes that are still high but decreasing after a major perturbation (see next section), but with generally low CH4 emissions compared to systems inundated with freshwater. While the theoretical background strongly supports the hypothesis of reduced CH4 fluxes in the presence of sulfate, the empirical evidence for rewetted peatlands is still sparse. However, for mangroves, another coastal ecosystem group with climate mitigation potential it has been recently shown that the offset of climate mitigation potential of mangroves was directly correlated with salinity, with lowest offset (i.e., lowest methane fluxes in relation to carbon sequestration) in high salinity regimes (Cotovicz et al. 2024). Based on the results of our study, the few studies on brackish water rewetting of peatlands, and studies on the blue carbon potential of coastal ecosystems, the question of rewetting of peatlands with brackish or freshwater might play an important role for the climate mitigation potential and should be considered where both options are possible."

Furthermore, we did some changes in the paragraph starting in line 600

"The comparison suggests that the $CO_2$ and $CH_4$ fluxes in the second summer after inundation were lower by a factor of 1.9 and 2.6, respectively compared to the first (**Fehler! Verweisquelle konnte nicht gefunden werden.**). A bias by the different sampling approaches appears negligible, as the fluxes calculated from the 9:00 to 15:00 time interval of the high-resolution data and the fluxes just based on the bottle data (low resolution but directly comparable with the earlier study from a methodological point of view) are very similar (Table 2). Although, as mentioned above, interannual variability rather than a trend cannot be excluded in the 2nd year after inundation, the reduction in $CO_2$ and $CH_4$ emissions is perfectly in line with the hypothesis of decreasing GHG fluxes in the years following the inundation. In general, the decay of vegetation from the former drained peatland and the decomposition of the organic-rich top soil, foster strong mineralization of OM and fuel $CO_2$ and $CH_4$ production (Heyer and Berger, 2000; Hahn-Schöfl et al., 2011). Our findings are in line with the results of a recent study on the longterm development of a rewetted fen in northeastern Germany (Kalhori et al., 2024). The authors monitored the $CO_2$ and $CH_4$ fluxes over a period of 13 years, and found a decreasing trend of $CO_2$ and $CH_4$ emissions, and a switch from a $CO_2$ source to a $CO_2$ sink after more than a decade. With decreasing availability of degradable OM, stabilization of the microbial communities and potentially the establishment of new vegetation characteristic for brackish wetlands, a further reduction in GHG emission is thus expected."

In line 620 is a sentence added, addressing the diurnal cyclicity:

"Consequently, studies on organic-rich flooded peatlands and shallow coastal areas [...] This is in line with the observations of diurnal cyclicity reported from other ecosystems important in the context of climate mitigation (e.g., Belikov et al., 2019; Huang et al., 2019; Metya et al., 2021), it is therefore recommended to adjust the sampling and calculation interval of the biogeochemical parameters as well as relevant parameters for flux estimation (e.g., wind speed) to resolve these internal frequencies."

To match the conclusion with the overall changes in the manuscript stated by the reviewers, the structure of the conclusion was adapted. Most of the changes resulted from the reorganization and only minor contextual changes were made. The conclusion is now divided into three parts: Technology Assessment, Findings and Implications. To make the conclusion easier to read, the entire section is copied here.

"The measurement of key marine physicochemical variables in a shallow brackish water column of a rewetted coastal peatland using sensor-equipped landers was, to the best of our knowledge, demonstrated for the first time. With the deployment of the sensor-equipped landers, new challenges arose, such as the extensive technical infrastructure required or the difficult conditions of the water-covered peatland. The latter has affected the reliability of the state-of-the-art sensors and led to lower data quality. However, this circumstance is less significant when considering the given amplitudes of the fluctuations of the system. With improvements, e.g., new pump generations, sensor measurements can become more reliable in the future. Overall, the lander-based autonomous monitoring technique used in this study was successful in capturing GHG variability in a shallow water environment.

The results showed that the rewetted peatland was characterized by a generally high system variability. In particular, GHG partial pressures were dominated by long-term (multi-day) variability and diurnal cycles. The short-term variability was pronounced for $pCO_2$ and $O_2$, as variables that are predominantly biologically influenced, but also the temperature and $pCH_4$ were affected by fast changes as well as diurnal cyclicity. Derived from these observations, pronounced differences between daytime and nighttime can be resolved. Hence, to achieve a comprehensive biogeochemical interpretation, and in particular for the derivation of GHG fluxes, this diurnal effect must be considered in shallow coastal waters and rewetted peatlands with a permanent water column.

The evaluation of GHG fluxes showed that the almost permanently oversaturated conditions for $pCO_2$ and $pCH_4$ mean that the peatland was a source of atmospheric GHGs. Comparing the fluxes to a study conducted in the same area one year prior to our study, emissions decreased by a factor of 1.9 and 2.6, respectively. Although limited

data comparability and potential interannual variability have to be acknowledged, the continued decline in GHG emissions, especially in the relatively low $CH_4$ emissions, is in line with the hypothesis of a positive evolution of the peatland into a potential C sink (or small $CO_2$-source) with comparably low $CH_4$ emissions under brackish water influence.

The rewetting of formerly drained peatlands as well as the extension of shallow coastal ecosystems are currently discussed as potential measures for GHG emission reduction or even the creation of net negative emissions (CDR in Blue Carbon schemes). Our study strongly suggests that the monitoring, reporting and verification (MRV) schemes for these measures might not only require integrated assessment of carbon burial and GHG emissions, but that the short-term variability and cyclicity require new observational approaches to resolve the appropriate time scales. Moreover, time-integrated monitoring - and accounting - for several years after perturbation (i.e., rewetting) is required to correctly assess the potential and true impact of peatland restoration and other land-use-change options for climate mitigation."

**2. Smaller comments from reviewer 1**

**Line 30)** *The diurnal variability led to a pronounced discrepancy between the measurements during the day and at night as well as depending on the location, resulting in CO 2 and CH4 fluxes that varied by a factor of 2.1–2.3 and 2.3–3.0, respectively.*

**The sentence is not fluent, and discrepancy might sound too negative for what you are describing.**

> Reply: Thank you for pointing this out. Differences between daytime and nighttime GHG fluxes were expected, but of this magnitude were a key finding of this study. Such differences make it clear that discrete samplings during daytime can lead to difficulties in interpretation. We make the following adjustments:
>
> "Depending on the location, the diurnal variability led to pronounced differences between the measurements during the day and night, so that the CO2 and CH4 fluxes varied by a factor of 2.1–2.3 and 2.3–3.0, respectively, with higher fluxes occurring over daytime."
>
> Furthermore, to make the figures easier to understand, we provided the information on the diurnal differences recorded by the landers in line 427:
>
> "Hence, atmospheric GHG fluxes are 2.1- (lander 2) and 2.3-fold (lander 1) higher for $p$CO$_2$ and 2.3- (lander 2) and 3.0-fold (lander 1) higher for $p$CH$_4$ during the day than at night (Table 2**Fehler! Verweisquelle konnte nicht gefunden werden.**)."

**Line 74)** *However, the effects of brackish water on GHG emissions are still unclear, although beneficial effects such as lower CH4 emissions compared to rewetting with freshwater are likely due to the availability of sulfate (SO42−)*

**One sentence to explain here why SO4 is beneficial would be good.**

> Reply: The information as to why SO4 is probably positive comes in the following sentences and is closely related to the triggered activity of SRB. However, we can understand your point and therefore we restructured the entire part, beginning from line 72 until the end of the paragraph:
>
> "However, the effects of brackish water on GHG emissions are still unclear, although beneficial effects such as lower CH$_4$ emissions compared to rewetting with freshwater are likely due to the availability of sulfate (SO$_4^{2-}$), which could promote the activity of sulfate-reducing bacteria (SRB). SRB can limit CH$_4$ production, as they outcompete methane-producing microorganisms (methanogens) for substrates (Segers and Kengen, 1998; Jørgensen, 2006; Segarra et al., 2013). Further, the availability of SO42− favors anaerobic oxidation of methane (AOM), which could keep CH$_4$ emissions low (e.g., Boetius et al., 2000; Knittel and Boetius, 2009)."

**Line 92) Is it possible to add one sentence, somehow describing what a lander is? Most people wont be familiar with this term. Is it a floating device? Ankered to the ground? Does it move around within a certain area? Approx size? I know that it is described in more detail later, but it helps readers to have a little bit of an idea already.**

> Reply: This is an important point, therefore we add the following in line 93: "In this work, two lander systems were used, […]. The landers were placed as fixed platforms on the sediment surface with cabled power supply."

**Line 105: In the Introduction, twice a reference to an earlier study is made (Pönisch and Breznikar et al., 2023). Since your study seems to be a follow up study of this earlier study, maybe the reader**

can be informed briefly what were the main findings of the first study were, and how your set up/goals were different (maybe more details in discussion, but maybe already 1 or 2 sentences in the Introduction). In this way, your article is readable without forcing the reader to look up the othr article.

Reply: This is a good hint. One main outcome was the comparison of the "brackish GHG fluxes" to "freshwater GHG fluxes". Therefore, we add two sentences after line 101:

"[…]. Pönisch and Breznikar et al. (2023) showed that $CO_2$ fluxes were high in the first year of rewetting with brackish water, while $CH_4$ fluxes were low compared to freshwater rewetting. Their study however relied on weekly to biweekly discrete water sampling and could not resolve variability on shorter time scales."

**Line 146: wrong reference formatting. This is visible at more places in article, please check this. (for example, line 433:**

*Moreover, we used the published data from (Pönisch and Breznikar et al., 2023)*

**instead of**

*Moreover, we used the published data from Pönisch and Breznikar et al., (2023)*

Reply: Thanks for the comment. The formatting will match the journal's guidelines in the typeset article. For clarification on this particular reference: This publication is a shared first-authorship with both first authors contributing equally to the work. The citation program presently used has trouble to correctly represent this, but it will be adjusted to match the journal's guidelines in the final typeset manuscript.

**Line 164: The CONTROS HydroC® CO2 only goes to 1000 uatm. I assume that the authors have ordered a different version/adaptation? Specify that.**

Reply: Thank you for your comment and attention to detail. The manufacturer's product data sheet states the measuring range as 200 - 1000 µatm, however, with the footnote that other ranges are possible on request. This is the case here: We use the regular version of the sensor, but have asked the manufacturer whether he can extend the calibration range until 6000 µatm. This is possible without technical adjustments, only with adaptation of the calibration points and calibration curve to the desired, extended range. This process was also done in another study and this is already stated in the appendix (see line 760): <<In a recent study, an NDIR unit was calibrated up to 25,000 µatm with only minor adjustments of the polynomial calibration curve (Canning et al., 2021).>>

**Line 171:  *The resolution of data acquisition was set to 5 minutes (min) for lander 1 and 10 min for lander 2:* Why the difference? Add a sentence to clarify this.**

Reply: The difference is due to a problem with the programming of the sensor or, better stated, a problem with the data processing unit (software problem). After discovering this during deployment, we decided to leave the settings as they were. There is no scientific reason for this. I inserted the following sentence after the sentence in line 172:

"[…] was set to 5 minutes (min) for lander 1 and 10 min for lander 2. A technical limitation led to the different recording interval."

**Line 314 (and other places): it is unclear which instrument measures pCH4 (ppm) and which measure cCH4 (nmol L), and which formula is used to convert between both. Add a reference or formula.**

Reply: Thanks for the comment. We are trying to solve this situation before line 314 by adapting the data handling section in line 259:

"Data processing, analysis and visualization were performed using R (R Core Team, 2022). The R packages that were used to calculate $p$CO2 (based on bottle CT and pH data), to convert $p$CH4 (measured by HC-CH4 sensors) to concentrations, and CH4 concentrations (derived from bottle data) to pCH4 are descried in Appendix D."

Furthermore, we added more information in Appendix D. We added a further sentence and a reference in line 844.

"The conversion between pCH4 and cCH4 was performed according to Wiesenburg and Guinasso (1979)."

**Line 497: white space missing**

Reply: Thanks. I also changed the number "12" to the word "twelve", as the numbers up to twelve are normally written as words.

**Line 565: Clear comparison to literature for fluxes. But why is this not done a little bit for concentrations pCH4 and pCO2. Are these concentrations high compared to sea water? Or (German) rivers? As also mentioned in the first general comment, place your results better in perspective.**

Reply: With the two major comments by you and by referee 2, we have changed parts of the discussion section to put the results in a broader context.

**Line 596: *The most important ones are*. → sentence incomplete**

Reply: Thanks. A punctuation point was wrong. The sentence is changed to: "The most important ones are slightly different sampling height, which was in Pönisch and Breznikar et al., 2023 ~ 20 cm below the water surface and ~ 60–90 cm in our study; different sampling approach; and possible inter-annual variations, which cannot be addressed with only two years."

**Line 600: *The comparison suggests that the CO2 and CH4 fluxes in the second summer after inundation were lower by a factor of1.9 and 2.6, respectively compared to the first (Table 2).***

**These are interesting numbers. Can they be compared to numbers from other studies? Could such a reduction of 1.9 to 2.6 really be real? The authors bring these numbers back in the Conclusion, so it is an important number. Discuss better.**

Reply: That's right, we also found these figures interesting, but they should be treated with caution. We also pointed this out in the subsequent discussion - interannual variability and different methods lead to complex comparability.

Unfortunately, a comparison with other studies is very difficult, as there are hardly any/no comparable boundary conditions for which a comparison would be appropriate.

We assume that the decline is in particular due to the progressive decomposition of organic material. OM can come from dead plant biomass, peat or newly formed biomass through primary production. As the Pönisch and Breznikar study identified this availability of OM as the main driver for high CO2 emissions, it seems realistic that a reduction in the availability of OM for biological processes will reduce emissions. In principle, the reduction seems realistic to us.

Overall, we think that the basic arguments of the discussion are already there, but need to be better linked in the text. We have therefore rearranged the section and linked it better. We have also added literature. We did changes from line 604

"Although, as mentioned above, interannual variability rather than a trend cannot be excluded [...]. In general, the decay of vegetation from the formerly drained peatland and the decomposition of the organic-rich topsoil, foster a strong mineralization of OM and promote $CO_2$ and $CH_4$ production (Heyer and Berger, 2000; Hahn-Schöfl et al., 2011). The observed decline between the first and second year is realistic, as this decomposition of OM was to be expected and represents a typical transition phase (Kalhori et al., 2024)."

3. **Technical comments from reviewer 2**

**Line 19 – "to" is not an appropriate connector in "unlike to rewetting" and to make a subtler point there is a logical inconsistency in saying "unlike x, y is less studied", either of these alternatives would be more natural: "compared with x, y is less studied" or "unlike x, y remains understudied".**

Reply: Thank you for this comment. We have changed the sentence in line 19 to:

"Compared with rewetting with freshwater, the effects of rewetting with brackish, sulfate-containing water are less studied, although positive effects are expected as sulfate-reducing bacteria may become established and might out-compete methane-producing archaea (methanogens) for substrates, resulting in lower CH4 emissions."

**Line 67 – grammar errors in "considered as one of the potent measure". "Considered to be" is the appropriate construction here and "measure" should be plural. But it would be more appropriate to use an indefinite article eg. "considered to be a potent measure".**

Reply: Thank you, we have corrected the error by changing the sentences in line 67 to:

"Rewetting of degraded peatlands reduces $CO_2$ emissions by preventing aerobic decomposition of OM and is considered to be a potent measure to mitigate global warming through land-use change."

**Line 105 – I have not seen this double author et al. style before. It seems you want to give equal credit to the first two authors? In any case, this is a question for the editors.**

Reply: This publication is a "shared first authorship" with both first authors contributing equally to the work. The citation style of "Pönisch and Breznikar et al. (2023)" will be adjusted to match the journal's guidelines in the final typeset manuscript.

**Line 108 - Perhaps a different term to 'lander' could be used since this term is not likely familiar to more terrestrial peatland ecologists.**

Reply: You may be right. Reviewer 1 also made a similar comment. We have therefore already inserted a sentence in line 93 ("The landers were placed as fixed platforms on the sediment surface with cabled power supply."). In addition, we make the following changes to the sentence from line 109 onwards.

"Two submersible platforms were used for autonomous multi-parameter investigations in the shallow water (~ 0.5 m) of the rewetted peatland through integrated high-resolution sensor measurements. The entirety of the carrier frame, the power supply, the technical units for sensor control and the sensors are referred to as landers and were deployed as stationary measuring units at the sediment-water surface. Each lander system was equipped with [...]"

**Line 113 – "customized deployments"? What information is meant to be conveyed here?**

Reply: The landers are highly modular in their equipment and programming. They can therefore be applied to various scopes, such as extreme events (storms), which require a particularly high measurement resolution. To convey this, we add the following to line 113:

"Sensor scheduling, time stamping, and data recording were centralized in the Data Processing Unit (DPU) and allow customized deployments, for example long-term deployments or shorter-term deployments during extreme events such as storms."

**Line 119 – is the indent necessary?**

Reply: We think that the list is helpful to quickly recognize important adjustments on the one hand and to signal important features of the deployment very quickly on the other hand. The final layout and level of indentation will be determined by the journal's guidelines in the typeset manuscript.

**Line 125 – annual cycle of what?**

Reply: The term can be confusing. Therefore, we change the sentences in line 125 to:

"The time of deployment was chosen based on the study of the annual cycle of GHG dynamics from the first year after rewetting and based on weekly to bi-weekly sampling (Pönisch and Breznikar et al., 2023), which indicated that the summer season is the most important and dynamic with respect to GHG-fluxes."

**Line 204 – "bottle data" is not the most appropriate term to refer to the manually collected dissolved gas concentration data, perhaps another phrase could be found.**

Reply: To clearly distinguish measurements based on manually collected water samples and sensor-based measurements, we want to clearly indicate their origin from the Niskin bottle sampling. Therefore, we want to keep the term "bottle data", which is also commonly used for manually sampled validation data in other hydrographic studies. However, we rearrange the paragraph from line 202 to introduce the bottle data sampling device first.

"To validate the sensor-based measurements, discrete field measurements were taken during the lander deployment at lander 1 (in the central peatland area) and at lander 2 (in the connecting channel). Undisturbed water was taken manually using a 5-L Niskin bottle. The bottle was deployed horizontally from a small working boat and its closure noted with an exact time stamp. Altogether 9-sampling sessions were carried out in the direct proximity to the landers (Table C 1; bottle data for all discrete sampled variables can be found at https://doi.org/10.1594/PANGAEA.964758; (Pönisch et al., 2024)). Water from Niskin bottle sampling was analyzed using established laboratory methods as described below."

**Line 206-207 – these 2 sentences are unnecessary.**

Reply: We think that in the context of the very strong spatial variability in the peatland water column (discussed e.g. in section 4.5) together with a shallow water column (~1 m) the information of a horizontally deployed Niskin bottle (itself being 0.4 m long) is very important. Furthermore, in a water column in which scientists can move around with waders, it cannot be readily assumed that a boat was used for sampling. To encourage good practice in follow-up studies, we want to stick to give those details in the methods section.

**Line 234 – GHGs should be $CO_2$ and $CH_4$**

Reply: We change the term "GHGs" into $CO_2$ and $CH_4$ in line 234:

"All involved variables (i.e., $CO_2$, $CH_4$, wind speed, temperature, salinity, atmospheric-equilibrium conditions) were averaged hourly to obtain more robust values and matching timestamps."

**Line 240 – perhaps more could be said about the Schmidt number and the linear interpolation.**

Reply: We change the text as follows, which gives both more detail on the interpolation and reference to the source used with detailed background on the Schmidt number itself:

"The Schmidt number was approximated by a linear interpolation in salinity between the freshwater and seawater values (Wanninkhof, 2014). The Schmidt number depends on the gas, the temperature and to a minor degree, the salinity of the water."

**Line 242 - Is Utö the most appropriate data source for atmospheric GHG concentrations? I presume the authors chose it because it is also situated in a Baltic coastal setting, however it is 800 km away. If this is the most appropriate measurement site for atmospheric concentrations, some justification for its choice could be included**

Reply: The ICOS Station Üto is the nearest high quality atmospheric measurement station reflecting marine air. Inland stations, of which there are only very few in higher proximity, are more prone to local land-derived sources. An analysis of the "offset" between the marine station Mace Head (Altlantic Air masses) and Utö is given in Appendix Figure 2 of Jacobs et al, 2021 (https://doi.org/10.5194/bg-18-2679-2021). We therefore believe that Utö is the best choice for our study site.

**Line 245 - awkward reference to the previous work here, consider rephrasing**

Reply: We rephrase the sentence in line 245:

"Moreover, the same parameterization was used in Pönisch and Breznikar et al. (2023). The use of the same parameterization facilitated comparison of the results of our work with this previous study."

**Line 248 – no need to mention the results table in methods.**

Reply: We have deleted the link.

**Line 251 - could there be a clearer way to explain this?**

Reply: We change the sentences from line 251 into:

"ASE derived from our high-resolution time series to represent the day-night bias. Daily averages for 00:00 UTC ± 1 hour and 12:00 UTC ± 1 hour were isolated (resulting in ~ 200 data points for each calculation)."

**Line 258 – since this data is presented before the fluxes should this section on the methods go before ASE**

Reply: Thank you for the comment, we change section 2.6 with 2.5 to put "Data processing and analysis" before "Air-sea exchange (ASE) calculation".

**Line 260 – presumably "descried" is a typo here.**

Reply: Correct, it should read "described" and is changed accordingly.

**Line 261 – for what scientific purpose were the correlations conducted?**

Reply: To make the purpose more clear, we have added the following sentence in line 262:

"Spearman correlation coefficients (rs) were calculated [...]. To identify potential drivers, processes and mechanisms of CO2 and CH4 variability, the presence or absence of (strong) correlations helps to identify and discuss potential causal relationships."

**Line 262 – save figure references for results.**

Reply: Agreed. The Figure 3 reference was removed from the methods section.

**Line 277 – no unit for salinity?**

Reply: We have followed the work of Millero, F.J. 1993, (What is PSU? Oceanography 6(3):67) and Millero, F.J. 2015 (History of the Equation of State of Seawater, https://doi.org/10.5670/oceanog.2010.21), and a summary exists here (http://www.coastalwiki.org/wiki/Salinity). It was stated: "The practical salinity scale was defined as conductivity ratio with no units" and "This definition is adopted by all national and international oceanographic organizations". On this basis we suggest to keep salinity without units.

**Line 295 – unnecessary clause**

Reply: We have changed to:

"Although occasionally low $O_2$ values were detected, measurements indicate a predominantly oxygenated water column, with slightly lower mean $O_2$ values at lander 2 (Figure 2f, Table 1)."

**Line 299 – "short-term"; not sure what is meant by "d; Sect. 3.2, Table 1"; how low are "lower values".**

Reply: We change the term "short-scale" into "short-term" in line 299.

The "d" in the term "d; Sect. 3.2, Table 1" is the abbreviation for day. As the definition is not necessary because it is generally valid, we have deleted the abbreviation.

In June, the pCO2 value was generally lower compared to the rest of the deployment, as can be seen from the smoothing line in the graph in Figure 2a "lower values". We have inserted "average" to make this more clear in line 299:

"Sustained lower average values occurred at the beginning of the deployment (early June), but then changed to on average higher values (> 1000 µatm) during most of the deployment at both locations."

**Line 303 – "strong negative correlation"**

Reply: As we have introduced various effect sizes with the Spearman correlation, we refer to this information for the strength of a correlation and must be linked there: The strength of a correlation must therefore be "with a strong effect size". If I also changed to "strong negative correlation", I would have "strong" twice in the sentence. We would therefore like to stick with the original sentence.

**Line 304 – the results might start with this comparison and also the part on the same comparison for $CH_4$**

Reply: With this comparison, we show how the sensor data behaves compared to the discrete data. As the results and biogeochemical features of the sensor-measurements should be the main focus, and both reviewers suggested to put a stronger focus on the biogeochemical results rather on technical considerations, we have decided to leave the statements on sensor comparison with the discrete data at the end of the respective paragraphs.

**Line 313 – rather than always referring to the landers, the locations could be used, since the location is what is of interest. This point applies throughout.**

Reply: Though the positions of the two landers is clearly described in the site description, we will introduce the site information ("central part"; "channel); at some places in the text to make it easier for the reader to follow. Thank you for the suggestion.

**Line 352 – Short-term variability of what?**

Reply: Thank you. We have changed the headline into:

"Short-term variability and diurnal cycles of the measured variables"

**Line 355-357 - This sentence belongs in methods.**

Reply: Agree. Please find the answer under the next comment.

**Line 364-366 – These 2 sentences also to methods**

Reply: We agree that both lines 355-357 and 364-367 should transfer into the methods. We have inserted a new paragraph after the line 267 in the section "2.6 Data handling and analysis"

"In order to show the diurnal cyclicity and the relationships between the variables affected by the diurnal cycles, the high-resolution measurement time series (lander 1 and lander 2) were divided into hourly bins and a mean value was calculated for each hour of the day, resulting in a diurnal distribution pattern. Furthermore, to show the magnitude of diurnal variability of the variables, the mean diurnal variability was calculated. For this purpose, we divided the high-resolution data into 24-hour intervals, each starting at midnight. Then, for each interval, the difference between the minimum and maximum was determined. Subsequent determination of the mean, minimum, and maximum yields an approximation of the magnitude of diurnal variability."

The sentences in section 3.2 are changed accordingly from line 384:

"The variables $pCO_2$, $pCH_4$, temperature, and oxygen showed pronounced short-term variability and diurnal cyclicity, [...]. The diurnal cyclicity and the relationships between the variables affected by the diurnal cycles was made visible by calculating the distribution of hourly mean values (see section 2.5). The distribution indicated that $pCO_2$ and $pCH_4$ showed an inverse character compared to temperature, $O_2$, and the wind speed (Figure 4). [...]"

The sentences in section 3.2 are changed accordingly from line 395:

"To show the magnitude of daily variability, the mean, minimum, and maximum of 24-hour intervals were calculated (see section 2.5) and summarized in Table 1. The mean daily range for $pCO_2$ of ~ 4000 µatm is substantial. [...]"

**Line 378 - why not just say hourly?**

Reply: Yes, we have changed "one-hour-parts" into "hourly sections" in line 378 and also in line 853.

"The distribution of the mean values of pCO2, pCH4, temperature, O2 and wind speed of hourly sections over 24-hours."

**Line 379 – why switch between quantile and percentile?**

Reply: Agreed. We change the text to use percentile throughout.

**Line 385 – "shared observation" is not really the right phrase here**

Reply: Yes, we have simplified as follows:

"All three events, although different in nature, […]"

**Line 408 – What caused this outflow? Some explanation would be interesting if there is one; the phrasing about the water level relative to the sensor is a little clumsy.**

Reply: The outflow occurs because the peatland is directly connected to the Kubitzer Bodden and this is connected to the Baltic Sea. If the water levels drop there, the peatland follows. In addition, local wind regimes can lead to an outflow or inflow. This has already been described in line 135. Nevertheless, we make a change and rephrase the sentence to make it less clumsy in line in line 408.

"In late July, we observed an outflow of water from the peatland towards the *Kubitzer Bodden.* The outflow caused a strong lowering in the water column, which drained large areas of the rewetted peatland and caused the pressure measurements of the CTD-O2 sensor to reach ~ 0 dbar."

**Line 416-419 – Is this not already in methods, if not it should be moved there.**

Reply: Yes, we streamlined the first paragraph of this section as follows

"GHG fluxes for $CO_2$ and $CH_4$ were derived from the entire high-resolution sensor data and from the different scenarios: from bottle data only, during daytime, during nighttime, and using data of a previous study to isolate GHG fluxes for a direct year-to-year comparison (Table 2)."

**Line 429-436 – this table heading seems more like a section of the methods.**

Reply: You are right. With the figures and tables, we follow the idea that the objects should be understandable as independent/stand-alone objects. That is why we have added more information in the caption so that the figures can be understood without searching in the manuscript. As the calculation is not so comprehensible, we would like to stick to the rather methodical description.

**Line 439 – tell the variables**

Reply: We have changed the sentence in line 439 to:

"The deployment of two landers equipped with sensors for the high-resolution determination of the marine variables pCO2, pCH4, temperature, salinity, hydrostatic pressure, $O_2$, turbidity, water velocity, $c(PO_4^{3-})$ and chlorophyll *a* in a coastal peatland revealed large temporal and spatial variations of the measured variables."

**Line 442 – A claim including the phrase "covering a comparable study area" is problematic when the study only includes measurements from 2 points, consider rephrasing.**

Reply: We understand the point. Therefore, we have rephrased into:

"To our knowledge, there is no study that covers a comparable environmental setting with a similar temporal data resolution."

**Line 518 - remove sentence starting "The effect of this…"**

Reply: We remove the sentence.

**Line 583-586 – this section is a little clunky**

Reply: Yes, we revised this section:

"Most of the studies mentioned above report annual GHG fluxes. Since our study only covers the summer months, comparability with these studies is limited, as annual $CO_2$ and $CH_4$ emissions are normally highest in the late summer months. To allow a more direct comparison of the development of greenhouse gas emissions at our study site, we used the published data from Pönisch and Breznikar et al. 2023."

**Line 596 – Grammar**

Reply: Thank you. We have changed the both sentences in line 596 into:

„In addition to a high degree of comparability due to comparable boundary conditions, this approach also has limitations. The most important of these are slightly different sampling height [...]"

**Additional changes:**

- Changes in the caption of Figure 4 (line 378) to make it easier to understand.
  "The daily cyclicity of the mean values of $p\text{CO}_2$, $p\text{CH}_4$, temperature, $O_2$, and wind speed derived from hourly binning. The box plots show the median and the 25 / 75 % percentiles. The whiskers indicate the 5 / 95 % percentiles, and the red points denote the mean values."
- Changes in the caption of Figure E 3 (line 853) to make it easier to understand.
  "The daily cyclicity of the mean values of $p\text{CO}_2$, $p\text{CH}_4$, temperature, $O_2$, and wind speed derived from hourly binning. The box plots show the median and the 25 / 75 % percentiles. The whiskers indicate the 5 / 95 % percentiles, and the red points denote the mean values."
- We would like to make an additional change to Figure A2 by pixelating the faces.